# The Hippo terminal effector YAP boosts enterovirus replication in type 1 diabetes

Shirin Geravandi[1,11], Huan Liu[1,11], Heena Pahwa[1,11], Murali Krishna Madduri[1,11], Farah Atawneh[1], Adib Miraki Feriz[2], Sahar Rafizadeh[1], Annabelle Elisabeth Kruf[1], Mona Khazaei[1], Pouria Bahrami[1], David Gotti[1], Mohamed Elawour[1], Ruth M. Elgamal ◉[3], Ausilia Maria Grasso[1], David Bund[1], Blaz Lupse[1], Zahra Azizi[1,4], Omar Zabad[1], Karim Bouzakri[5], Marc Horwitz ◉[6], Alberto Pugliese[7,8,9], Kathrin Maedler ◉[1,12] ✉ & Amin Ardestani ◉[1,10,12] ✉

Type 1 diabetes (T1D) risk has been associated with enteroviral infections, particularly coxsackieviruses B (CVB). Cellular host factors contributing to virus-induced islet autoimmunity remain unclear. We show that the Hippo pathway effector Yes-associated Protein (YAP) is markedly upregulated in the exocrine and endocrine pancreas of T1D and at-risk autoantibody-positive (AAb⁺) donors, along with its target CTGF. YAP expression correlates with CVB RNA presence, often in or near infected cells. YAP overexpression enhances CVB replication, islet inflammation, and β-cell apoptosis, whereas its inhibition halts viral replication in primary and immortalized pancreatic cells. In exocrine-islet co-cultures, CVB triggers YAP and target gene expression. In mice, chronic β-cell YAP expression impairs glucose tolerance, abolishes insulin secretion, and promotes β-cell dedifferentiation. Mechanistically, YAP, in complex with its transcription factor TEAD, induces its own negative regulator MST1. MST1 inhibition boosts viral replication and reduces β-cell apoptosis, constituting a negative feedback loop in which the reciprocal antagonism between YAP and MST1 balances viral replication and β-cell death during CVB infections. YAP is thus an important host factor for enteroviral amplification, offering a potential antiviral target in T1D.

Type 1 diabetes (T1D) is a multi-factorial inflammatory disorder characterized by the autoimmune destruction of insulin-producing pancreatic β-cells, mediated by immune cell recruitment and infiltration of the whole pancreas and the local release of pro-inflammatory cytokines and chemokines[1]. Eventually, this process leads to β-cell apoptosis, impaired insulin secretion and development of hyperglycemia[2]. Although genetic predisposition is a key determinant in the development of T1D, environmental factors play their part,

[1]University of Bremen, Islet Biology Laboratory, Centre for Biomolecular Interactions Bremen, Bremen, Germany. [2]Wellcome Trust Sanger Institute, Wellcome Genome Campus, Hinxton, Cambridge, UK. [3]Biomedical Sciences Graduate Program, University of California, San Diego, La Jolla, CA, USA. [4]Department of Molecular Medicine, School of Advanced Technologies in Medicine, Tehran University of Medical Sciences, Tehran, Iran. [5]UMR DIATHEC, EA 7294, Centre Européen d'Etude du Diabète, Université de Strasbourg, Fédération de Médecine Translationnelle de Strasbourg, Strasbourg, France. [6]Department of Microbiology and Immunology, University of British Columbia Vancouver, Vancouver, Canada. [7]Diabetes Research Institute, Department of Medicine, Division of Endocrinology and Metabolism, Miami, FL, USA. [8]Department of Microbiology and Immunology, Leonard Miller School of Medicine, University of Miami, Miami, FL, USA. [9]Department of Diabetes Immunology & The Wanek Family Project for Type 1 Diabetes, Arthur Riggs Diabetes & Metabolism Research Institute, City of Hope, Duarte, CA, USA. [10]Biomedical Institute for Multimorbidity (BIM), Centre for Biomedicine, Hull York Medical School (HYMS), University of Hull, Hull, UK. [11]These authors contributed equally: Shirin Geravandi, Huan Liu, Heena Pahwa, Murali Krishna Madduri. [12]These authors jointly supervised this work: Kathrin Maedler, Amin Ardestani. ✉e-mail: kmaedler@uni-bremen.de; Amin.Ardestani@hyms.ac.uk

either as potential triggers, or accelerators. Enteroviruses, and especially Coxsackievirus B (CVB) strains, have been linked to increased T1D risk, and are suspected to play a role in the initiation and progression of islet autoimmunity[3,4]. Enteroviruses are small, non-enveloped, positive single-stranded RNA viruses of the Picornaviridae family[5]. CVBs are highly effective in infecting isolated human islet cells; their RNA and capsid protein were found in both the endocrine and exocrine pancreas of biopsies from living adults with recent-onset T1D as well as in organ donor pancreata from individuals with T1D. The presence of viral proteins and RNA is associated with MHCI-hyperexpression by islet cells, local inflammation and β-cell destruction[6–13]. In this context, recent findings from the most extensive multi-laboratory, multi-approach study by the Network for Pancreatic Organ Donors with Diabetes (nPOD) virus group reveal a strong association between the enteroviral capsid protein VP1 and residual β-cells in both preclinical and diagnosed T1D. VP1 positivity and islet human leukocyte antigen I (HLA-I) hyperexpression were observed during the autoantibody-positive stage, supporting the hypothesis that enteroviral infections may contribute throughout T1D progression and promote islet-specific HLA-I upregulation[14]. Collectively, epidemiological and tissue studies suggest that persistent, low-grade enteroviral disposition in the pancreas may contribute to T1D pathogenesis[15,16]. In quiescent cells, viral RNA can retain[17] for many years, even in the absence of infectious virus production, and CVBs remain persistent in T1D[15], causing constant inflammation[9].

While most studies have exclusively investigated enteroviral expression within islets, CVB infection and enteroviral RNA have also been reported in the exocrine pancreas in donors with T1D[8,9] as well as a preferential exocrine infection observed in mice[18]. By using a single molecule-based fluorescent in situ hybridization (smFISH) method[19], we have recently shown that enteroviral RNA is substantially increased in pancreases from organ donors with T1D and with disease-associated autoantibodies (AAb+) with the majority of virus-positive cells scattered in the exocrine pancreas[9]. Infected regions outside of islets are wired by immune cells and may constitute a potential reservoir for the ongoing inflammation to spread to islets.

Pancreatic enteroviral disposition may contribute to the development of T1D by various mechanisms, including initial direct destruction of β-cells due to virus infection; viral persistence and chronic stimulation and recruitment of immune cells to the islets to promote local inflammation, β-cell injury and subsequent release of autoantigens, which then trigger autoreactive T-cell responses ultimately mediating "bystander damage"[18] and β-cell death[3,15]. Another possible mechanism is "molecular mimicry", in which immune reactivity is driven by similarity of viral and β-cell epitopes. Similar hypotheses are applicable to autoimmune diseases in general, but presently it is unclear whether viruses directly initiate autoimmunity and target cell destruction or only accelerate this process[20].

In order to efficiently replicate, viruses hijack the cellular machinery and signaling pathways. While external and internal receptors for enterovirus entry and sensing are known[21,22], the endogenous host factor(s), their regulation in response to virus infections, and the molecular mechanisms which lead to excessive stimulation of the immune system remain elusive. Pathways which regulate host's cellular survival and proliferation may allow a virus to attack the cell replication machinery.

Hippo signaling represents an evolutionarily conserved pathway that controls organ size, tissue homeostasis, and cellular survival; it has been linked to the pathophysiology of cancer and metabolic diseases[23,24]. Yes-associated protein (YAP) is the transcriptional co-regulator and major terminal effector of the Hippo pathway. The activity of YAP is mainly regulated through a phosphorylation-dependent inhibition mechanism by the Hippo central kinases, mammalian STE20-like protein kinase 1 and 2 (MST1/2) and large tumor suppressor 1 and 2 (LATS1/2). Upon MST1/2 activation by physiological or pathological signals, MST1/2 phosphorylate and activate the LATS1/2 kinases, which in turn directly phosphorylate YAP on multiple sites, leading to YAP inactivation through its cytoplasmic retention and/or its degradation by the proteasome machinery[25]. In contrast, when Hippo signaling is inhibited, YAP can freely translocate into the nucleus where it interacts with several different transcription factors such as the TEA domain family members (TEAD) and stimulates the expression of genes responsible for cell turnover, differentiation and regeneration[23]. The Hippo pathway has major control over pancreas development and β-cell survival, regeneration and function[26–28]. YAP is broadly expressed in pancreatic progenitor cells in the developing pancreas and is indispensable for pancreatic cell identity through directing cell fate decisions and organ morphogenesis[29,30]. While YAP's presence is maintained in the exocrine pancreas and is essential for its function and plasticity, its expression is very low or undetectable in terminally differentiated, adult endocrine islet cells[31–33]. Importantly, we and others have previously shown that re-expression of active YAP induces human β-cell proliferation, indicating that the absence of YAP in adult human β-cells correlates with their low-replication capacity and β-cell quiescence[33,34]. YAP is also linked to innate immunity to balance host antiviral immune responses[35,36].

Here, we show YAP as a dysregulated factor and initiator of the immune disbalance in T1D, and its functional significance in enteroviral replication starting in the exocrine pancreas and promoting islet inflammation and β-cell apoptosis.

## Results

### YAP is highly upregulated in the pancreas of T1D and AAb+ organ donors

Based on the fact that YAP is expressed in the human exocrine pancreas and directly linked to innate immunity and host inflammatory responses, we first examined the endogenous expression of YAP in the exocrine pancreas. Immunohistochemistry (IHC) for YAP was performed and analyzed in paraffin-embedded pancreatic tissue from organ donors with T1D (n = 15), AAb+ (n = 15) and age and BMI-matched non-diabetic controls (n = 13) from the well-characterized cohort of organ donors from nPOD (Network for Pancreatic Organ Donors with Diabetes; Table S1)[37]. YAP protein expression, represented as %YAP-positive area in the exocrine pancreas, was significantly higher in T1D (mean 19.95%) than in AAb+ (mean 14.09%) and nondiabetic individuals (mean 11.97%) (Fig. 1A, B). The increased YAP expression was uniform across whole pancreas sections (Figure S1 for larger pancreas scans). This increase in YAP-positive area in T1D donors was also confirmed when analyzed as the mean per donor (Figure S2A). Moreover, a modest but significant increase in YAP-positive area in exocrine regions was also observed in AAb+ donors compared to nondiabetic controls (Fig. 1A, B). Consistent with previous findings[31,38], ductal and terminal-duct centro-acinar cells expressed the highest levels of YAP in the exocrine pancreas (Figure S2B). The majority of AAb+ and T1D donors abundantly expressed YAP within centro-acinar and ductal cells, while much less ductal YAP expression was observed in non-diabetic controls (Figure S2B).

YAP expression is minimal or absent in endocrine cells, including β-cells[33,34]. To investigate whether intra-islet expression of YAP in T1D is increased, we quantified the number of YAP-positive cells within the islet area. The frequency of YAP-positive cells was significantly higher in islets from T1D donors (mean 3.05%) compared to both AAb+ donors (mean 1.78%) and non-diabetic controls (mean 0.64%) (Fig. 1C, D, and S2C), with a moderate increase also observed in AAb+ donors relative to controls. To determine which cell types were YAP-positive in AAb+ and T1D islets, tissue sections were stained for YAP and chromogranin, a late endocrine marker. Consistent with the higher intra-islet YAP expression observed in T1D donors, also the percentage of YAP/chromogranin double-positive cells was significantly higher in islets from T1D donors (mean 0.62%) than in AAb+ (mean 0.10%) or

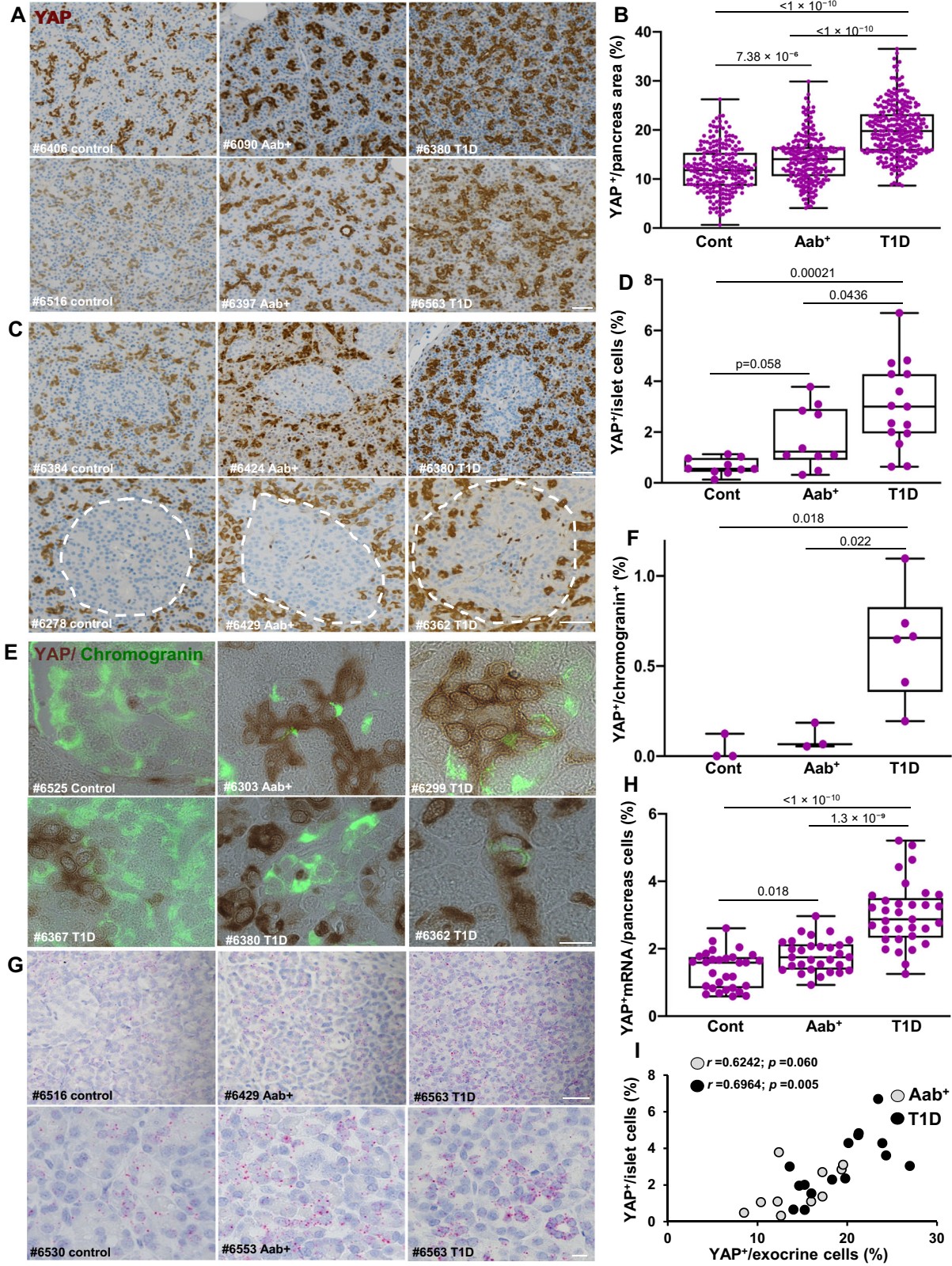

nondiabetic (mean 0.04%) donors (Fig. 1E, F). A comparison of YAP-positive cells in islets (Fig. 1D) with those co-stained for chromogranin (Fig. 1F) suggests that up to 19% of the YAP-positive cells in islets originate from endocrine cells in T1D.

The higher YAP protein abundance was paralleled by elevated *Yap1* mRNA expression. As determined using the highly sensitive in situ hybridization (ISH) RNAscope method, *Yap1* mRNA levels were significantly increased in donors with AAb+ (mean 1.8 puncta per cell) and T1D (mean 3.01) compared with nondiabetic controls (mean 1.37) (Fig. 1G, H). In addition, the expression of *Yap1* was higher in pancreases from T1D compared to AAb+ donors (Fig. 1G,H). Importantly, exocrine YAP levels highly correlated with endocrine YAP expression in T1D (r = 0.6964; p = 0.005) donors, while there was a similar trend in AAb+ (r = 0.6242; p = 0.060) (Fig. 1I). These data indicate an association

**Fig. 1 | YAP is highly upregulated in the pancreas of T1D and AAb⁺ organ donors.** YAP protein and *Yap1* mRNA labeling were analyzed in FFPE sections of pancreases from 13 control, 15 AAb⁺ organ donors without diabetes and 15 donors with T1D from the nPOD pancreas collection. **A**, **B** Representative images from different donors (**A**) and quantification (**B**) of the percentage of YAP⁺ area in the exocrine pancreas from FFPE sections of control donors without diabetes ($n = 229$ independent positions from 13 donors), donors without diabetes but expressing T1D-associated autoantibodies (AAb⁺) ($n = 223$ independent positions from 15 donors), and donors with T1D ($n = 284$ independent positions from 15 donors). **C**, **D** Representative images (**C**) and quantification (**D**) of the percentage of YAP⁺ cells within islets of controls ($n = 10$), AAb⁺ ($n = 10$), and donors with T1D ($n = 15$) of the number of islet cells. **E**, **F** Representative images (**E**) and quantification (**F**) of YAP (brown), and late endocrine marker chromogranin (green) double-positive cells from controls ($n = 3$; 16671 islet cells), AAb⁺ donors ($n = 3$; 14237 islet cells), and

donors with T1D ($n = 6$; 15116 islet cells). **G**, **H** Representative images (**G**) and quantification (**H**) of *Yap1* mRNA (pink) by RNAscope in situ hybridization of controls ($n = 30$ independent positions from 3 donors), AAb⁺ donors ($n = 30$ independent positions from 3 donors), and donors with T1D ($n = 33$ independent positions from 3 donors). **I** Association of YAP protein expression between endocrine islets and exocrine pancreas in AAb⁺ ($n = 10$; grey circles) and in donors with T1D ($n = 15$; black circles). All box plots showing single analytes and median (box and whiskers; min to max show all points). **A**, **C**, **G** Sections were counterstained with Hematoxylin. Data are expressed as means ± SEM. *P*-values were calculated by one-way ANOVA with Holm-Sidak multiple comparisons correction for (**B**, **D**, **F**) and by two-tailed unpaired Student *t*-test (Spearman) for (**I**). Scale bars depict 50 μm (**A**, **C**, **G**-upper panel) and 10 μm (**E**, **G**-lower panel). Source data are provided as a Source Data file.

of YAP upregulation as common modulator in both pancreas compartments with T1D; not only in islets but also in the exocrine pancreas. YAP expression and patients' clinical parameters revealed no correlation between YAP and age, BMI or Hb1AC in AAb⁺ and T1D donors (Figure S2D–F). We then performed sub-cluster analyses of the same AAb⁺ pancreata, which included four multiple AAb⁺ donors (three double AAb⁺ and one triple AAb⁺) and 11 single AAb⁺ donors (Table S1). In the exocrine pancreas, the %YAP-positive area did not differ between single and multiple AAb⁺ donors. In contrast, the number of intra-islet YAP-positive cells was significantly higher in multiple AAb⁺ compared to single AAb⁺ donors (Figure S3A, B). Also, analysis of β-cell area revealed a significant reduction in multiple AAb⁺ donors compared to single AAb⁺ donors (Figure S3C). As hypothesized from data in our previous analysis of donor pancreata[9], we found a significantly increased presence of CD45-positive immune cells within islets in multiple AAb⁺ donors compared to single AAb⁺ donors, normalized to both islet number or β-cell area in the pancreas (Figure S3D, E). These data indicate that YAP-positive cells in islets increase in multiple AAb⁺ donors compared to single AAb⁺ donors, positively correlating with the loss of β-cell area and presence of CD45-positive immune cells within islets.

## YAP target genes are upregulated in the pancreas of T1D organ donors

YAP upregulation does not necessarily indicate activation of its downstream transcriptional program. To determine whether YAP/TEAD signaling is functionally active in the pancreas during T1D, we re-analyzed single-cell RNA-seq (scRNA-seq) data from the Human Pancreas Analysis Program (HPAP) (Fig. 2)[39]. UMAP embedding revealed distinct cell clustering by diabetic status (Non-Diabetic, ND; T1D), sample origin (nPOD, UPenn), and annotated cell types (Fig. 2A–C). This integrated visualization highlights the cellular heterogeneity of the human pancreas and enables cross-comparison of disease states across independent datasets (Fig. 2). In β-cells from individuals with T1D, Gene Ontology (GO) Biological Process enrichment analysis identified top-ranked pathways linked to T cell-mediated cytotoxicity and antiviral defence responses (Fig. 2D). Similarly, pathways previously implicated in T1D - including inflammatory signaling, innate immunity, and apoptosis - were significantly upregulated in β-cells from individuals with T1D (Fig. 2E), confirming that this dataset recapitulates known β-cell transcriptional alterations in T1D. Importantly, Gene Set Enrichment Analysis (GSEA) revealed that the mRNA-based Hippo pathway score and YAP target genes (using a curated set of 22 well-established genes[40]) are significantly enriched not only in β-cells but also in α-cells. This is also observed in major YAP-expressing pancreatic cells, including exocrine ductal and pancreatic stellate cells, in the pancreas of individuals with T1D compared to the non-diabetic group (Fig. 2F). Thus, the elevation of YAP signature genes in the pancreas of individuals with T1D indicates that YAP is functionally active in several cell types within the pancreas.

To confirm this, we performed RNAscope analysis for connective tissue growth factor (CTGF), a well-established YAP target gene[41] that was also included in the GSEA analysis. Expression analysis throughout the pancreas revealed a significant upregulation of CTGF-positive cells in both AAb⁺ and T1D pancreases, similar to YAP. This was presented as an independent position capturing donor heterogeneity (Fig. 3A) or as the mean per donor (Fig. 3B). Also, the mean number of CTGF puncta per cell -categorized as 5-15 puncta per cell or clustered CTGF (>15 puncta per cell)- was markedly higher in T1D and AAb⁺ donors compared to controls, where such was rarely seen (Fig. 3C). In T1D, a general increase in CTGF expression throughout the pancreas, in terms of CTGF⁺ puncta per cell and their colocalization with YAP⁺ cells were particularly evident (Fig. 3D). Altogether, these findings show not only elevated YAP expression, but also its activity in the T1D pancreas.

## YAP colocalizes and correlates with enteroviral RNA expression in the pancreas

Recent research indicates that YAP plays a complex and bidirectional role in regulating innate immunity. On one hand, it balances inflammation and host's antiviral immune responses supporting cellular survival during infection[35,36]. On the contrary, YAP can also drive inflammation and activate pro-inflammatory pathways[38,42–45]. To determine YAP's complex role in pancreatic inflammation and its association with enteroviral infection in the pancreas of AAb⁺ and T1D donors, we analyzed YAP's cellular colocalization with two diabetogenic β-cell-tropic strains of CVB; CVB3 and CVB4 (CVB3/4) RNA. Double ISH-RNA analysis of *Yap1* and CVB3/4 RNAs allowed us to systematically localize and quantify RNA throughout the whole pancreas sections. Due to the expected absent/very low number of virus-positive cells in the control group[9], such analysis was only possible in AAb⁺ and T1D donors. Using single-cell analysis of CVB3/4 RNA and *Yap1* mRNA staining, we categorized infected cells into three groups: 1) cells with both YAP and viral RNA present in the same cell ("YAP⁺/CVB⁺"), 2) cells with viral RNA present in cells in close proximity of neighbor YAP-positive cells ("n-YAP⁺/CVB⁺") and 3) cells with no YAP but positive for viral RNA ("YAP⁻/CVB⁺"; Fig. 4A and S4). *Yap1* mRNA and enteroviral RNA mainly colocalized in the same cell, or Yap-positive cells were in close proximity to infected cells (Fig. 4A-C). The number of YAP⁺/CVB⁺ cells was significantly higher than both n-YAP⁺/CVB⁺ and YAP⁻/CVB⁺ cells in both AAb⁺ and T1D donors (Fig. 4C), suggesting that YAP expression is indeed induced in CVB-infected cells. Direct comparison of YAP⁺/CVB⁺ cells between human pancreas donors also confirms their increase in T1D; the mean number of YAP⁺/CVB⁺ cells with clustered viral RNA (>10 puncta per cell) were markedly higher in T1D than AAb⁺ donors (mean, 29 in T1D versus 7 in AAb⁺ for cluster infections; Fig. 4D). This confirms the increase in viral RNA reported previously by us using smFISH[9]. In addition to their cellular co-expression, YAP expression in the exocrine pancreas showed trends of positive correlation with the number of virus-expressing cells within the same region in AAb⁺ ($r = 0.6193$; $p = 0.08$) and T1D donors

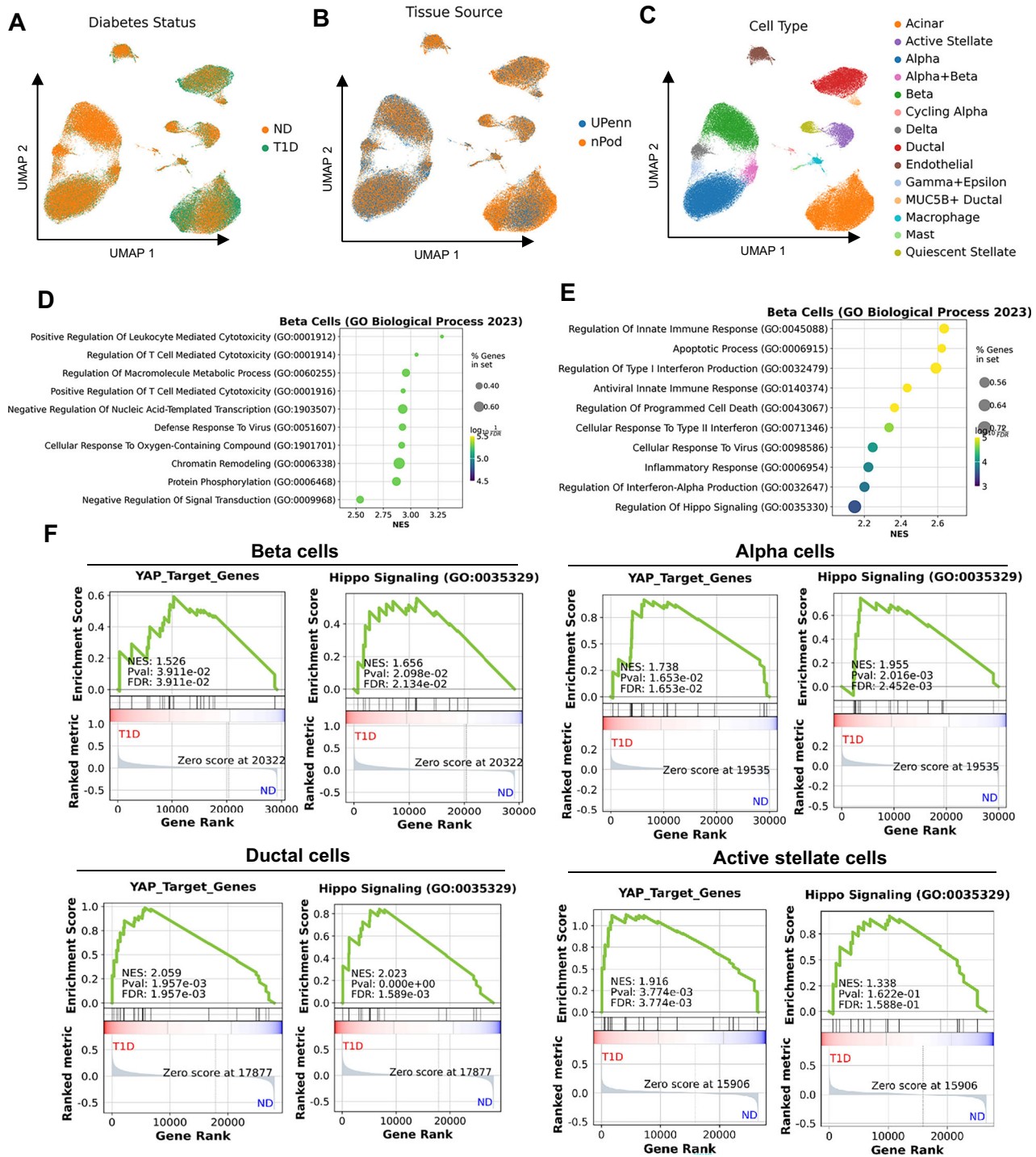

**Fig. 2 | Re-analysis of scRNA-seq data from pancreas of healthy and type 1 diabetic donors. A** UMAP visualization of cells colored by diabetic status (ND = Non-Diabetic, T1D = Type 1 Diabetes). **B** UMAP showing the sample sources from the Gaulton study (nPOD: Network for Pancreatic Organ Donors with Diabetes, UPenn: University of Pennsylvania). **C** UMAP representation of original study-defined cell types. **D** Dot plot of the top ten significantly enriched GO Biological Process (GO BP) terms in β-cells (T1D vs. ND). **E** Dot plot of T1D-relevant pathways (GO BP) significantly enriched in β-cells (T1D vs. ND). **F** GSEA of YAP target genes and Hippo signaling pathway in β-, alpha-, ductal-, and activated stellate cells. GSEA was performed using the GSEAPY tool with the GO Biological Process 2023 gene sets, applying a Kolmogorov−Smirnov-like enrichment score. Statistical significance was assessed via permutation testing followed by false discovery rate (FDR) correction.

($r = 0.5149$; $p = 0.06$; Fig. 4E). To confirm YAP-virus colocalization in the pancreas of AAb+ and T1D donors at a single cell level, we complemented classical YAP-IHC staining with enteroviral RNA smFISH, which our laboratory have previously established to identify and localize enteroviral RNA in pancreata[19]. In line with CVB3/4-YAP RNA expression, YAP-protein/viral RNA double-positive cells were detected in AAb+ and T1D donors (representative images shown in Fig. 4F). Many infected pancreatic cells expressed YAP, while YAP was not expressed in the single enteroviral RNA+ cell found in control. Comparative analysis of YAP protein-enteroviral RNA co-positive cells showed an

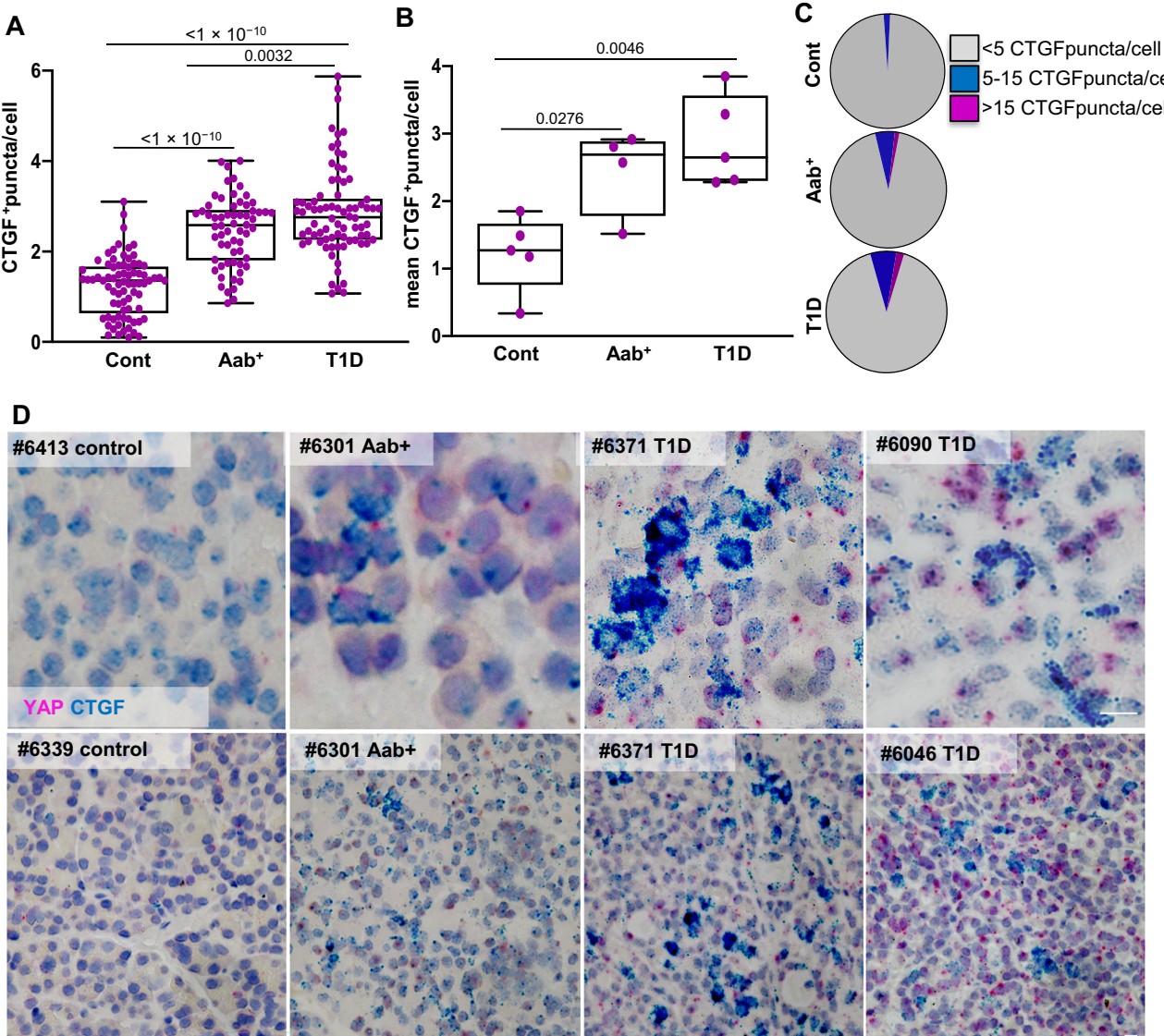

**Fig. 3 | CTGF is upregulated in the pancreas of T1D and AAb⁺ organ donors.** RNAscope in situ hybridization for CTGF (turquois) was performed on controls ($n = 76$ independent positions from 5 donors), AAb⁺ donors ($n = 62$ independent positions from 4 donors), and donors with T1D ($n = 78$ independent positions from 5 donors); double RNAscope was performed for YAP (pink). Quantification of CTGF⁺ puncta/cell is presented as (**A**) all independent positions, (**B**) the mean value for each analysed pancreatic section from each donor, and (**C**) the percentage of cells with <5 (grey), 5-15 (blue) and >15 CTGF⁺ (pink) puncta. **D** Representative images display double RNAscope for *CTGF* (turquois) and *Yap1* mRNA (pink), counter-stained with Hematoxylin, shown larger (upper) and smaller (lower) magnification. Both box plots showing single analytes and median (box and whiskers; min to max show all points). Data are expressed as means ± SEM. *P*-values were calculated by one-way ANOVA with Holm-Sidak multiple comparisons correction. Scale bars depict 10 μm. Source data are provided as a Source Data file.

increased numbers of both YAP⁺/CVB⁺ as well as n-YAP⁺/CVB⁺ in T1D compared to AAb⁺ donors (Fig. 4G).

To determine whether pancreatic YAP hyperexpression is exclusive to the human pancreas or represents a broader feature of diabetes development, we analyzed active (non-phosphorylated) and total YAP protein expression in the NOD mouse pancreas at the stage of early insulitis and mild hyperglycemia. In non-diabetic wild type C57Bl/6 J mice, we observed no YAP expression in islets but confirmed both nuclear and cytosolic YAP expression in ductal cells (Figure S5). In contrast, single YAP-positive cells were clearly detectable in NOD mouse islets, particularly at sites of insulitis. However, no notable increase in YAP expression was observed in the exocrine pancreas, and overall expression levels were much lower compared to the human pancreas (Figure S5). CVB accelerates diabetes in NOD mice, with mice developing diabetes as early as one-week post-infection[18,46]. This makes CVB-infected NOD mice an ideal model to investigate whether

pancreatic YAP expression contributes to CVB-induced diabetes. Notably, YAP-positive cells increased seven days after CVB3 or CVB4 infection, particularly in and around inflamed islets (Figure S5). These findings suggest that YAP expression is a pathological feature in T1D and is further enhanced by CVB infection.

Together, these results in the pancreas suggest a pathological association between YAP and enteroviruses and raises the question, whether the presence of YAP rather induces than balances enteroviral replication, and/or whether the infection per se may be a principal inducer of *Yap* transcription.

## CVB infection induces YAP expression and hyper-activity in islet-exocrine co-cultures

To determine whether infection induces YAP and its target genes in a human pancreas in vitro model, we analyzed endogenous *Yap1* expression and the regulation of its target genes in response to CVB

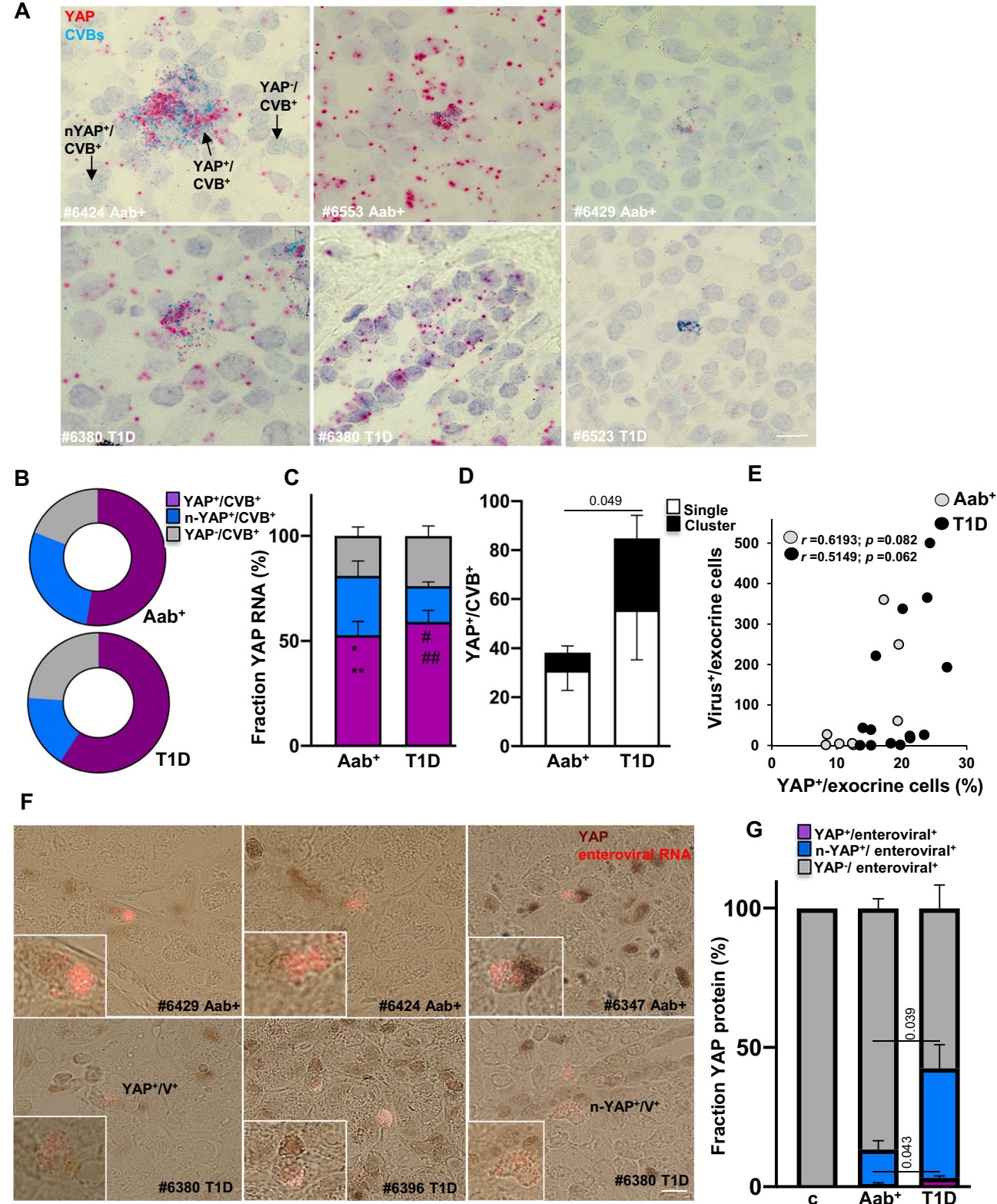

infection in islet-exocrine co-cultures from the same donor (Fig. 5A). This resulted in a transient upregulation of *Yap1* mRNA within 6 hours post-infection, which persisted for up to 12 h. At this point, YAP target genes *AMOTL2, ANKRD1, and CTGF* were also upregulated (Fig. 5B). By 24 hours post-infection, the expression levels of *Yap1* and its target genes returned to baseline (Fig. 5B). CVB-induced *Yap1* and *CTGF* upregulation was confirmed by double RNAscope for Yap1 and *CTGF*

(Fig. 5C). CVB infection led to YAP upregulation in human islets, correlating with dense CTGF expression alongside YAP, suggesting increased YAP activity. YAP-transduced human islets were used as a positive control to confirm YAP overexpression and the resultant upregulation of CTGF (Fig. 5C). To further support CVB-induced transcriptional upregulation of YAP target genes, we re-analyzed publicly available bulk RNA-seq data from human stem cell-derived β

**Fig. 4 | YAP colocalizes and correlates with enteroviral RNA expression in the pancreas. A–D** Detection and quantification of *Yap1* mRNA (pink) and viral RNA-CVB3/4 (turquois) by RNAscope in situ hybridization from FFPE nPOD pancreas sections of AAb+ (*n* = 9) and T1D donors (*n* = 10). **A** Representative images of *Yap1*/CVB-RNA double labelling from AAb+ and T1D pancreatic sections and (**B**, **C**) total distribution and quantification throughout the whole pancreas section differentiated in YAP-viral RNA double positive cells (YAP+/CVB+; purple), CVB-positive cells in close proximity of YAP-positive neighbor cells (n-YAP+/CVB+; blue) or YAP-negative but CVB-RNA-positive cells (YAP-/CVB+; gray). **D** Quantification of all viral RNA-positive cells throughout the whole pancreas section in AAb+ and T1D donors presented as the mean number of single (white; 5-10 single puncta/cell) or cluster (black; >10 single puncta/cell) infected cells. **E** Association between YAP protein expression and number of enterovirus-positive cells by smFISH for enteroviral RNA detection in AAb+ (*n* = 9) and T1D (*n* = 14) donors. **F** Representative microscopical images of enteroviral RNA (red; Stellaris probes) and YAP protein (brown; IHC) expression in the pancreas showing YAP+/Enterovirus+ cells (YAP+/V+) and enteroviral positive cells in close proximity of YAP-positive neighbor cells (n-YAP+/V+). **G** Quantification of YAP-protein+/viral smFISH+ cells (YAP+/CVB+; purple), viral smFISH+ cells in close proximity of YAP-positive neighbor cells (n-YAP+/CVB+; blue) or YAP-negative but viral smFISH+-positive cells (YAP-/CVB+; gray) throughout the whole pancreas; control represents only one single enteroviral RNA+ cell, which was YAP-negative (*n* = 3 for both AAb+ and T1D donors). Data are expressed as means ± SEM. *P*-values were calculated by one-way ANOVA with Holm-Sidak multiple comparisons correction for C, and two-tailed unpaired Student *t*-test for D, E (Spearman) and G. *$P$ = 0,0219 YAP+/V+ vs. n-YAP+/V+; **$p$ = 0,00053 YAP+/V+ vs. YAP-/CVB+ for Aab+ group; § $p$ = 8.5 × 10⁻⁷ YAP+/V+ vs. n-YAP+/V+; §§ $p$ = 0,00011 YAP+/V+ vs. YAP-/CVB+ for T1D group. Scale bars depict 10 μm. Source data are provided as a Source Data file.

cells (SC-β) infected with CVB4[47], where the expression of canonical YAP target genes including *AMOTL2, Cyr61 and ANKRD1* was induced (Figure S6).

CVB-induced YAP upregulation was further validated by Western Blot analysis, which showed significant upregulation of total YAP protein at 24 h post-infection in human islet-exocrine pancreas co-cultures (Fig. 5D). Alongside total YAP, both active YAP (non-phosphorylated) and phosphorylated YAP (P-S127 YAP, representing cytoplasmic sequestration) increased following CVB infection. However, normalization to total YAP protein revealed no significant changes in the ratio of non-phosphorylated or phosphorylated to total YAP (Fig. 5E). This suggests that transcriptional upregulation of YAP and the resultant increase in total YAP protein is the primary mechanism by which CVB activates functional YAP, without altering YAP phosphorylation levels.

### YAP enhances coxsackievirus replication and potentiates coxsackievirus- induced islet inflammation and β-cell apoptosis

To investigate a link between YAP and CVB infection and its functional significance on β-cells, they were infected with CVB4 and CVB4 (MOI of 5 and 10 for INS-1E β-cells and human islets, respectively)[48,49], together with the adenoviral mediated transduction of a constitutively active form of YAP (YAP-S127A). YAP overexpression was sufficient to enhance viral replication seen by the substantially increased CVB3 and CVB4 genomic RNA, relative to the control LacZ transduced INS-1E cells (Fig. 6A) and human islets (Fig. 6B). The pro-viral effect of YAP was also confirmed by the increased level of the enterovirus-specific viral capsid protein VP1 upon YAP overexpression, compared to the control LacZ group in both INS-1E β-cells (Fig. 6C, D) and human islets (Fig. 6E, F). Immunofluorescence of VP1 and insulin verified the significant increase in the number of the VP1-positive β-cells by YAP overexpression in CVB-infected human islets, in comparison to control LacZ overexpression (Fig. 6G, H). This supports the hypothesis that YAP hyper-activation potentiates viral replication. Further, microscopy analysis of infected cells revealed the abundant YAP/VP1/insulin triple-positive cells in primary human islets suggesting the cell-autonomous action of YAP (Figure S7A).

Besides β-cells, pancreatic exocrine cells and ductal cells in particular are highly susceptible to CVB infections[50]. As the exocrine pancreas such as adult ductal cells naturally express YAP, we investigated whether endogenous YAP has a similar pro-viral effect. We used verteporfin (VP), a chemical inhibitor of the YAP/TEAD complex[51], which blocked downstream actions of YAP. In human islet (50% purity)/exocrine co-cultures infected with CVB3 or CVB4, CVB replication - analyzed by double staining for VP1 and insulin - was reduced in the presence of VP compared to control (Fig. 6I, and S7B). Staining for VP1 and the ductal marker CK19 showed VP1-CK19 co-positive cells in both CVB3 and CVB4 infected human ductal cells (Fig. 6J), and also here, the inhibition of YAP by VP led to the reduction in CVB3 and CVB4 replication as determined by the quantification of VP1/CK19 double-

positive cells (Fig. 6J, K). VP also significantly abolished CVB4 RNA genome replication in the infected human ductal cell line PANC1 (Figure S7C). The efficiency of VP to inhibit YAP signaling was verified by mRNA analysis of YAP's target gene CTGF, which was reduced by VP (Figure S7D). Consistently, VP treatment reduced the number of VP1-positive cells in CVB4-infected PANC-1 cells compared to untreated infected controls (Figure S7E, F). These findings support an essential role for YAP in promoting CVB replication across both β-cells as well as exocrine ductal cells.

As YAP potentiates CVBs replication in both primary and immortalized β-cells, we further investigated whether this higher virus replication also increases apoptosis. CVBs highly induce β-cell apoptosis[22]. YAP overexpression promoted a significant increase in CVB-mediated β-cell apoptosis as determined by caspase-3 cleavage, a universal marker of apoptosis, in INS-1E β-cells (Fig. 6L, M) as well as in human islets (Fig. 6N, O). TUNEL staining together with insulin confirmed loss in insulin in response to viral infection, as reported before[15,52–56], together with the increased level of β-cell apoptosis in human islets upon YAP overexpression compared to LacZ-overexpressed controls (Fig. 6P, Q).

As inflammatory/innate immunity responses mediate the pathophysiological mechanisms from enteroviral infection to T1D[57,58] and YAP was shown to be linked to inflammatory reactions[59], we next assessed the impact of YAP on islet inflammation during CVB infections. In line with previous reports[11,57,58], infection of human islets with CVB3 and CVB4 induced a strong type I interferon response, evidenced by the upregulation of IFN-β (*IFNB1*) mRNA and the consequent production of of interferon-stimulated genes (ISGs), including *CXCL10* and *OAS1* (Figure S7G–I). This response was accompanied by increased expression of pattern recognition receptors (PRRs) and enteroviral sensors, such as MDA-5 (*IFIH1*), RIG-I (*DDX58*), and *TLR3* (Figure S7J–L). Indeed, YAP overexpression further enhanced not only *IFNB1* mRNA expression but also CVB-induced expression of *CXCL10* and *OAS1* (Fig. 6R–T) as well as of *IFIH1, DDX58*, and *TLR3*, compared to LacZ-transduced control cells (Fig. 6U–W; the magnitude of the response varied between individual donors). In line with the gene expression data, overexpression of YAP potentiated the secretion of CXCL10 by infected human islets (Fig. 6X). All these data indicate that YAP-overexpressing islets presented higher levels of antiviral response components under CVB infections.

### YAP's pro-inflammatory effect depends on viral amplification

Our data indicate an increase in viral replication, accompanied by a CVB-induced inflammatory response and cell death, which subsequently leads to a higher rate of viral spread and a vicious cycle with viral progeny. This hypothesis was confirmed by using polyinosinic-polycytidylic acid poly(I:C), a replication-deficient synthetic analog of double stranded RNA which mimics viral infection. In contrast to CVB, YAP overexpression blocked poly(I:C)-induced β-cell apoptosis (Fig. 7A, B) compared to the LacZ-transduced control group in INS-1E

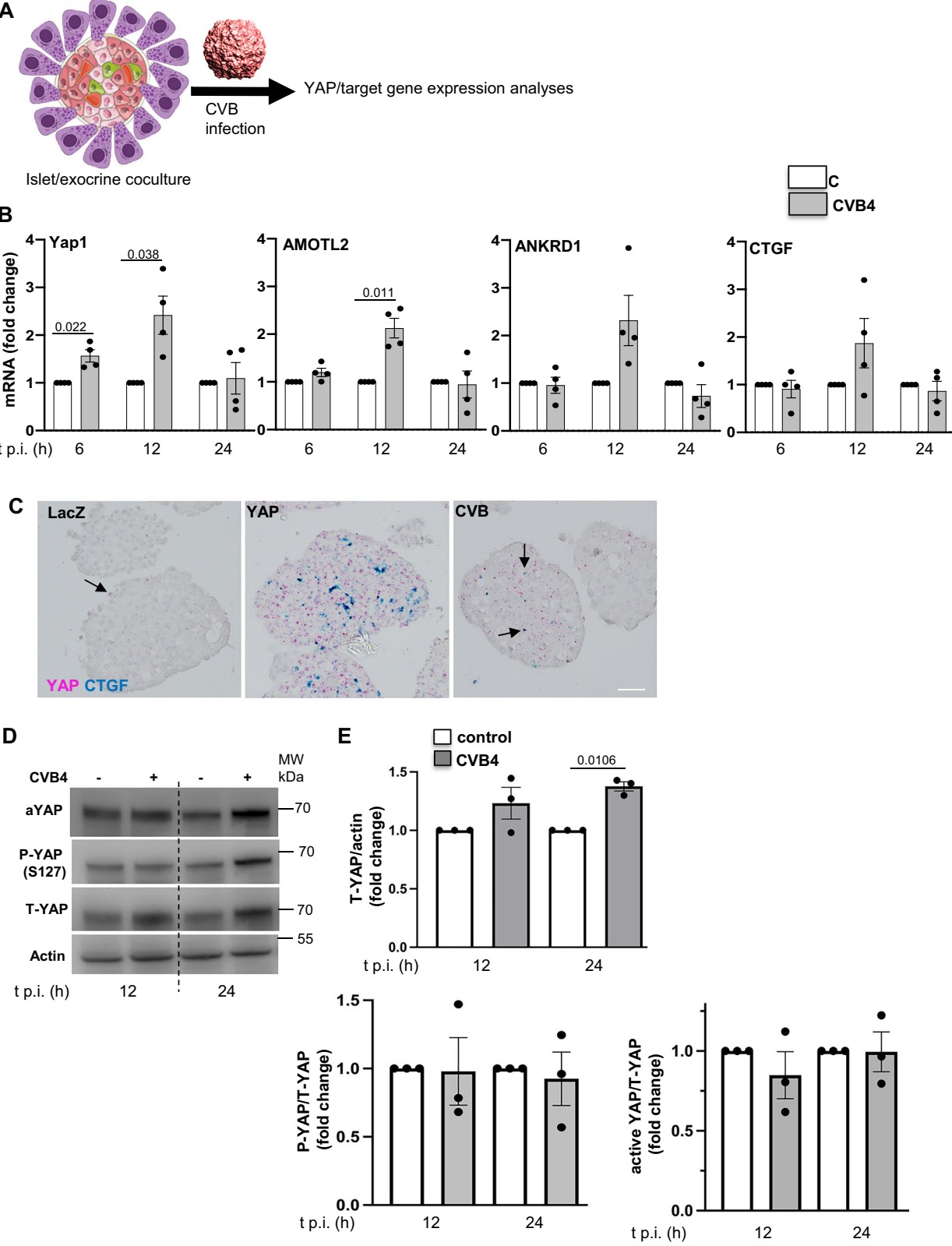

**Fig. 5 | Coxsackievirus infection induces YAP expression and activity in islet-exocrine co-cultures. A** Islet-exocrine co-cultures from the same donor were infected with CVB4 (MOI = 10) for 6–24 h. **B** qPCR analysis of *Yap1, AMOTL2, ANKRD1* and *CTGF* mRNA expression, normalized to actin (*n* = 4 organ donors). **C** Representative images of double RNAscope for *CTGF* (turquoise) and *Yap1* mRNA (pink) in human islets transduced with LacZ or YAP or infected with CVB4. **D** Representative Western blot image and (**E**) pooled quantitative densitometry

analysis of human islet-exocrine co-cultures, showing total, phosphorylated (S127), and active YAP (*n* = 3 organ donors). Data are expressed as means ± SEM. *P*-values were calculated by two-tailed paired Student *t*-test. Scale bar depicts 20 μm. **A** Image adapted from Servier Medical Art (https://smart.servier.com/), licensed under CC BY 4.0 (https://creativecommons.org/licenses/by/4.0/). Source data are provided as a Source Data file.

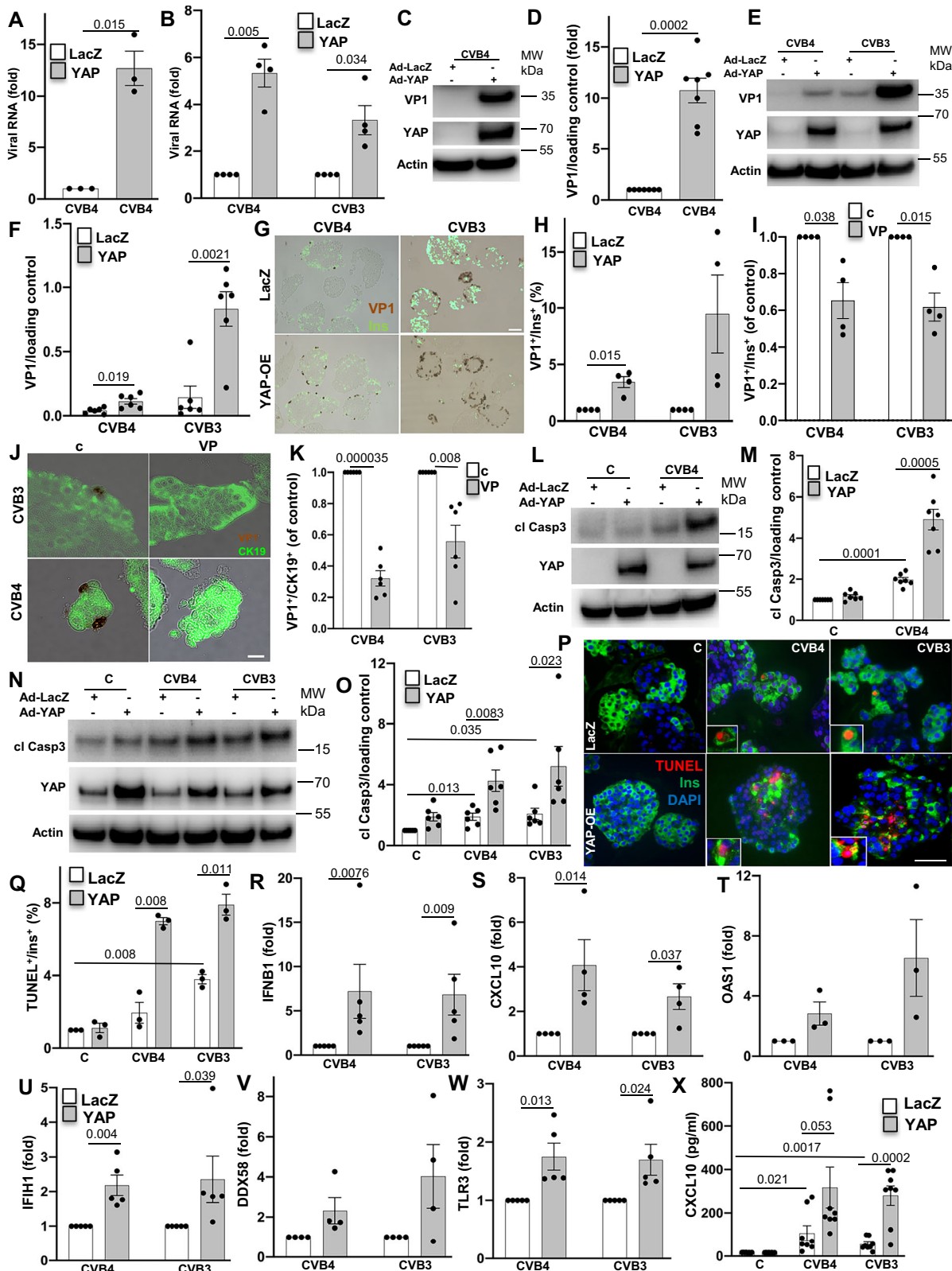

cells, suggesting that the pro-apoptotic function of YAP depends on CVB replication. Importantly, unlike in a CVB-infected environment, YAP overexpression in human islets transfected with poly(I:C) did not potentiate the inflammatory response (Fig. 7C). Additionally, YAP did not enhance poly(I:C)-induced CXCL10 production (Fig. 7D). These data further support the idea that YAP-induced inflammatory responses require CVB replication and amplification. Inflammatory

responses varied substantially between individual human islet donors, with poly(I:C) inducing a 350- to 1,200-fold increase in CXCL10 expression. Notably, YAP did not further amplify this response when each islet isolation was analyzed independently (Figure S8). To further demonstrate that viral amplification drives YAP-induced inflammation, we blocked viral replication using the viral capsid inhibitor pleconaril[60], an antiviral drug that has shown promising effects in a

**Fig. 6 | YAP enhances coxsackievirus replication and potentiates coxsackievirus-induced islet inflammation and β-cell apoptosis. A, C, D, L, M** INS-1E cells and (**B, E–H, N–X**) human islets transduced with Ad-YAP or Ad-LacZ control and then infected with CVB4 (MOI = 5) for 24 h (INS-1E) or CVB3 and -4 (MOI = 10) for 48 h (human islets). **A, B** Intracellular CVB3 or -4 RNA genome of (**A**) INS-1E cells (*n* = 3 independent experiments) and (**B**) human pancreatic islets (*n* = 4 organ donors). **C–F** Representative Western blots and pooled quantitative densitometry analysis of VP1 in (**C, D**) INS-1E cells (*n* = 7 independent experiments) and (**E, F**) human islets (*n* = 6 organ donors). **G, H** Representative images (**G**) and quantitative percentage of VP1-positive β-cells (**H**) are shown (*n* = 4 organ donors). **I–K** Human islets (50%) co-cultured with exocrine cells infected with CVB3 and −4 (MOI = 10) for 48 h and treated with or without 2.5 uM verteporfin (VP) for the last 24 h. **I** Quantitative percentage of %VP1/insulin⁺ cells (*n* = 4 organ donors). **J** Representative images and (**K**) quantitative percentage of %VP1/CK19⁺ cells (*n* = 6

independent positions from two organ donors). **L–O** Representative Western blots and pooled quantitative densitometry analysis of cleaved caspase 3 in (**L, M**) INS-1E cells (*n* = 7 independent experiments) and (**N, O**) human islets (*n* = 6 organ donors; endogenous YAP expression under control conditions stems from exocrine cells, which typically remain even in highly purified human islets cultures). **P, Q** Representative images (**P**) and quantitative percentage of TUNEL-positive β-cells (**Q**) are shown (*n* = 3 organ donors). **R–W** qPCR analysis for (**R**) *IFNB1*, (**S**) *CXCL10*, (**T**) *OAS1*, (**U**) *IFIH1*, (**V**) *DDX58*, and (**W**) *TLR3* mRNA expression in isolated human islets normalized to actin (**R, U, W**: *n* = 5, **S, V**: *n* = 4, **T**: *n* = 3 organ donors). **X** Secreted CXCL10 analyzed by ELISA in the culture media (*n* = 8 independent samples from five organ donors). Data are expressed as means ± SEM. *P*-values were calculated by two-tailed paired (**A, B, D, F, H, I, K, M, O, Q**) or unpaired (**X**) or ratio paired (**R–W**) Student *t*-test. Scale bars depict 50 μm (**G, P**) and 10 μm (**J**). Source data are provided as a Source Data file.

proof-of-concept clinical study in T1D[61]. In vitro, during CVB4 infection, pleconaril effectively blocked both basal and YAP-induced viral replication in human islets (Fig. 7E). It also fully prevented YAP-induced *IFNB1, CXCL10* and *OAS1* under CVB4 infection (Fig. 7F). Together, these data confirm that YAP's pro-apoptotic and pro-inflammatory effects are dependent on active viral replication.

## Chronic YAP re-expression induces diabetes by impairing insulin secretion and inducing β-cell dedifferentiation

To establish a pathological connection between diabetes and elevated YAP levels in β-cells of organ donors with T1D, we generated doxycycline (dox)-inducible β-cell specific homozygous (YAP⁺/⁺) Rip-Ins2-TetO-hYAP1-S127A mice ("β-YAP";) through crossing inducible active YAP overexpressing mice (TetO-YAP^Ser127A)[62] with mice carrying the tTA tetracycline transactivator under the control of the insulin promoter[63] (Fig. 8A). Dox administration in adult mice led to robust β-cell selective induction of YAP in isolated islets already after 2 days, confirmed by IHC and Western Blot (Fig. 8B, C). YAP was then transiently induced in β-cells over a two-week period (Fig. 8D), which resulted in elevated blood glucose levels and impaired insulin secretion during an intraperitoneal glucose tolerance test (i.p. GTT) in both male and female mice (Fig. 8E–J). Specifically, β-YAP mice exhibited delayed glucose clearance and reduced insulin levels throughout the GTT, indicating compromised glucose homeostasis. To assess whether this defect was β-cell autonomous, we isolated islets from β-YAP and control adult mice under the same treatment conditions and performed ex vivo glucose-stimulated insulin secretion (GSIS) assays. Notably, insulin secretion in response to glucose was significantly abolished in β-YAP islets compared to controls (Fig. 8K, L), confirming that YAP activation intrinsically impairs β-cell function.

We further investigated whether YAP overexpression caused a loss of β-cell identity or dedifferentiation and analyzed the expression of functional genes, including insulin (*Ins1, Ins2*), key β-cell transcription factors (*Pdx1, NeuroD1, MafA, Nkx2.2, Nkx6.1*, and *Glis3*), as well as critical genes involved in glucose sensing and metabolism (*GCK, Slc2a2, ABCC8, KCNJ11*). Expression levels of all genes were highly downregulated in β-YAP islets (Fig. 8M), suggesting loss of β-cell identity and functionality upon 2-weeks YAP re-expression in islets. We further investigated β-cell dedifferentiation by ALDH1A3, a universal marker of β-cell dedifferentiation[64], which showed a significant induction of ALDH1A3/insulin double-positive cells (Figure S9A, B), as well as upregulation of ALDH1A3 in isolated islets from β-YAP islets (Figure S9C), compared to controls. These findings collectively indicate β-cell dedifferentiation by homozygous YAP overexpression.

Consistent with our previous findings in human islets[34], YAP activation possessed a strong pro-proliferative capacity in both male and female mice, as determined by the quantification of double-positive insulin and Ki67 or pHH3 cells, compared to non-Dox-treated YAP-negative littermates (Fig. 8N–Q). The induction in β-cell replication was accompanied by a significant increase in the insulin-positive

area and β-cell mass in both male and female mice (Fig. 8R–U). Highly proliferating β-cells exhibits metabolic immaturity[65], as they simultaneously downregulate metabolic and functional genes, including those related to glucose metabolism, insulin expression and secretion, to allocate energy and cellular resources toward increasing replication. Thus, YAP-induced β-cell immaturity and dedifferentiation could be a consequence of YAP-induced proliferation. To investigate whether enhancing β-cell maturation could reverse impaired insulin secretion and the loss of β-cell identity upon YAP overexpression, we used H1152, a chemical inhibitor of ROCK, which has been shown to increase insulin secretion and β-cell maturation[66]. While YAP activation abolished glucose-induced insulin release, H1152 treatment significantly restored insulin secretion in YAP-overexpressing islets (Figure S9D, E). In line with this, H1152 exposure of YAP-overexpressing mouse islets elevated the gene expression of most β-cell identity and functionality markers, this was significant for a subset of genes (*Nkx6.1, NeuroD1, Slc2a2, ABCC8, KCNJ11, Glis3*; Figure S9F). These findings indicate that YAP-induced impaired insulin secretion and compromised cellular identity are reversible and could be restored by enhancing β-cell maturation. Taken together, long-term overexpression of YAP in β-cells is associated with deleterious metabolic consequences.

## YAP expression correlates with increased cell proliferation in CVB4-infected cells

YAP is a strong promoter of cell proliferation, as demonstrated in our previous work with isolated human islets[34] and in this study with YAP-overexpressing mouse β-cells. To directly link YAP expression with proliferation under CVB infection, we reanalyzed a publicly available scRNA-seq dataset from CVB4-infected human pancreatic cells[67]. This dataset includes 41,125 cells from seven human pancreatic islet/exocrine batches, visualized using UMAP and categorized into CVB-infected and mock-treated groups (Figure S10A–C). To identify cell types expressing *Yap1*, we examined its expression in control mock treated (uninfected) cells across cell types with at least 20 cells per type. Pancreatic ductal cells, stellate cells, and acinar cells exhibited the highest *Yap1* expression, consistent with patterns observed in pancreatic tissues from the Gaulton study (Figure S10D). *Yap1* expression in key YAP-expressing cells (ductal, activated stellate, and acinar cells) was elevated in CVB4-infected cells compared to mock-treated counterparts (Figure S10E). Next, CVB4-treated cells were categorized into infected and non-infected groups based on CVB4 polyprotein gene expression. Cells with non-zero polyprotein expression were labelled infected, while those with zero expression were considered non-infected. To evaluate the proliferative status of infected cells in relation to YAP expression, we focused on ductal and stellate cells, two major YAP-expressing pancreatic cell types with relative proliferative activity. Infected cells were stratified into "YAP1-high" and "YAP1-low" groups based on the median *Yap1* expression per cell type. GSEA on YAP1-high vs. YAP1-low infected cells confirmed enrichment of YAP target genes within the Hippo signaling pathway

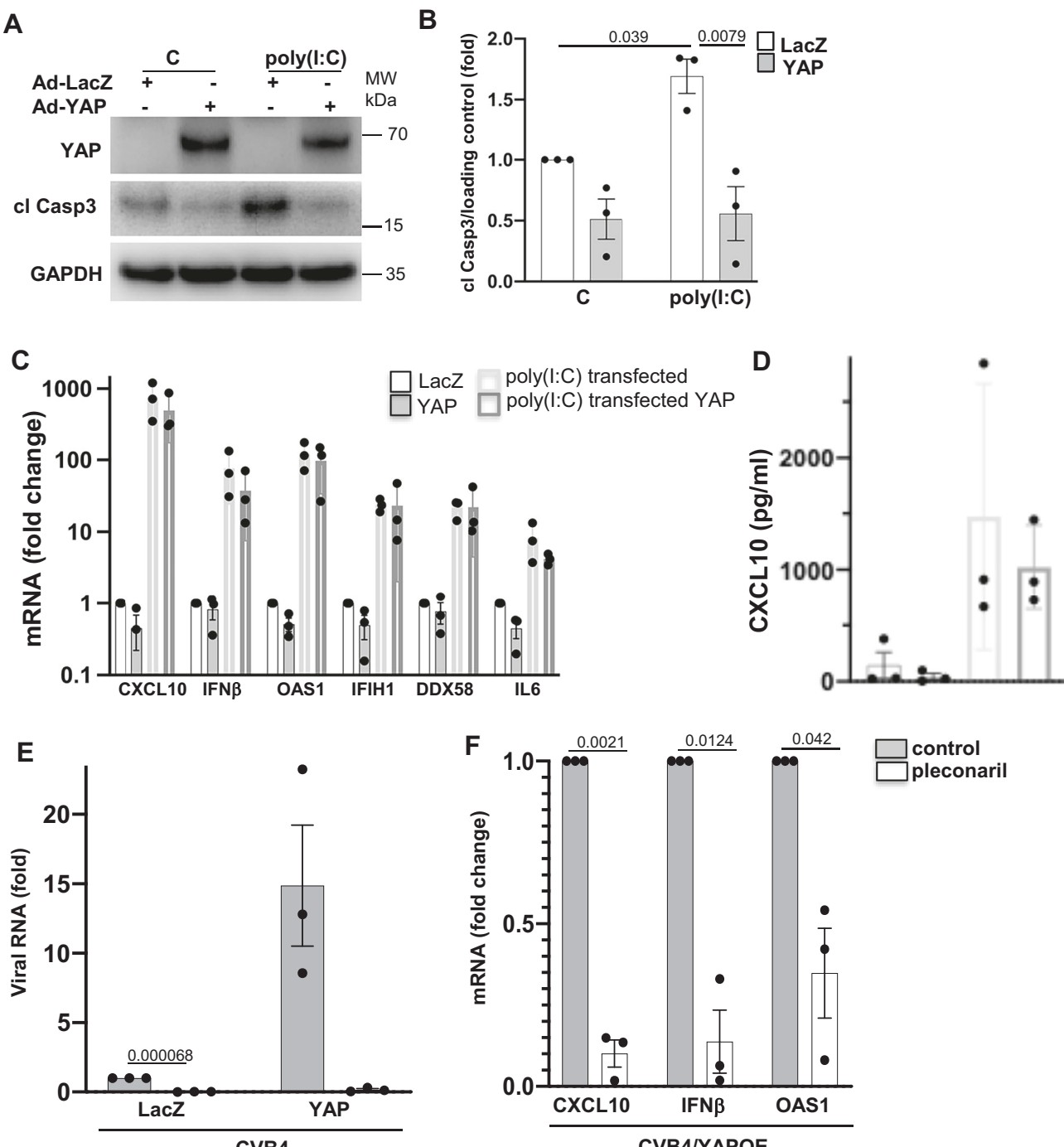

**Fig. 7 | YAP's pro-inflammatory effect depends on viral amplification.** INS-1E cells (**A**, **B**) or human islets (**C**, **D**) were transduced with Ad-YAP or Ad-LacZ control and then treated (INS-1E) or transfected (human islets) with Poly(I:C). **A** Representative Western blot and (**B**) pooled quantitative densitometry analysis of cleaved caspase 3 in INS-1E cells (*n* = 3 independent experiments). **C** qPCR analysis for *CXCL10, IFNB1, OAS1, IFIH1, DDX58* and *IL6* expression in isolated human islets, normalized to actin (*n* = 3 organ donors). **D** secreted CXCL10 analyzed by ELISA in the culture media (*n* = 3 organ donors). **E**, **F** Human islets were transduced with Ad-YAP or Ad-LacZ control for 24 h and then infected with CVB4 (MOI = 10) for 48 h with or without treatment with 10 mM pleconaril (*n* = 3 organ donors). **E** Intracellular CVB4 RNA genome. **F** qPCR analysis for *CXCL10, IFNB1,* and *OAS1* expression in isolated human islets, normalized to actin. Data are expressed as means ± SEM. *P*-values were calculated by two-tailed paired Student *t*-test. Source data are provided as a Source Data file.

(Figure S10F). Additionally, GSEA revealed significant upregulation of proliferation-related pathways, including positive regulation of cell population proliferation, DNA replication, mitotic cell cycle, and mRNA transcription, in the CVB4-infected YAP1-high subpopulation compared to YAP1-low cells (Figure S10F). These findings highlight a strong correlation between YAP expression and proliferation in CVB4-infected pancreatic cells, supporting the concept that CVB replication is more efficient in actively dividing cells enriched for high YAP expression.

## A YAP-TEAD-MST1 feedback loop controls CVB replication and cell death

Dynamic and precise control of YAP activity by the upstream Hippo components is important to ensure proper cell stress response under

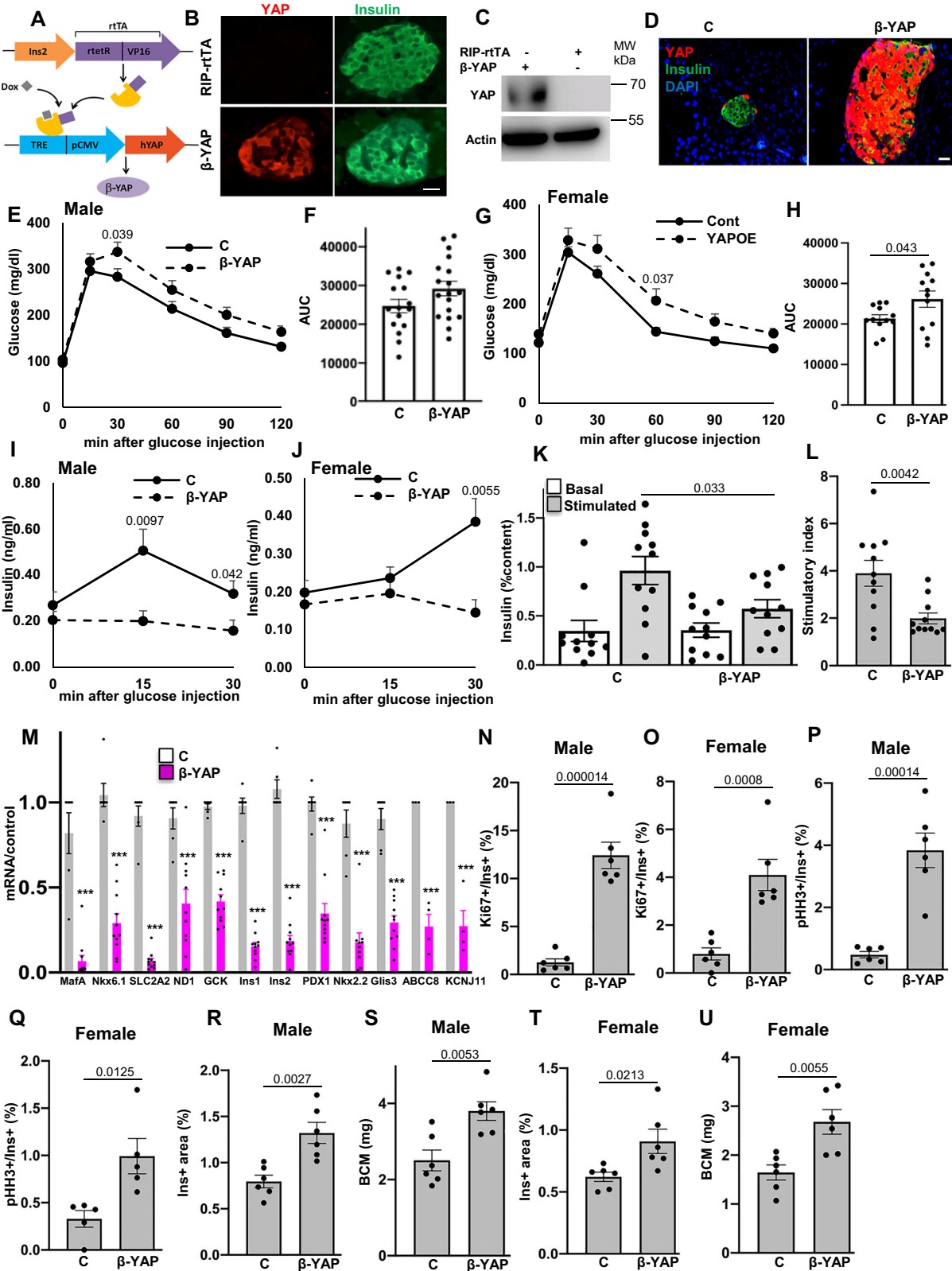

physiological condition or upon invasion of pathogen. In the course of analyzing the Hippo pathway, we have surprisingly noticed an increase in total MST1 protein level in the YAP-overexpressing INS-1E cells (Fig. 9A, B) and human islets (Fig. 9C, D), suggesting a so far undiscovered Hippo feedback loop, in which YAP in its function as transcriptional co-regulator induces *STK4* (gene encoding MST1) transcription. Indeed, the amount of *STK4* mRNA was substantially

increased in INS-1E cells overexpressing active YAP compared to control cells (Fig. 9E). We then examined whether this feedback mechanism operates in vivo using β-YAP-OE transgenic mice. In line with data in cultured cells, a significant increase of *STK4* expression was evident in islets isolated from β-YAP-OE mice (Figure S11A) further supporting a role for YAP in *STK4* transcriptional regulation. Importantly, CVB4 infection itself triggered the induction of MST1 in both INS-1E cells and

**Fig. 8 | YAP re-expression induces diabetes by impairing insulin secretion and inducing β-cell dedifferentiation. A** Scheme how β-YAP mice were generated by crossing RIP-rtTA with TetO-YAP$^{Ser127A}$ mice. **B** IHC and **C** Western Blot confirmation of YAP induction in pancreatic islets after 2 days i.p injection of Dox. **D** YAP was transiently induced by doxycycline (DOX) administration in drinking water for 2 weeks (β-YAP) and results compared to -DOX/-YAP (control; **C**). **E–H** intraperitoneal glucose tolerance test (ipGTT) and respective AUC analyses in β-YAP and control male (**E, F;** $n = 16$ C, $n = 18$ β-YAP) and female (**G, H,** $n = 12$/group) mice. **I, J** Insulin levels during an ipGTT measured before (0 min) and 15/30 min after glucose injection in β-YAP and control male (**I;** $n = 11$/group) and female (**J;** $n = 9$ C, $n = 11$ β-YAP) mice. **K, L** Islets were isolated from β-YAP and control mice, cultured overnight and subjected to an in vitro GSIS. **K** Insulin secretion during 1 h-incubation with 2.8 mM (basal) and 16.7 mM glucose (stimulated), normalized to insulin content and (**L**) stimulatory index denotes the ratio of stimulated to basal insulin secretion ($n = 11$). **M** RT-PCR for *MafA, Nkx6.1, Slc2a2, NeuroD1, GCK, Ins1, Ins2, Pdx1, Glis3* ($n = 6$ C, $n = 11$ β-YAP mice), *Nkx2.2* ($n = 6$ C, $n = 10$ β-YAP mice), *Abcc8, and Kcnj11* ($n = 3$ C, $n = 4$ β-YAP mice). Microscopical analyses of β-proliferation by Ki67 (**N, O**) and pHH3 (**P, Q**) in both (**N, P**) male and (**O, Q**) female mice expressed as percentage of Ki67- ($n = 6$ mice/group) or pHH3- ($n = 6$ male and $n = 5$ female mice/group) positive β-cells. Insulin-positive area (**R, T**) and β-cell mass (**S, V**) in both (**R, S**) male and (**T, U**) female mice ($n = 6$ mice/group). Data are expressed as means ± SEM. *P*-values were calculated by two-tailed unpaired Student *t*-test for all except by a mixed-effects model with Holm-Sidak multiple comparisons correction for (**E, G**). ***$p < 0.001$ compared to control; Scale bars depict 20 μm. Source data are provided as a Source Data file.

human islets (Figure S11B–E) suggesting that there may be a YAP-mediated feedback mechanism that occurs during CVB infection. We performed additional experiments to determine whether CVB4 infection and YAP overexpression synergistically increase MST1 levels. While each independently had a consistent effect - both CVB4 infection and YAP overexpression induced MST1 - no synergistic increase in MST1 was observed (Figure S11F).

As YAP mostly acts through TEAD transcription factors (TEAD1-4) to regulate gene expression, we sought to mechanistically uncover the transcriptional regulatory activity of YAP/TEAD on MST1 (Figure S11G). The YAP/TEAD inhibitor VP reduced the transcriptional upregulation of *STK4* induced by YAP, compared to untreated INS-1E cells (Fig. 9F). Consistently, VP fully reversed the induction of MST1 protein expression in YAP-overexpressing cells in both INS-1E cells (Fig. 9G, H) and human islets (Fig. 9I, J) in a dose-dependent manner. VP also triggered degradation of exogenous YAP as mechanism to block YAP downstream signaling (Fig. 9G, I). The loss-of-function YAP mutant carrying the S94A substitution abolishes its interaction with TEADs, rendering it transcriptionally inactive[41], and thus serves as a useful tool to further dissect the molecular basis of YAP/TEAD-mediated MST1 induction. Unlike the active form of YAP, overexpression of the YAP-S94A mutant failed to induce MST1 expression at both the mRNA and protein levels compared to GFP-transfected INS-1E cells, demonstrating that YAP stimulates MST1 in a TEAD-dependent manner (Figure S11H–J). Also, a genetically encoded fluorescently-tagged competitive inhibitor that blocks binding between YAP and TEAD ("TEAD inhibitor (TEADi)"[68], attenuated *STK4* mRNA and MST1 protein levels in YAP-overexpressing cells (Figures S11K–M). Altogether, we conclude that a YAP/TEAD mediated transcriptional induction of *STK4* and consequently elevated MST1 protein abundance constitute a negative feedback loop.

We then examined whether *STK4* is a direct transcriptional target of the YAP/TEAD complex. Two putative TEAD1-binding motifs were identified in the rat *STK4* promoter region by using a transcription factor-binding site prediction platform, the Eukaryotic Promoter Database (ED)[69] (Figure S11N). To experimentally confirm this, we used a luciferase reporter assay to examine whether the transcriptional rate of the *STK4* promoter could be stimulated by YAP. The *STK4* promoter region including a 1.5 kb sequence proximal to the transcription start site was cloned into an pEZX-PG04.1 reporter vector and transfected into HeLa cells. We then generated a HeLa cell line stably expressing conditional Gaussia Luciferase (GLuc) reporter located downstream of the *STK4* promoter and constitutively secreted Alkaline Phosphatase (SEAP) which was used as internal control for normalization. Dual reporter analysis showed that YAP overexpression significantly increased luciferase activity- as indicated by the ratio of secreted Gluc and SEAP-, compared to LacZ control, and this response was abolished by VP (Fig. 9K). Chromatin immunoprecipitation (ChIP) coupled with qPCR (using two pairs of primers to amplify *STK4* promoter region) in INS-1E cells transduced with YAP or corresponding LacZ control was conducted to check whether the YAP/TEAD transcriptional complex directly interacts with the promoter region of *STK4* gene. ChIP data

using anti-YAP antibody and specific primers for the *STK4* promoter showed that YAP specifically binds to the *STK4* proximal promoter- as represented by fold enrichment in YAP occupancy- in INS-1E cells overexpressing YAP but not in the LacZ-overexpressing cells, which was again blocked by VP (Fig. 9L, M). Positive control primers to amplify *ANKRD1*, a well-established direct target gene of the YAP/TEAD complex[70], and a negative control IgG verified ChIP specificity (Fig. 9L–N). All these complementary methods indicate that the YAP/TEAD complex occupies the *STK4* promoter and exerts *STK4* expression induction in β-cells, confirming the postulated negative feedback loop. While previous studies have identified established feedback mechanisms within the Hippo pathway, such as YAP-LATS[71] or YAP-miR-YAP[72], our identification of a YAP-MST1 negative feedback loop provides further insight. To determine whether this feedback loop is specific to β-cells or represents a more universal regulatory mechanism, we performed additional experiments. Our data reveal a distinct specificity for the upregulation of MST1 induced by YAP within β-cells. This effect was not observed in other cell types tested, including HeLa (cervical) and HEK293 (kidney) cells, in which YAP overexpression did not alter STK4 mRNA expression (Figure S12A) or total MST1 protein levels (Figure S12B).

To test the functional relevance of this YAP-MST1 loop during CVB infection, we performed MST1 knockdown experiments. siRNA-mediated depletion of endogenous MST1 enhanced VP1 production, whereas at the same time attenuated apoptosis in CVB4-infected YAP-transduced cells, compared to control siScr transfected counterparts (Fig. 10A, B). Consistently, immunofluorescence and qPCR analyses revealed that MST1 silencing in INS-1E cells resulted in significantly higher CVB4 replication as represented by increased VP1-positive infected cells in the siMST1-YAP-CVB4 group compared to the corresponding siScr-YAP-CVB4 control (Fig. 10C, D) as well as by increased intracellular CVB4 RNA genome (Fig. 10E). To further confirm the anti-viral action of MST1, we used the dominant-negative form of MST1. Amino acid substitution mutation of the critical lysine within the ATP binding site (K59 for MST1) with alanine compromises MST1 kinase activity, thus MST1 is inhibited[73]. Infection of INS-1E cells transfected with MST1-K59 led to a marked enhancement of intracellular VP1 accumulation compared to the GFP-overexpressing cells, while inhibition of MST1 markedly attenuated the level of cleaved caspase-3 in YAP-overexpressing cells upon CVB4 infection (Fig. 10F, G). Also, microscopy analysis of VP1-positive cells showed that MST1-K59 introduction stimulated an increase in CVB4 replication (Fig. 10H, I). Similar to the immunofluorescent staining, genetic MST1 antagonism largely induced the viral copies of CVB4 RNA compared to the GFP-transfected control group (Fig. 10J) further indicating that MST1 blocks CVB4 replication.

These results highlight a dual role for MST1 during viral infection. Inhibition of this pro-apoptotic kinase clearly reduced virus-induced cell death. Conversely, MST1 also regulates viral replication through YAP. Given that MST1 is an upstream inhibitor of YAP in the classical Hippo cascade, and activated YAP induced the expression of MST1, YAP-mediated MST1 upregulation might at the end serve as a negative

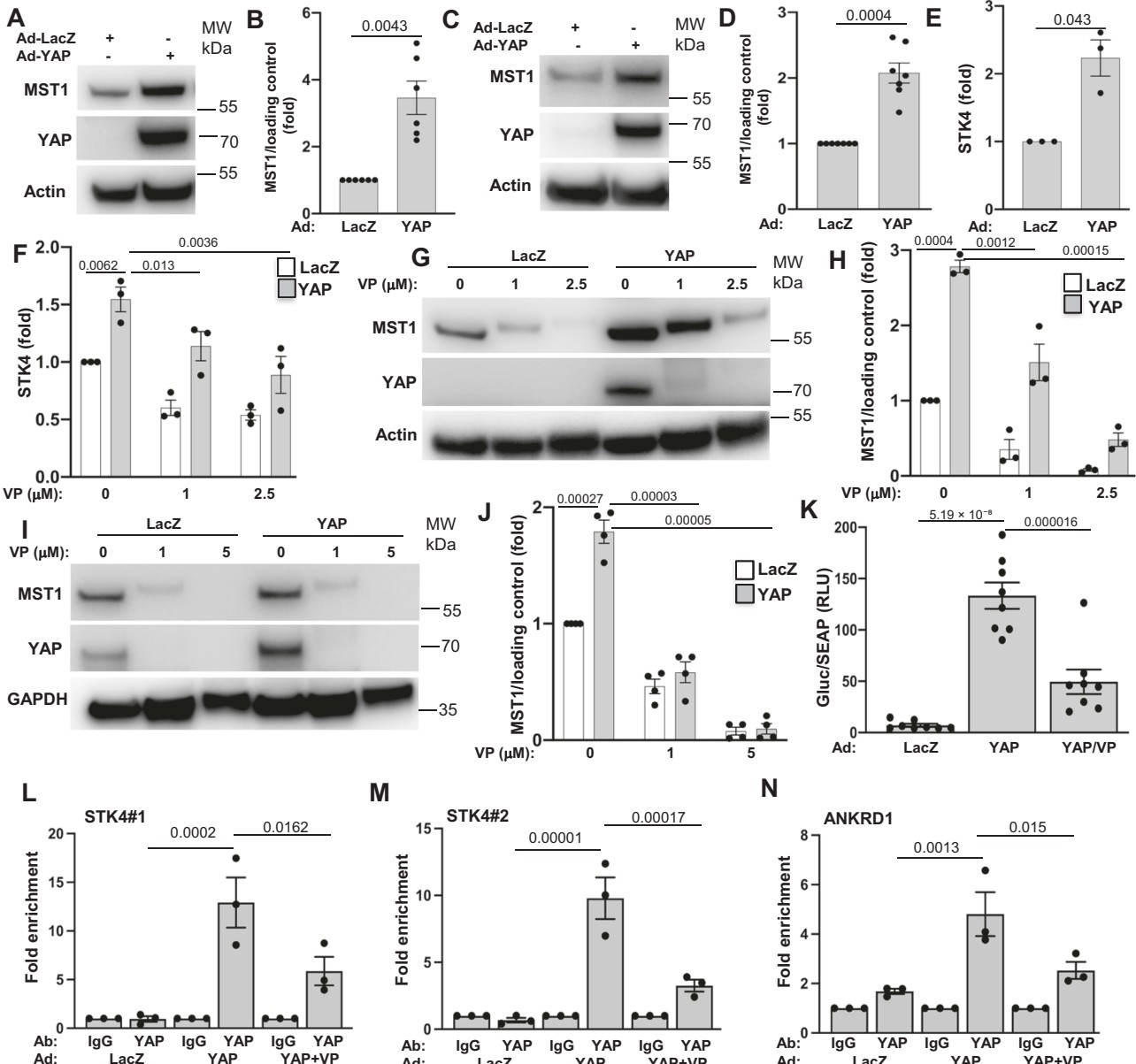

**Fig. 9 | A YAP-TEAD axis regulates the level of MST1. A, B, E** INS-1E cells and (**C, D**) human islets transduced with Ad-YAP or Ad-LacZ control for 48 h. **A–D** Representative Western blot and pooled quantitative densitometry analysis of MST1 in (**A, B**) INS-1E cells ($n = 6$ independent experiments) and (**C, D**) human islets ($n = 7$ organ donors). **E** qPCR for *STK4* mRNA expression in INS-1E cells normalized to actin ($n = 3$ independent experiments). **F, G, H** INS-1E cells and (**I, J**) human islets transduced with Ad-YAP or Ad-LacZ control for 48 h treated with or without 1–5 μM verteporfin (VP) for last 6 h (INS-1E) or 24 h (human islets). **F** qPCR for STK4 mRNA expression in INS-1E cells normalized to actin ($n = 3$ independent experiments). **G–J** western blots and pooled quantitative densitometry analysis of MST1 in (**G, H**) INS-1E cells ($n = 3$ independent experiments) and (**I, J**) human islets ($n = 4$ donors). **K–N** Hela or INS-1E cells transduced with Ad-YAP or Ad-LacZ control for 48 h treated with or without 1 μM verteporfin (VP) for the last 24 h. **K** Hela cells culture media was analyzed for activities of both GLuc and SEAP and data presented as the relative change in normalized GLuc to SEAP ($n = 8$ independent experiments). **L–N** ChIP from INS-1E cells was performed with control IgG, or YAP antibody as indicated ($n = 3$ independent experiments). The presence of (**L, M** *STK4* and (**N**) *ANKRD1* promoters was detected by PCR. Data presented as fold enrichment in which ChIP signals are divided by the IgG-antibody signals, representing the fold increase in signal relative to the background signal. Data are expressed as means ± SEM. *P*-values were calculated by one-way (**K–N**) and two-way (**F, H, J**) ANOVA with Holm-Sidak multiple comparisons correction, and two-tailed paired Student *t*-test for B,D,E. Source data are provided as a Source Data file.

feedback loop to limit excessive YAP hyper-activation and subsequent CVB replication and amplification; thus, the YAP-MST1 feedback mechanism plays an important role in regulating the viral replication machinery.

## Discussion

There is abundant support for an association of enterovirus infections as a trigger for progression to T1D[16]. Only little is known about the complex enteroviral-host interactions which ultimately may determine the outcome of viral infections in the pancreas. Dysregulated interactions may trigger islet autoimmunity and T1D. In this study, we show that YAP, a principal transcriptional effector of the Hippo pathway, was highly upregulated in the exocrine pancreas of AAb[+] and T1D organ donors. This suggests that pathological disturbance in T1D starts in the whole pancreas. Few YAP-positive cells were also observed within islets of AAb[+] and T1D donors, where they have never been seen in controls

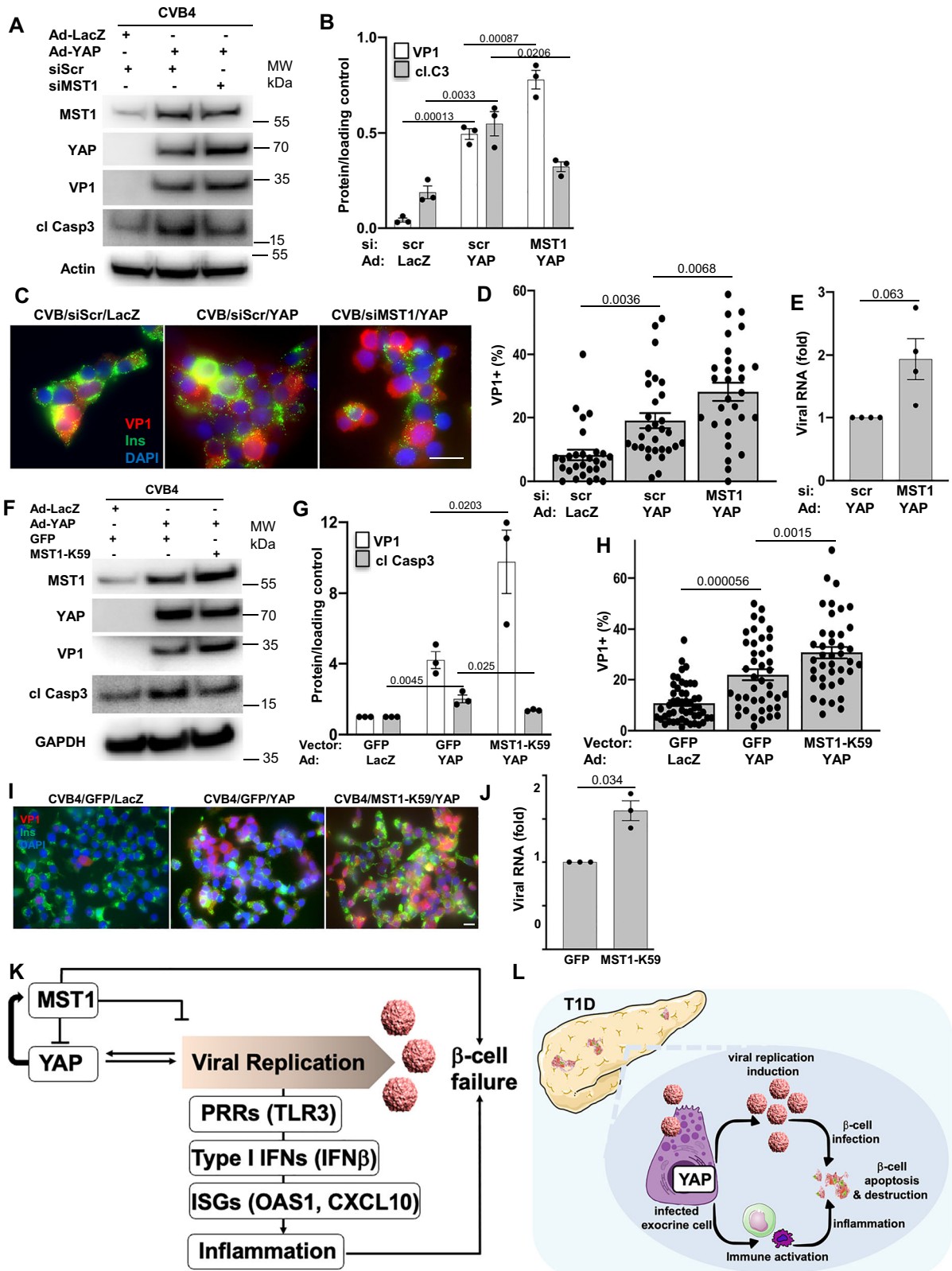

and are physiologically disallowed. Even a small sub-cluster analysis of single and multiple AAb⁺ donors - where multiple AAb⁺ individuals have a markedly higher risk of developing T1D[74] -revealed that intra-islet YAP expression correlated with early β-cell area loss and increased islet immune cell infiltration. Such YAP expression was associated with enteroviral infections; the majority of CVB-infected pancreatic cells were either colocalized with YAP or located in close proximity to YAP-

positive cells in AAb⁺ and T1D pancreases. Cell-culture models of β-cells, human islets as well as human exocrine pancreatic cells showed that YAP hyperactivation directly fostered CVB replication, potentiated β-cell apoptosis and enhanced the expression of genes involved in innate immunity and antiviral defence. Conversely, pharmacological targeting of YAP blocked CVBs replication in YAP-expressing primary and immortalized pancreatic exocrine cells. Experiments involving

**Fig. 10 | A YAP-TEAD-MST1 feedback loop controls CVB replication and cell death. A–E** INS-1E cells transfected with siMST1 or control siScr and then transduced with Ad-YAP or Ad-LacZ control for 48 h. All cells were infected with CVB4 (MOI = 5) for last 24 h. **A, B** Representative Western blot and pooled quantitative densitometry analysis of MST1, VP1 and cleaved caspase 3 in INS-1E cells (*n* = 3 independent experiments). **C, D** Representative images (**C**) and quantitative percentage of VP1-positive cells (**D**) are shown (*n* = 28 independent positions for C-LacZ; *n* = 30 for YAP; *n* = 29 for siMST1-YAP). **E** Intracellular CVB4 RNA genome of INS-1E cells (*n* = 4 independent experiments). **F–J** INS-1E cells transfected with MST1-K59 or control GFP constructs and then transduced with Ad-YAP or Ad-LacZ control for 48 h. All cells were infected with CVB4 (MOI = 5) for last 24 h. **F, G** Representative Western blot (**F**) and pooled quantitative densitometry analysis (**G**) of MST1, VP1 and cleaved caspase 3 in INS-1E cells (*n* = 3 independent experiments). **H, I** Quantitative percentage of VP1-positive cells (**H**) and representative images (**I**; *n* = 50 independent positions for C-LacZ; *n* = 42 for YAP; *n* = 40 for MST1-K59-YAP). **J** Intracellular CVB4 RNA genome of INS-1E cells (*n* = 3 independent experiments). Data are expressed as means ± SEM. *P*-values were calculated by one-way ANOVA with Holm-Sidak multiple comparisons correction for (**B, D, G, H**) and by two-tailed paired Student *t*-test for (**E, J**). Scale bars depict 10 μm. Source data are provided as a Source Data file. **K, L** Our model how a vicious cycle of YAP expression and CVB replication in the human pancreas may lead to T1D. **K** At the molecular level, YAP induces the expression of its own negative regulator MST1, through a feedback mechanism, thereby limiting YAP-driven viral replication and promoting apoptosis of infected cells. YAP is highly elevated in the pancreas of patients with T1D where it boosts enteroviral replication, induces a strong IFN response, and promotes islet inflammation, ultimately leading to β-cell apoptosis and destruction. **L** Persistently infected exocrine cells, where YAP promotes viral replication, may drive T1D by serving as viral reservoirs that facilitate islet infection and by triggering local inflammation that attracts immune cells and damages β-cells. We extend our gratitude to Richard E. Lloyd for kindly providing the enterovirus image. **K, L** Image adapted from Servier Medical Art (https://smart.servier.com/), licensed under CC BY 4.0 (https://creativecommons.org/licenses/by/4.0/). Source data are provided as a Source Data file.

transgenic mice with inducible β-cell-specific overexpression of YAP clearly demonstrated the metabolic consequences associated with long-term selective overexpression of YAP in pancreatic β-cells. These data directly link pathological YAP upregulation observed in the pancreas and islets of patients with T1D to β-cell failure and metabolic deregulation.

Having identified a pathological link between YAP and enteroviruses, we next asked whether YAP promotes enteroviral replication or whether the infection itself induces *Yap1* transcription. Our findings demonstrate that CVB infection leads to transcriptional upregulation of *Yap1* in human pancreatic islets and exocrine tissue, representing the dominant regulatory mechanism. This suggests a potential vicious cycle in which CVB infection enhances YAP expression, which in turn promotes inflammation and the accelerated β-cell destruction, characteristic of T1D. We propose several upstream pathways through which CVB may induce *Yap1* transcription: (1) ER stress and UPR activation: CVB-induced ER stress and UPR activation via PERK is likely to increase *Yap1* transcription, as PERK knockdown has been shown to reduce ER stress-induced *Yap1* mRNA increases[75], and enteroviruses are known to activate the PERK pathway[76]. (2) Transcription factor modulation: e.g., the GABPβ transcription factor complex, known to regulate *Yap1* promoter activity[77] and implicated in viral infections such as HIV-1[78], may also be activated during CVB infection to promote *Yap1* transcription. (3) Disruption of microRNA-mediated repression: *Yap1* mRNA turnover is tightly regulated by microRNAs. CVB infection has been shown to alter host microRNA expression in human islets, including the downregulation of miR-149-5p[79], whose inhibition has been reported to relieve repression of *Yap1*, resulting in increased *Yap1* mRNA expression[80]. Post-transcriptional mechanisms may also contribute. For instance, CVB-induced disruption of cell polarity and tight junctions[81] may inhibit the Hippo pathway (MST1/2, LATS1/2), reducing YAP phosphorylation and promoting its stabilization and nuclear translocation. Together, these mechanisms provide a biologically plausible framework for how CVB infection regulates YAP expression at multiple levels. Further mechanistic studies are needed to fully elucidate these possibilities.

Our detailed mechanistic work identified MST1 as a direct YAP/TEAD target forming a cell-intrinsic feedback loop. This YAP-MST1 bidirectional interaction may act as "molecular brake" to restrict excessive YAP-driven viral replication and amplification, to promote discarding of infected host cells and to finally put the viral replication machinery on hold (Fig. 10K). Thus, we identified YAP as a pro-viral, and MST1 as an anti-viral factor. This seems neither specific to pancreatic exocrine and endocrine cells nor to CVBs. YAP also promotes viral replication and production during SARS-CoV-2 or influenza infections[82,83], while MST1 inhibits SARS-CoV-2 replication[82]. Accordingly, MST1 genetic deficiency enhances the susceptibility to pathogen

infections as well as presents autoimmune symptoms (e.g., hypergammaglobulinemia and autoantibody production)[84–87]. Given high levels of remaining enteroviral RNA deposits in YAP-overexpressing cells in the pancreas in AAb+ and T1D, it is possible that this YAP-MST1 feedback is no longer fully functional. Such hypothesis requires further research.

An imbalance between immune activation and immune protection is a key pathological element of autoimmune diseases such as T1D. Previous investigations highlight the important regulatory function of YAP in inflammatory signaling. While highly complex and context- and cell type-dependent, its dysregulation is connected to inflammatory-related disorders such as atherosclerosis, non-alcoholic steatohepatitis (NASH), inflammatory bowel disease, pancreatitis and pancreatic cancer[59]. YAP has both, pro- as well as anti-inflammatory actions[88]. For example, YAP balances inflammation and supports tissue regeneration and repair, as *Yap* mRNA therapy improves cardiac function through anti-inflammatory mechanism in ischemia-reperfusion injury[89], or it blocks antiviral signaling to balance the host response which is vital for cellular survival during infection[35,36]. On the contrary, YAP can also be pro-inflammatory i.e., YAP drives hepatic inflammation in NASH[42], where YAP directly enhances the expression of pro-inflammatory cytokines through the formation of YAP/TEAD complex in the promoter region of inflammatory cytokines[42]. Likewise, YAP antagonism blocks the secretion of pro-inflammatory cytokines by neoplastic cells[38], and YAP genetic loss in pancreatic neoplastic epithelial cells results in a decrease in the number of CD45+ immune cells in the pancreas, together with the progression of pancreatic ductal adenocarcinoma (PDAC)[38]. As we show, YAP was a positive regulator of islet inflammation during CVB infection with an exaggerated interferon response that could initiate autoimmunity and loss of β-cells in T1D. In the CVB infection model of human islets and co-culture with exocrine cells, YAP's pro-inflammatory effect was dependent on viral replication. This was confirmed by the lack of potentiation by YAP in the replication-deficient poly(I:C) model and the inhibition of YAP-induced inflammation under CVB infection by the viral capsid inhibitor pleconaril. Pleconaril efficiently halts CVB replication in human islets during persistent CVB infection[60], and, in this study, also prevented the subsequent YAP-induced interferon response. Its use has recently been linked with preservation of residual insulin production in new-onset T1D, as demonstrated in a proof-of-concept clinical study[61]. Both poly(I:C) and pleconaril-treated CVB serve as critical controls, modeling the presence of viral RNA in the absence of productive replication. In these settings, YAP retains its canonical functions, and does not promote inflammation or β-cell death. These results underscore the requirement for active viral replication to convert YAP's role from protective to pathogenic. Nonetheless, we recognize that CVB-specific factors - such as viral proteins or replication intermediates absent in

poly(I:C) - may also contribute to the full inflammatory phenotype observed during infection in the presence of YAP. Further mechanistic dissection of these interactions is needed in the future.

The innate antiviral immunity i.e., the IFN response, is a key event in the course of autoimmunity and β-cell destruction. Type I IFN in islets triggers HLA-I[90], and HLA-I hyperexpression is a hallmark of pancreas pathology in T1D[91]. The transcriptional signature of IFN responses precedes islet autoimmunity[92], and several polymorphisms within the interferon signature are genetic risk factors for T1D[93,94]. In fact, incubation of islets with type I and III IFNs or boosting IFN response limits viral replication and associated cell injury in pancreatic islets[95,96]. If the activation of the IFN response is excessively prolonged or intense, it can also trigger autoimmune reactions in the islets and cause damage to β-cells. Interestingly, YAP has been implicated in innate immunity and was previously shown to negatively regulate the type I IFN response through blockade of antiviral signaling proteins TBK1 and/or IRF3[35,36]. Our finding here, that YAP upregulated the interferon response during CVB infection in the pancreas is somewhat paradoxical, given YAP's potent inhibitory action on the antiviral response. One explanation for this paradox could be that the higher innate immune/antiviral response observed in YAP-overexpressing cells is primarily derived from an insufficient eradication of the virus (possibly through existing genetic polymorphisms in the interferon signature). Another possibility is that MST1, YAP's target gene identified in our study, enhances the antiviral response by (1) classical inactivation of YAP which would relieve the TBK1/IRF3 suppression, (2) direct activation of IRF3 as reported before in a different context[97], or (3) degradation of IRAK1, a negative regulator of type 1 IFN signaling[98]. In any case, boosted antiviral response is unable to protect YAP-overexpressing cells against cell death caused by massive viral replication indicating that the classical intrinsic regulatory function of YAP/MST1 in antiviral signaling is overridden by the YAP-driven CVB amplification. In support of this argument, YAP did not potentiate β-cell apoptosis or inflammation in the presence of pleconaril or replication-deficient viral mimic poly(I:C). This finding confirms that cell death, lysis and inflammation during CVB infection result from high levels of viral replication. Consistent with these results, UV-inactivated CVB which is incapable of replication, does not induce β-cell death[22], and the production of proinflammatory cytokines and chemokines is dependent on active viral replication[57].

Aberrant upregulation of YAP- marked by robust cytoplasmic and nuclear localization of YAP in ductal and centro-acinar cells- is not limited to the pancreas in T1D; other pancreatic disorders, including PDAC and pancreatitis present elevated expression of YAP[99–101]. YAP and its well-known target gene CTGF are robustly increased in pancreatitis[32,101–103], an inflammatory disease of the exocrine pancreas manifested by extensive loss of the normal exocrine parenchyma, fibrosis and inflammation, and both exocrine and endocrine functional failure. Commonly upregulated YAP in T1D as well as in PDAC and pancreatitis suggests that the Hippo/YAP pathway may play a general and central role in the pathogenesis of pancreatic disorders. Supported by using genetically engineered mouse models, pancreas-specific deletion of MST1/2 or LATS1/2, which is functionally equivalent to YAP activation, recapitulate T1D, PDAC or pancreatitis in terms of robust immune cell infiltration, widespread inflammation, fibrosis, reduced pancreas mass, exocrine dysfunction and disrupted islet architecture[31,32,102]. Importantly, genetic loss of YAP or CTGF neutralization is sufficient to rescue the phenotype[32,102] indicating that YAP is a key driver of such pancreatic structural and functional abnormalities. YAP is also induced by STZ-induced diabetes in the Kras model of pancreatic cancer in mice, with normalization of glycemia correlating with reduced pancreatic YAP expression[104]. This is in line with our observations of CVB3/4-induced potentiation of hyperglycemia and the corresponding increase in YAP expression within islets, ductal and acinar cells in NOD mouse model.

Notably, various environmental and metabolic factors, e.g., viral infections, inflammation, obesity, or diabetes have the potential to induce PDAC or pancreatitis[105,106]. Also, a significant number of patients diagnosed with PDAC or pancreatitis have impaired glucose tolerance or diabetes[107,108]. Although these pancreatic disorders differ mechanistically and phenotypically in many ways, YAP may function as a major hub of transcriptional convergence in the crosstalk between pancreatic cells and immune cells in response to microenvironmental cues such as infections or cellular transformation upon injury. YAP signaling could therefore be an important therapeutic target for pancreatic comorbidity disorders.

While the classical perspective regards T1D as a β-cell specific disease, recent findings indicate that T1D is a disorder that involves the entire pancreas in which the loss of functional β-cell mass is most evident[109,110], together with the decreased pancreas mass[109–113], immune cell infiltration and inflammation of the exocrine pancreas[1,114], and exocrine dysfunction/insufficiency[115,116]. An abnormal exocrine-endocrine cell interplay has been linked to the development of MODY8, a monogenic form of diabetes inherited in a dominant manner, in which a mutant gene expressed selectively in acinar cells induces impaired β-cell function and loss[117]. In a recent study, we have systematically shown the predominant presence of enteroviral RNA in the exocrine pancreas in patients with T1D[9]. This suggests that enteroviruses do not primarily target islet cells but the whole pancreas providing a pathological connection between T1D-related changes in the exocrine pancreas and the development of disease. Enteroviral infections in the exocrine pancreas can induce fulminant T1D marked by extensive inflammation with inflamed (CXCL10-positive) and/or infected (VP1-positive) ductal and acinar cells surrounded by immune cells such as T-cells indicating the existence of non-neglectable immune responses to enteroviral infection and subsequent cell injury in the exocrine pancreas[118]. In line with this, previous studies reported that, in addition to islets[119], CXCL10 expression is induced in the exocrine tissue in T1D[120] and gene expression analyses show the robust antiviral signature mainly in the exocrine pancreas in T1D[121].

A dysregulated crosstalk between the exocrine and endocrine pancreas may have a more important role in the development of T1D than previously believed. Persistently infected exocrine cells in the pancreas, where viral replication is promoted by YAP, could be a trigger for a chronic immune cell attack and the subsequent development of T1D in two ways: firstly, the persistently infected exocrine cells may act as "cellular reservoirs" that enhance viral replication in the pancreas, leading to higher viral loads and more efficient spread of the virus to the islet cells; and secondly, local inflammation triggered by the infected exocrine cells may directly harm β-cells and attract immune cells to infiltrate the islets, ultimately leading to the destruction of β-cells (Fig. 10L). Additionally, viral infections are known to dysregulate host gene expression[122,123]. Our data show increased YAP expression and downstream targets in human exocrine cells post CVB4 infection. Our experimental cells were from non-diabetic individuals and subjected to an acute infection; the observed upregulation of YAP subsided over time. However, under persistent or unfavourable conditions, enteroviral infections may cause sustained or gradual increases in YAP expression in chronically infected tissues, resulting in prolonged hyperinflammation and the development of autoimmunity. YAP upregulation could either be a viral strategy to enhance replication by hijacking host machinery or part of the host's anti-inflammatory response, which paradoxically promotes hyperinflammation during persistent infections. In support of the former, prior studies have shown that pro-survival and proliferative signaling pathways, such as those regulated by Myc, can create a more permissive environment for viral replication[124–127]. Similarly, our reanalysis of single-cell RNA-seq data from CVB-infected pancreatic cells revealed a strong correlation between YAP expression and proliferative signatures, including enrichment of Myc targets in high-YAP expressing

cells. These findings suggest that CVB may exploit YAP-driven pro-liferative programs to enhance replication, thereby amplifying inflammation. Therefore, antiviral therapies for early T1D may effectively halt virus-induced cytokine storms and prevent progression to autoimmunity. Although anti-viral treatments and anti-enteroviral vaccines have shown some efficacy in T1D, they would need to be started in genetically susceptible individuals either before or at the time of initial infection to effectively prevent disease progression.

The complex exocrine-islet-immune interactions require further mechanistic investigations, with major emphasis on immune cell responses and paracrine factors, in analogy with other pancreatic diseases. They will be key for targeted interventions for T1D.

## Methods

### Ethical regulations
Research in this paper complies with all relevant ethical regulations; experiments involving human islets and pancreatic sections from organ donors have been granted by the Ethics Committee of the University of Bremen. Experiments involving mice were approved by the Bremen Senate (Die Senatorin für Gesundheit, Frauen und Verbraucherschutz). Organ donors are not identifiable and anonymous.

### Human islets and nPOD pancreas collection
All human islet experiments were performed in the islet biology laboratory, University of Bremen. Human islets were distributed by the two JDRF and NIH-supported approved coordination programs in Europe (Islet for Basic Research program; European Consortium for Islet Transplantation ECIT) and in the US (Integrated Islet Distribution Program IIDP).

Formalin-fixed paraffin-embedded (FFPE) pancreatic tissue sections were obtained from well-characterized organ donors from the nPOD[37] throughout a large collaborative initiative. Access to these tissues requires an application process, as the material is restricted. Donor IDs were selected in a fully random and unbiased manner for all analyses, except for matching age, BMI and gender across the three groups: control ($n = 14$), Aab$^+$ ($n = 16$) and T1D ($n = 15$; Table S1). Ideally, we aimed to analyze >10 donors for each part of the study. However, due to limited availability, we primarily performed 3–5 donors or independent rounds of staining, ensuring equal donor representation across all groups. The analyses were performed in a blinded manner and independently conducted immediately after each round of staining. Further demographic donor data are available upon reasonable request.

### Islet isolation, cell culture and treatment
Human islets were isolated from pancreases of nondiabetic organ donors (both male and female) at University of Lille, Strasbourg and ProdoLabs and cultured on Biocoat Collagen I coated dishes (#356400, Corning, ME, USA). The clonal rat β-cell line INS-1E was kindly provided by Claes Wollheim (Geneva & Lund University). The immortalized cell lines HeLa and HEK293 were purchased from American Type Culture Collection (ATCC, Manassas, VA, USA). The human pancreatic exocrine ductal cell line PANC-1 was generously provided by Cenap Güngör, Universitätsklinikum Hamburg-Eppendorf (UKE, Germany). PANC-1 cells were cultured in complete DMEM (Invitrogen, CA, USA) medium at 25 mM glucose. Human islets were cultured in complete CMRL-1066 (Invitrogen) medium at 5.5 mM glucose. Hela and INS-1E cells were cultured in complete RPMI-1640 (Sigma Aldrich, Missouri, MO, USA) medium at 11.1 mM glucose. All media included with L-glutamate, 1% penicillin-streptomycin and 10% fetal bovine serum (FBS). INS-1E medium was supplemented with 10 mM HEPES, 1 mM sodium pyruvate and 50 μM β-mercaptoethanol. In some experiments, human islets, INS-1E cells and PANC-1 cells were additionally cultured with 1–5 μM YAP/TEAD inhibitor verteporfin (#SML0534, Sigma Aldrich, USA) for 6 h–24 h or treated or transfected with 2–10 μg poly(I:C) for 24 h (#P9582; Sigma Aldrich). HeLa cells were cultured with 2 μg puromycin-dihydrochlorid (P9620, Sigma, USA) for positive clonal selection.

### YAP-transgenic, NOD mice and mouse islet isolation
β-cell-specific YAP overexpressing (YAP-OE) mice were generated by crossing inducible active YAP overexpressing mice (TetO-YAPSer127A, provided to our lab in collaboration with Fernando Camargo, Boston Children's Hospital, Boston, MA, background C57Bl/6)[62] with mice carrying the tetracycline transactivator (tTA) under the control of the insulin promoter (RIP-rtTA mice, background C57Bl/6, kindly provided by Al Powers, Vanderbilt University Medical Center, Nashville, TN, USA)[128]. In the Rip-Ins2-TetO-hYAP1-S127A mice, rtTA gene becomes activated specifically in the islet β-cells due to the Ins2 promoter. Upon doxycycline (a tetracycline analog) treatment, the rtTA protein in these cells can bind to the tet-response element (TRE) and subsequently causing the transcription of the constitutively active form of YAP gene which is under a CMV promoter element. This system enables a fine-tuned spatio-temporal control over the expression of the aYAP gene in the pancreatic β-cells. All the experiments were done on 8-10 weeks old female and male mice and genotype of the mice is in homozygous condition. Pancreatic islets were isolated after 2 weeks doxycycline induction through drinking water in the mice. Islets from β-cell specific YAP-OE and respective control mice were isolated by pancreas perfusion with a Liberase TM (#05401119001, Roche, Mannheim, Germany) solution[119] according to the manufacturer's instructions and digested at 37 °C, followed by washing and handpicking.

Normoglycemic age- and sex-matched non-obese diabetic (NOD; NOD/ShiLtJ, Strain #:001976, RRID:IMSR_JAX:001976, background Jcl:ICR) mice at 11-12 weeks old were injected intraperitoneally with either 400 plaque-forming units (pfu) CVB3 or CVB4 or DMEM vehicle and pancreata dissected 1-week post-infection.

Mice were euthanized by gradual-fill carbon dioxide inhalation. All mice used in this experiment were housed in a temperature-controlled room with a 12 h light-dark cycle and were allowed free access to food and water in agreement with NIH animal care guidelines, §8 German animal protection law, German animal welfare legislation and with the guidelines of the Society of Laboratory Animals (GV-SOLAS) and the Federation of Laboratory Animal Science Associations (FELASA), or with protocols approved by UBC Animal Care Committee (ACC) (NOD mice).

### Glucose tolerance test and insulin secretion
For intraperitoneal glucose tolerance tests (ipGTT), mice were fasted overnight for 12 h and injected intraperitoneally with glucose (B.Braun, Germany) at a dose of 1 g/kg body weight. Blood samples were collected at time points 0, 15, 30, 60, 90, and 120 min for glucose measurements using a Glucometer (FreeStyle; Abbott, IL, USA). Blood samples for insulin secretion were collected before (0 minutes) and after (15 and 30 min) intraperitoneal injection of glucose (2 g/kg body weight) and measured using an ultrasensitive mouse ELISA kit (ALPCO Diagnostics, NH, USA).

### Viruses and virus purification and titration
Enteroviruses CVB3 (Nancy) and CVB4 (JVB) were kindly provided by Andreas Dotzauer (University of Bremen, Germany). Fetal Rhesus Kidney-4 (FRhk-4) cell line was used for the preparation and isolation of virus stocks. FRhk-4 cells were infected with CVB3 or CVB4 viruses for 2 h and were cultured for 2–3 days until visualization of the cytopathic effect. The supernatant from these cells was harvested after 3 rounds of freezing and thawing followed by centrifugation for 10 min at 720 x g to precipitate cell debris. Virus purification was carried out by the sucrose gradient method using an ultracentrifuge. First supernatant was centrifuged at 4500 x g for 10 min. Further, it was centrifuged for 12 h at 120,000 x g in 40% sucrose gradient buffer (40%

sucrose, 10 mM Tris pH 7.5 100 mM NaCl and 1 mM EDTA). The invisible pellet was resuspended in 1x PBS. Aliquoted viral stocks were stored at −80° C. The TCID50 (tissue culture infectious dose 50%) was determined using serial dilutions. Briefly, FRhK-4 cells were seeded in duplicates in 96-well plates. They were infected for 2 h in serum-free media with serial dilutions of viral stocks. The cytopathic effect was determined under a light microscope and the TCID50 was calculated accordingly to Spearman-Kärber.

### Virus infection of human islets or cell line

INS-1E or PANC-1 cells were infected with CVB4 virus at MOI (multiplicity of infection) of 5 based on the CVB-permissive cell line FKRH4. The cytopathic effect determined under light microscope and TCID$_{50}$ calculated according to Spearman-Kaerber. Virus stocks were diluted in serum free medium and cells were inoculated with 750 μl at 37 °C and 5% CO$_2$. Control cells were incubated only with 750 μl of serum-free medium. After 2 h infection, cells were washed three times with 1xPBS and media was replaced by 10% FCS supplemented media for 24 h. Infection of Human islets was performed with CVB3 or CVB4 viruses at MOI 10 under the same condition. For human islets 6–48 h post-infection endpoint was chosen and then cells were harvested for staining as well as protein or RNA analysis. The culture supernatants were collected for measuring secreted CXCL10.

### Adenovirus transduction

The adenoviruses control Ad-CMV-b-Gal/LacZ (#1080) and Ad-CMV-h-YAP1-S127 (custom production) were purchased from VECTOR BIOLABS, PA, USA. Isolated human islets or INS-1E cells were infected with Ad-LacZ or Ad-YAP at a multiplicity of infection (MOI) of 100 (for human islets) or 10 (for INS-1E) for 4 h in CMRL-1066 or RPMI-1640 medium without FBS respectively. After 4 h incubation, adenoviruses were washed off with PBS and replaced by fresh complete medium which contains 10% FBS. Human islets or INS-1E cells were collected for staining, as well as RNA and protein isolation after 48–72 h transduction.

### Plasmids and siRNAs

To knock down MST1, SMARTpool technology was used (Dharmacon, CO, USA). A mix of ON-TARGETplus siRNAs directed against the following sequences: rat MST1 (#L-093629-02) sequences CUCCGAAACAAGACGUUAA; CGGCAGAAAUACCGCUCCA; CGAGAUAUCAAGGCGGGAA; GGAUGGAGACUACGAGUUU. An ON-TARGETplus nontargeting siRNA pool (Scramble; siScr) served as controls.

Following plasmids have been used: Kinase-dead (MST1-K59; dnMST1) was kindly provided by Dr. Junichi Sadoshima and Dr. Yasuhiro Maejima (UMDNJ, New Jersey Medical School). pCMV-Flag-YAP-S94A was a gift from Kunliang Guan (Addgene plasmid # 33102; http://n2t.net/addgene:33102; RRID: Addgene_33102)[41]. pCEFL EGFP-TEADi was a gift from Ramiro Iglesias-Bartolome (Addgene plasmid # 140144; http://n2t.net/addgene:140144; RRID: Addgene_140144)[68]. pCMV-flag S127A YAP was a gift from Kunliang Guan (Addgene plasmid # 27370; http://n2t.net/addgene:27370; RRID: Addgene_27370)[129]. GFP plasmid was used as a control.

### Transfection

GFP, EGFP-TEADi, MST1-K59, YAP-S94A, and pCMV-flag S127A YAP plasmids were used to overexpress these proteins in INS-1E cells. 100 nM MST1 or scr siRNAs were used for the transfection in INS-1E cells. To achieve silencing and overexpression, jetPRIME® transfection reagent (#114-75; Polyplus transfection, France) was used to deliver desired siRNA, DNA or poly(I:C) into INS-1E cells according to manufacturer's instructions. In brief, jetPRIME buffer was mixed with siRNA/DNA and vortexed for 10 s, then jetPRIME® transfection reagent was added and vortexed for 1 s. The mixture was stand at room temperature (RT) for 10 minutes after quick spin. The jetPRIME-siRNA or DNA complexes were then added to complete RPMI-1640 to transfect INS-1E cells. Transfection efficiency was estimated by fluorescent microscopy of GFP.

### Glucose-stimulated insulin secretion (GSIS)

Insulin secretion in response to glucose stimulation (GSIS) was assessed in isolated mouse islets. The islets were initially pre-incubated in Krebs-Ringer bicarbonate buffer (KRB) with 2.8 mM glucose for 30 min, followed by exposure to fresh KRB containing 2.8 mM glucose for 1 hour (basal) and an additional 1 h in KRB with 16.7 mM glucose (stimulated). After washing with 1xPBS, the islets were lysed using RIPA buffer to measure total insulin content, and insulin levels were quantified using human and mouse insulin ELISA kits (ALPCO Diagnostics, NH, USA). The secreted insulin values were normalized to the insulin content.

### Western Blot analysis

Human or mouse islets, INS-1E, HeLa or HEK293 cells were washed three times with ice-cold PBS after medium removal and lysed with RIPA lysis buffer (50 mM Tris HCl pH 8, 150 mM NaCl, 1% NP-40, 0.5% sodium deoxycholate, 0.1% SDS) supplemented with Protease and Phosphatase Inhibitors (Thermo Fisher Scientific (TFS), MA, USA). Samples went under multiple freeze-thaw cycles and finally incubated on ice for 30 min with intermittent vortexing. The cell lysates were centrifuged at 16,000 x g for 20 min at 4 °C and the clear supernatant containing the extracted proteins were kept at −80 °C for storage. Protein concentrations were measured by the BCA protein assay (TFS). Equivalent amounts of protein from each condition were run on a NuPAGE 4–12% Bis-Tris gel (Invitrogen; CA, USA) and electrically transferred into PVDF membranes. Membranes were blocked at RT using mixture of 2.5% milk (Cell Signaling Technology/CST, MA, USA) and 2.5% BSA (SERVA Electrophoresis GmbH, Heidelberg, Germany) for 1 h and incubated overnight at 4 °C with rabbit anti-cleaved caspase-3 (#9664), rabbit anti-Total YAP (#14074, clone D8H1X, CST), rabbit anti-active YAP (#ab223126, clone EPR19812, abcam), rabbit anti-Phospho-YAP (Ser127) (#4911), rabbit anti-MST1 (#3682), rabbit anti-ALDH1A3 (#NBP2-15339), rabbit anti-GAPDH (#2118), rabbit anti-β-actin (#4967; all CST), and mouse anti-Enterovirus/VP1 (clone 5-D8/1 #M7064, Dako). All primary antibodies were used at 1:1,000 dilution in 1xTris-buffered saline plus Tween-20 (1xTBS-T) containing 5% BSA and 0.5% NaN$_3$. Later, membranes were incubated with horseradish-peroxidase-linked anti-rabbit or anti-mouse secondary antibodies (Jackson ImmunoResearch, PA, USA) and developed using Immobilon Western chemiluminescence assay system (Millipore, MA, USA). Analysis of the immunoblots was performed using Vision Works LS Image Acquisition and Analysis software Version 6.8 (UVP BioImaging Systems, CA, USA).

### Measurement of CXCL10 release

CXCL10 secretion into culture media from controls and virus infected isolated human islets was measured by Human CXCL10/IP-10 DuoSet ELISA kit (#DY266-05, R&D Systems, MN, USA) according to the manufacturer's instructions.

### qPCR analysis

Total RNA was isolated from cultured isolated islets or INS-1E/PANC-1 cells using TriFast (PEQLAB Biotechnologie, Germany). 500–1000 ng of RNA were reverse transcribed to cDNA (RevertAid reverse transcriptase, Thermo Fisher Scientific (TFS), MA, USA). Quantitative RT-PCR was carried out as previously described in ref. 27 using Biosystems StepOne Real-Time PCR system (Applied Biosystems, CA, USA) with TaqMan assays or SybrGreen (Applied Biosystems). TaqMan® Gene Expression Assays were used for Stk4 (#Hs00178979), CTGF (#Hs01026927-g1), CXCL10 (#Hs00171042), IFNB1 (#Hs02621180),

OAS1 (#Hs00973637), DDX58 (#Hs01061436), TLR3 (#Hs01551078), IFIH1 (#Hs00223420), IL6 (#Hs99999032), YAP1 (#Hs00371735), AMOTL2 (Hs#01048101), ANKRD1 (Hs#00173317), Tuba1a (#Hs00362387), ACTB (#Hs99999903), Stk4 (#Mm00451755), Tuba1a (#Mm00846967), Stk4 (#Rn01750112), ACTB (#Rn00667869), Nkx2.2 (#Mm00839794_m1), Nkx6.1 (#Mm00454961_m1), NeuroD1 (#Mm01946604_s1), MafA (#Mm00845206_s1), Slc2A2 (#Mm00446229_m1), Abcc8 (#Mm00803450_m1), Glis3 (#Mm00615386_m1), Gck (#Mm00439129_m1C41), KcnJ11 (#Mm00440050_s1), Ins1 (#Mm04207513_g1), Ins2 (#Mm00731595_g1), PDX1 (#Mm00435565_m1), and), and ACT (#Mm00607939_s1). EV-RNA was detected by using a SybrGreen primer pair (forward: 5′- CGGCCCCTGAATGCGGCTAA-3′; reverse: 5′-GAAACACGGACACCCAAAGTA-3′). The relative changes in gene expression were analyzed by ΔΔCT method.

## Chromatin immunoprecipitation (ChIP) assay

$4 \times 10^6$ INS-1E cells were dual-cross-linked consecutively with 2 mM disuccinimidyl glutarate (DSG, #20593, TFS) for 45 min and 1% formaldehyde for 10 min. ChIP was performed according to the user's instructions for SimpleChIP Enzymatic Chromatin IP Kit (#9003, CST). In brief, chromatin DNA was digested with micrococcal nuclease (MNase). Immunoprecipitation reactions were carried out with chromatin extracts using IgG negative control or YAP antibodies (both CST) overnight at 4 °C. Proteinase K was added for de-crosslinking, and samples were incubated for 4 h in a water bath at 65 °C. Precipitated DNA was quantitated by real-time PCR analysis. The SybrGreen primers used in this study to amplify the promoter regions were: STK4#1 fw 5′ CCTCGACTTCCTCATGGCTG 3′, rev 5′ ACTAGGGACCCAATGAGCCT 3′; STK4#2 fw 5′ GCCAGCCTGTTTCTTCCTCT 3′, rev 5′ CTCCACGACTGGTGAGGTTT 3′; ANKRD1 fw 5′ GTGTGATGCACAATGCTTGC 3′, rev 5′CTTATCGGGAAGCCAGGGAC 3′. ANRD1, a YAP target gene, was used as a positive control. All ChIP signals were expressed as a fold enrichment (as a ratio of the YAP signal to the IgG signal for each respective condition).

## Dual reporter assay

HeLa cells were seeded into 6-well plates and transiently transfected with pEZX-PG04.1 reporter construct (#RPRM55953-PG04, Genecopoiea, MD, USA) using jetPRIME® transfection reagent. 48 h post-transfection, stable HeLa cells expressing conditional Gaussia Luciferase (GLuc) reporter located downstream of rat STK4 promoter and constitutive Secreted Alkaline Phosphatase (SEAP) was generated by puromycin selections. After selection, Hela cells were maintained in culture medium containing 2 µg/ml puromycin. HeLa stable cells were then transduced with Ad-LacZ or Ad-YAP treated with or without VP. After 48 h, medium was analyzed for activities of both GLuc and SEAP using the Secrete-PairTM Dual Luminescence and Gaussia Luciferase Assay Kit (Genecopoiea) per manufacturer's instructions. The data are presented as the relative change in normalized GLuc activities to SEAP.

## Immunofluorescence

Paraffin-embedded bouin-fixed human islets or human primary pancreatic cells were deparaffinized and rehydrated. INS-1E or PANC-1 cells were fixed with 4% PFA for 30 min followed by 4 min permeabilization with 0.5% Triton-X-100. Fixed or embedded cells were then blocked with blocking buffer containing 3% BSA and then incubated overnight at 4 °C with the following antibodies (single or double): guinea pig anti-insulin (#IR002, FLEX polyclonal DAKO), mouse, mouse anti-Enterovirus/VP1 (clone 5-D8/1 #M7064, Dako), mouse anti-chromogranin (#ab715, Abcam), mouse anti-cytokeratin 19/CK-19 (#15463, Abcam), rabbit anti-ALDH1A3 (#NBP2-15339), rabbit anti-Total YAP (#14074, clone D8H1X, CST), and rabbit anti-active YAP (#ab223126, clone EPR19812, abcam). The next day sections were incubated with Cy3-conjugated donkey anti-mouse (#715-165-150), FITC-conjugated

donkey anti-guinea pig (706-096-148) or FITC- conjugated donkey anti-mouse (#715-095-150) or anti-rabbit secondary antibodies (all from Jackson Immuno Research Laboratories, West Grove, PA; 1:100 dilution) for 1 h at RT or 37 °C. β-cell apoptosis in fixed human islet sections were performed by the terminal deoxynucleotidyl transferase-mediated dUTP nick-end labeling (TUNEL) technique according to the manufacturer's instructions (In Situ Cell Death Detection Kit, TMR red; Roche) and double stained for insulin. Slides were mounted with Vectashield with 4′6-diamidino-2-phenylindole (DAPI, #H-1200-10, Vector Labs).

## YAP immunohistochemistry

Detection of YAP protein in pancreatic tissue was carried out by classical immunohistochemistry (IHC) coupled with SuperBoost™ tyramide signal amplification (#B40931, Biotin XX Tyramide SuperBoost™ Kit, Streptavidin, TFS). After tissues deparaffinization and dehydration, endogenous peroxidase was quenched by 3% hydrogen peroxidase for 1 h at RT. Tissues were blocked by applying the blocking buffer for 1 h at RT and subsequently were incubated with rabbit anti-YAP (D8H1X, #14074, CST) antibody alone or in combination with mouse anti-chromogranin (#ab715, Abcam) antibody overnight. A day after, sections were washed with PBS and were incubated with rabbit poly-HRP-conjugated secondary antibody for 1 h at RT. To amplify the signal, a Tyramide working solution was prepared according to the manufacturer's instructions by adding the Tyramide solution and hydrogen peroxide into the reaction buffer. Sections were incubated for 10 min at RT followed by applying reaction stop regent for 3 min. The chromogenic detection was completed by applying ABC (Avidin/Biotin) system (VECTASTAIN® ABC-HRP Kit, Peroxidase-Standard, #PK-4000) for 1 h and DAB substrate (3,3′-diaminobenzidine-DAB Substrate Kit, Peroxidase-HRP, #SK-4100, all Vector Laboratories) for 5 min; both at RT. For the YAP-chromogranin double labeling, staining continued by using fluorescein isothiocyanate (FITC)-conjugated secondary donkey anti-mouse antibody (#715-095-150, Jackson Immuno Research Laboratories, West Grove, PA) for 1 h at RT. Counterstaining was performed by either DAPI or Hematoxylin.

## RNAscope assay

RNAscope mRNA in situ hybridization assay[130] for YAP or YAP/CVB3-4 double staining was performed using the RNAscope 2.5 HD Detection Duplex Reagent RNAscope kit (#322430, Advanced Cell Diagnostics) according to the manufacturer's instructions. Human Yap1 (#419131-C2; ACD), human CVB (#409301, V-CVB4; #409291, V-CVB3) and human CTGF (560581-C1; ACD) probes were used to detect Yap1, CVBs and CTGF mRNAs. Briefly, tissue sections were incubated for 1 h at 60 °C, deparaffinized and rehydrated by xylene and 100% ethanol for 10 and 2 min, respectively. Target retrieval was performed for 15 min at 95-97 °C, followed by protease treatment for 15 min at 40 °C. Probes were then hybridized for 2 h at 40 °C followed by repeated washing with wash buffer and then kept in 5x Saline-sodium citrate (SSC) buffer overnight. RNAscope amplification was carried out using two independent signal amplification systems based on HRP and AP labeled probes and ultimately visualized by red and green chromogenic substrates. At the end, sections were counterstained with Hematoxylin.

## Single molecule fluorescence in situ hybridization (smFISH)

smFISH was used to detect enterovirus mRNA in pancreatic tissue sections by using single-molecule oligonucleotide probes carried out according to the highly sensitive protocol that was previously established in our lab[19]. FISH Probes were synthesized by Stellaris® (Biosearch Technologies, Inc.; Petaluma, CA, USA), and labeled with Quasar 570[9,19]. The three probes sets recognizes various enteroviral strains for positive strand enteroviral RNA based on sequence similarities. FFPE sections were deparaffinized with Xylene for 30 min at 70 °C and

10 min at room temperature then rehydrated in 100, 95, and 70% ethanol for 20, 10, and a minimum 60 min respectively. Sections were covered with 0.2 M HCL for 20 min followed by washing in prewarmed 2xSSC for 15 min in a shaking water bath at 70 °C. For antigen retrieval, pepsin was used for 10 min in 37 °C humidified chamber and washed two times with PBS. Before hybridization, samples were equilibrated 2 times with buffer made by 10% formamide and 2XSSC. Probes hybridized overnight at 37 °C. Next day slides underwent several times of washing at 37 °C in a shaking water bath including 2xSSC plus 10% formamide for 40 min, 2xSSC 30 min, 1xSSC 30 min, 0.1xSSC for 20 min. Thereafter, classical immunostaining was performed for YAP and DAPI as detailed above. A 60x oil-immersion objective was used to acquire images by a Nikon Ti MEA53200 (NIKON GmbH, Düsseldorf, Germany) microscope.

### Morphometric analysis
Morphometric analysis involved the examination of ten sections per mouse, spanning the pancreas width. Pancreatic tissue area and insulin-positive area were quantified through computer-assisted measurements using a Nikon MEA53200 microscope (Nikon GmbH, Germany), and images were captured with NIS-Elements software from Nikon. The average percentage of β-cell fraction per pancreas was computed as the ratio of insulin-positive area to the entire pancreatic tissue area. Pancreatic β-cell mass was obtained by multiplying the β-cell fraction by the weight of the pancreas[27].

### Image analysis and quantification
Images were obtained using an inverse Nikon Ti2-A MEA54100 (NIKON GmbH, Düsseldorf, Germany) microscope with NIS-Elements Software (BR-ML). To quantify the YAP-positive area in the human exocrine pancreas, 229 different fields (independent positions) from 13 control donors, 223 from 15 AAb⁺ donors, and 284 from 15 T1D donors were analyzed for YAP intensity by Image J.JS (v0 5.6) and data presented as % of YAP-positive area. The YAP-positive fraction in the islet was quantified manually by counting the number of YAP-positive cells in the pancreatic islet normalized to the number of all chromogranin-positive cells in the pancreas. % double YAP/chromogranin-positive cells were quantified by the number of double positive cells normalized to the number of chromogranin-positive cells. The infection rate in INS-1E cells was calculated by counting the number of VP1-positive cells divided by all cells from 40-50 randomly captured images under the 60x objective throughout the well. In PANC-1 cells, the same analysis was carried out with 12 randomly captured images under the 20x objective. Total number of cells in each image was quantified by manually counting all DAPI-stained nuclei using NIS-elements and used for normalization and to calculate the percentage of VP1-positive cells in the respective images. To quantify YAP-CVB3/4 double positive cells from RNAScope, infected cells were classified into two categories, low or single infection (5-10 puncta/cell) and full or cluster infection (>10 puncta/cell). Neighboring YAP-positive cells were the cells located exactly next to the infected cell. RNAScope YAP-mRNA was quantified by counting cells with YAP⁺ puncta normalized to the number of all nuclei. CTGF RNA scope analysis was done at 400X using NIS elements software. At least 15 non-overlapping images from each donor were quantified; the total number of CTGF puncta was divided by the total number of nuclei in each field. In a 2nd analysis, number of nuclei were counted and classified to 0–4, 5–15 and > 15 CTGF puncta. Apoptosis and infection in isolated human islets were quantified by double-positive TUNEL/insulin or VP1/insulin cells normalized to all insulin-positive cells for each islet.

### Reanalysis of public bulk or single-cell RNA sequencing data
**In-silico**
**Data acquisition.** The raw data for T1D and normal pancreas samples were obtained from the study by Elgamal et al.[39], which conducted scRNA-seq on pancreas samples from individuals with T1D and non-diabetic controls. Additionally, we used the dataset from Yang et al.[67], which includes scRNA-seq data from human islets exposed to SARS-CoV-2, CVB4 and Mock conditions. From this dataset, we selected only the Mock and CVB4 samples. Furthermore, we incorporated RNA-seq data of human stem cell-derived β (SC-β) cells infected with CVB at various time points from Nyalwidhe et al.[47]. Data analysis was performed using the Scanpy pipeline[131].

**Pre-processing.** Data from Elgamal et al. were already preprocessed, no further modifications were made to the data. From Yang et al. we filtered out cells with fewer than 500 genes, more than 6000 genes, fewer than 1000 counts, or more than 60,000 counts. Additionally, cells with a mitochondrial gene fraction exceeding 15% were excluded. Normalization of gene expression was carried out to minimize biases in cell counts and enhance the comparability of intracellular expression levels. This was achieved using Scanpy's normalize total function or by calculating size factors for each cell. For integration and batch effect correction, we utilized scvi-tools[132], applying 4000 highly variable genes and including the sample object as a covariate. From RNA-seq data from Nyalwidhe et al. we performed normalization and logarithmic transformation before analyzing the expression levels of the genes of interest.

**Cell type identification and GSEA.** For the CVB-infected scRNA-seq data, we utilized CellTypist[133], an automated cell type annotation tool, and applied the pancreas-specific model to classify the cells. GSEA was performed using the GSEAPY[134] tool with the GO Biological Process 2023[135] gene sets, applying a Kolmogorov-Smirnov-like enrichment score. Statistical significance was assessed via permutation testing followed by false discovery rate (FDR) correction. A custom YAP-target gene signature comprises of 22 well-established genes, as identified by Wang et al.[40] and included the following genes: *CCN1*, *CCN2*, *AMOTL2*, *IGFBP3*, *F3*, *FJX1*, *NUAK2*, *LATS2*, *CRIM1*, *GADD45A*, *TGFB2*, *PTPN14*, *NT5E*, *FOXF2*, *AXL*, *DOCK5*, *ASAP1*, *RBMS3*, *MYOF*, *ARHGEF17*, *CCDC80*, and *MMP7*.

### Statistical analyses
All statistics were performed using GraphPad Prism software (GraphPad Software Inc.). Statistical comparisons between groups were analyzed for significance by a paired or unpaired two-tailed Student's t-test and a one-way or two-way analysis of variance (ANOVA) with Holm-Sidak multiple comparisons correction. A Spearman correlation analysis was used to assess the correlation between YAP protein expression and other markers. $P$ value < 0.05 was considered statistically significant. Data are presented as means ± SEM. The exact values of $n$ (refers to number of donors or mice, or number of independent biological experiments or independent measurements/positions) are reported in the figure legends. Exact $P$-values are reported either directly in the figures or in the corresponding figure legends.

### Reporting summary
Further information on research design is available in the Nature Portfolio Reporting Summary linked to this article.

## Data availability
Source data are provided as a Source Data file. Data for Fig. 2 are reused and available on the Gaulton lab webpage under accession Islet_expression_HPAP. Data for Figure S6 are available in the GEO database under accession code GSE145074. Data for Figure S10 are available in the GEO database under accession code as GSE247809. All other data are available in the article and its Supplementary files or from the corresponding author upon request. Source data are provided with this paper.

## Code availability

The code to reproduce the results is available at https://github.com/ArdestaniLab/YAP_T1D and is archived in Zenodo at https://doi.org/10.5281/zenodo.16755901.

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

## Acknowledgements

This research was performed with the support of the Network for Pancreatic Organ donors with Diabetes (nPOD; RRID:SCR_014641), a collaborative type 1 diabetes research project supported by JDRF (nPOD: 5-SRA-2018-557-Q-R) and The Leona M. & Harry B. Helmsley Charitable Trust (Grant#2018PG-T1D053, G-2108-04793). The content and views expressed are the responsibility of the authors and do not necessarily reflect the official view of nPOD. Organ Procurement Organizations (OPO) partnering with nPOD to provide research resources are listed at https://npod.org/for-partners/npod-partners/. We express our deep gratitude to the donors and their families. We are grateful to Irina Kusmartseva (University of Florida, Miami) for help with donor procurement and her encouragement and discussion throughout this study. We are grateful to our colleagues from the nPOD viral working group (nPOD-V) for discussions. Human pancreatic islets were kindly provided by the NIDDK-funded Integrated Islet Distribution Program (IIDP) at City of Hope, NIH grant no 2UC4DK098085, the JDRF-funded IIDP Islet Award Initiative, and through the ECIT Islet for Basic Research program supported by JDRF (JDRF award 31-2008-413). This manuscript used data acquired from the database (https://hpap.pmacs.upenn.edu/) of the Human Pancreas Analysis Program (HPAP; RRID:SCR_016202)[136]. HPAP is part of a Human Islet Research Network (RRID:SCR_014393) consortium (UC4-DK112217, U01-DK123594, UC4-DK112232, and U01-DK123716). We thank J. Kerr-Conte and Francois Pattou (European Genomic Institute for Diabetes, Lille) and ProdoLabs for high-quality human islet isolations, Katrischa Hennekens and Keno-Grace Bechtgold (University of Bremen) for excellent technical assistance and animal care, Petra Schilling (University of Bremen) for pancreas sectioning, Fernando Camargo (Boston Children Hospital, Boston, MA, USA) and AI Powers (Vanderbilt University medical Center, Nashville, TN, USA) for sharing transgenic mice for this study and Cenap Güngör, Universitätsklinikum Hamburg-Eppendorf (UKE) for PANC1 cells and Claes Wollheim (Geneva & Lund University) for INS-1E cells. This research used HeLa cells, which were originally derived from Henrietta Lacks without her knowledge or consent. We thank her and her family for their contributions to science. This study was supported by a Breakthrough T1D (formerly JDRF) grant (JDRF (3-SRA-2017-492-A-N) to the nPOD-Virus Group (PI: Alberto Pugliese), by the German Research Foundation (DFG; MA 4172/15-1, MA 4172/20-1 to K.M., AR 980/4-1 to A.A.) and by Breakthrough T1D (2-PAR-2014-275-I-X to K.M., 3-APF-2019-748-A-N to A.A.). The funders had no role in study design, data collection and analysis, decision to publish, or preparation of the manuscript.

## Author contributions

Conceptualization A.A.; Experimental design K.M., A.A., M.H.; Methodology S.G., H.P., K.M., A.A.; Investigation S.G., H.L., H.P., M.K.M., F.A., S.R., A.E.K., M.K., P.B., D.G., M.E., A.M.G., D.B., B.L., O.Z., R.M.E., K.M., A.A.; Formal Analysis S.G., H.L., H.P., M.K.M., F.A., K.M. and A.A.; bulk and scRNASeq-Reanalysis A.M.F.; Writing-Original Draft A.A.; Writing - Review & Editing S.G., H.L., H.P., M.K.M., A.M.F., Z.A., A.P., K.M., A.A; Resources K.B., M.H., A.P.; Funding Acquisition and Supervision K.M., A.A.

## Funding

## Competing interests

The authors declare no competing interests.
