## [Transparent Peer Review file · Nature Communications]

The Hippo terminal effector YAP boosts enterovirus replication in type 1 diabetes

Corresponding Author: Dr Amin Ardestani

Version 0:

Reviewer comments:

Reviewer #1

(Remarks to the Author)

Geravandi et al have discovered that the Hippo terminal effector Yes-associated protein (YAP) is expressed with increased intensity in pancreata of organ donors positive for pancreatic islet autoantibodies (AAb) or diagnosed with type 1 diabetes (T1D). In donors positive for coxsackievirus, YAP expression was seen in cells positive for coxsackievirus genome and in cells adjacent to virus positive cells. Through the results of various in vitro and ex vivo experiments in cells of pancreatic exocrine and endocrine origin, the authors concluded that presence of active YAP facilitates coxsackievirus replication. Increased virus replication was associated with higher expression of type I IFN and genes associated with innate immune responses, suggesting that YAP has a direct or indirect immunoregulatory function during coxsackievirus infection. In addition, it was found that the artificial introduction of active YAP or coxsackievirus infection increased the expression of STK4/MST1. MST1 is a negative regulator of YAP, as the authors also demonstrated. Based on their results, the authors propose that YAP is of importance for coxsackievirus replication but simultaneously induces MST1 as a negative feedback loop to avoid excessive activation of YAP and associated heightened virus replication.

This is a carefully performed study using a large panel of methods and material to study the possible role of active YAP in coxsackievirus infection. The authors present interesting data suggesting that YAP activation is associated with a concomitant induction of a negative feedback loop through the activation of MST1. An interesting scenario would have been if coxsackievirus infection resulted in increased YAP expression or activation of constitutively expressed YAP, as YAP is known to positively control the expression of genes involved in the cell cycle and coxsackievirus replication is much more efficient in cells which are actively dividing compared to those who are not (e.g. PMID: 11932410). However, judging by the data presented in Figures 3M, 5A and 5F (controls + CBV4), this does not appear to be the case. Instead, the virus induces the expression of MST1 which is shown by the authors to negatively regulate virus replication.

The study is of significant interest because it demonstrates novel cellular pathways that regulates coxsackievirus replication. Drugs that have antiviral activity against coxsackieviruses are lacking, and the findings presented here provide information about a new cellular pathway that could be targeted by a drug with a goal to attenuate virus replication. On the other hand, the study leaves the reader with the question of the biological role of increased pancreas YAP protein expression in the etiology of the disease type 1 diabetes. The title claims that “..YAP boosts enterovirus replication in type 1 diabetes”. Given that the authors have presented data clearly showing that coxsackievirus replication is facilitated by active YAP and that YAP is efficiently regulated by MST1, the authors would at least have to demonstrate that a) YAP is in active form in the pancreases studies (or signs thereof) and b) the expression of MST1 is not upregulated in the pancreases with increased YAP expression, to make this claim.

In addition to what was mentioned above, the following improvements would strengthen the work:

1. Please, explain in the text how the human pancreas cases were selected. Was it known in advance whether they were positive for viruses and selected based on this, or randomly picked (i.e. how well do the selected cases represent individuals with AAbs or type 1 diabetes)? The number of cases studied varies depending on what analysis was carried out, e.g. in Figure 1 A and B there were 13 controls, 15 AAb+ donors and 15 T1D donors compared to 10 controls, 10 AAb+ donors and 15 T1D donors in Figure 1 C and D. For RNA scope the numbers were different again. Why is it like this?

2. It was interesting to see that there was no correlation between YAP expression and age, BMI or Hb1AC. Was there any

difference in expression between single AAb+ and double/multiple AAb+ donors?

3. To appreciate the variation in YAP expression between donors, it would be of interest to see that the data presented in Figure 1B is presented as in Figure 1D, i.e. the mean YAP+/pancreas area (%) per donor and not per "independent position".

4. The human islet protein blot shown in Figure 3M suggests that YAP is expressed by isolated human pancreatic islets (e.g. Ad-LacZ control, lane 1). This seem to be at odds with what is shown in the tissue sections. Is CBV replication reduced if cells are treated with VP when infected in vitro?

5. Page 8, line 261 reads:also IFN-induced expression of CXCL10... should probably read:also virus-induced expression of CXCL10...

Reviewer #2

(Remarks to the Author)

The manuscript by Geravandi et al. investigates the role of the Yes-associated protein (YAP) in regulating enteroviral amplification in the pancreas. Cadaveric human donor pancreatic tissue sections, rat insulinoma cells (INS-1E), and primary human islets were used to elucidate the role of YAP in pancreatic beta cells. The authors identified that YAP expression was elevated in both the exocrine and endocrine compartments of autoantibody positive (AAb+) donors as well as in donors with type 1 diabetes (T1D). In addition, through overexpression and YAP inhibitors in both INS-1E and human islets, the authors show that YAP controlled the replication of Coxsackievirus B (CVB, a known inducer of islet inflammation). The authors also identify that the mammalian STE20-like protein kinase 1 (MST1) negatively regulates CVB induced viral replication. Overall, the study is interesting and provides new insight that potentially contributes to the understanding of T1D pathophysiology. Weaknesses include some concerns over rigor of tissue immunostaining and the lack of a disease model in vivo, which would otherwise make the relevance to T1D more convincing. Specific concerns are enumerated below:

Major

1. Figure 1A: The uniformity of YAP staining within the pancreas cannot be gauged using the representative images provided. A low-magnification image or a full scan of the pancreata stained with YAP with high-magnification insets (as shown currently) would be beneficial in gauging the uniformity of staining.
2. Supp. Figure 1B and Figure 1F: Based on the representative images provided, it does not appear that YAP is colocalized with chromogranin, and YAP staining seems to be outside the chromogranin staining. Yet, quantification in Figure 1F shows that there is an increase in the percentage of YAP+/Chromogranin+ cells. The evidence provided is not particularly convincing.
3. Figure 2C and D: There is discrepancy with the results- total percentage of YAP+/CVB+ cells in Figure 2C (purple) is slightly increased in T1D. However, Figure 2D shows a twofold increase in T1D.
4. Figure 2E: Please show also Virus+/Endocrine cells
5. Figure 2F: Please provide quantification of YAP+/Enterovirus mRNA+ cells
6. Figure 3O: There are more TUNEL+ cells in CVB3 or CVB4 treatment groups. However, it looks like TUNEL+ cells are outside the INS+ region.
7. Suppl. Figure 3F-G: Why is there a reduction in apoptosis in control samples with YAP overexpression while Figure 3M-N shows an increase?
8. Figure 5F-G: Why is there a reduction in cleaved caspase 3 while the viral load (VP1 protein level) is increased? The discussion points can be improved.
9. Inclusion of a disease model in vivo would have greatly strengthened the study and its relevance to T1D. Experiments using virus induced diabetes mouse models would add more impact to the study. For example, YAP inhibitors or beta cell specific-AAV8- based YAP overexpression system in virus induced diabetes mouse models would provide direct evidence of whether modulating YAP expression in the beta cells would alleviate virus induced diabetes or not.
10. Statistics: multiple different statistical tests and post-tests were performed for the data presented. The rationale for which each test was performed is not clear. A statement in the statistics sections of the methods should justify when and why each specific test was preferred.

Minor concerns:

1. The authors discuss the negative regulation of YAP by MST1 in both introduction and discussion, but no reference to prior literature is provided.
2. Figure 2A: No labeling is provided for the panels. Are the authors showing the following in the three panels? YAP+/CVB+, nYAP+/CVB+, or YAP-/CVB+
3. Figure 3A, D: Labeling is incorrect- First column says "C" whereas CVB4 treatments were performed.
4. Figure 4L-N: Individual data points needs to be shown.
5. VP1 levels seem to be elevated (Figure 5G) when compared to YAP-GFP but not labeled as significant (although there is at least 2-fold increase). Similar data with siRNA (Figure 5A-B) with just a slight increase is labeled as significant. Please clarify that all comparisons that are significant are labeled. Notably, it is indicated that the significance is assessed by two-tailed t test, rather than one-way ANOVA for multiple comparisons. Please clarify if this is correct. It would appear that ANOVA would be the more appropriate statistical test.

Reviewer #3

(Remarks to the Author)

Geravandi et al. found that the level of YAP was increased in the exocrine and endocrine pancreas of AAb+ and T1D organ donors. They show that CVB-infected pancreatic cells were co-localized with YAP or located near the YAP-positive cells in AAb+ 358 and T1D pancreases, suggesting a pathological association between YAP and enteroviruses. They also showed that hyperactivation of YAP in cell-culture models of β -cells enhances the replication of CVB, β -cell apoptosis, and the expression of genes involved in innate immunity and antiviral defense. Finally, they provide a negative feedback mechanism by showing that the level of MST is increased by YAP overexpression in a YAP-TEAD-dependent manner. Overall, the authors found an unexpected role for YAP as a host factor for enteroviral amplification in the pancreas and provide therapeutic points for treating T1D. The authors provided exciting findings, and the data quality is sound. However, the main problem of this manuscript is that the authors only provided their findings and co-relations without providing mechanisms for the role of YAP in fostering viral DNA replication and how CVB infection enhances YAP expression in T1D pancreases. Due to the prominent deficits, although their findings are interesting, I hesitate to support this manuscript to be accepted in Nature Communications. The following are specific points that should be improved.

1. The authors showed that the mRNA and protein levels of YAP were highly upregulated in the pancreas of T1D and AAb+ organ donors. It would be interesting to know how the level of YAP is upregulated. They should have examined whether the upstream regulators of Hippo signaling components are differentially regulated in the pancreas of T1D and AAb+ organ donors by examining the changes of phosphorylation MST, LATS, and YAP and nuclear localization of YAP. Also, it would be crucial to examine how CVB infection enhances the level of YAP.
2. The authors showed that hyperactivation of YAP in cell-culture models of β -cells enhances the replication of CVB and β -cell apoptosis. They should have examined how the activation of YAP enhances the replication of CVB. It is well-known that ectopic expression of YAP enhances the expression of genes involved in anti-apoptosis. The authors need to provide mechanisms for β -cell apoptosis by hyperactivation of YAP.
3. The authors show that infection of human islets with CVB3 and CVB4 induced a potent type I interferon response, and YAP overexpression further enhanced it. The authors need to provide mechanisms for how YAP synergistically enhances immune responses.
4. The authors showed an increase in total MST1 protein level in the YAP-overexpressing or CVB4 infected β -cells and suggested that it serves as a negative feedback loop. It would be interesting to know how CVB4 infection enhances, and both CVB4 infection and YAP OE synergistically increases the MST level.
5. It is curious to know that ectopic expression of YAP increased the level of MST in a pancreatic cell-specific manner. Usually, ectopic expression of YAP does not increase the level of MST in other cell lines.

Version 1:

Reviewer comments:

Reviewer #1

(Remarks to the Author)

The authors have conducted extensive additional studies in response to my comments and those of the other referees. The manuscript has been significantly improved, and I have no further requirements.

Reviewer #2

(Remarks to the Author)

The authors have performed a comprehensive revision and have addressed all my previous concerns. Excellent revision!

R. Mirmira

Reviewer #3

(Remarks to the Author)

Geravandi et al. have extensively revised the manuscript in response to the reviewers' comments. They have performed many experiments to answer my questions, many of which were common questions raised by other reviewers, and I really appreciate the authors' efforts. However, although the authors provide a lot of new data, in many cases they have not provided the specific answer to my question. In the first review, I wrote that the quality of their data was good. I did not ask for more data to confirm that CVB infection increases YAP mRNA levels. I accept their findings that CVB infection increases YAP levels and that increased YAP enhances CVB-mediated islet inflammation and β -cell apoptosis. However, my point was that the authors need to provide a mechanism for the large gap between CVB infection and the increase in YAP levels. I also asked the authors to provide mechanisms for how YAP enhances CVB-mediated islet inflammation and β -cell apoptosis. To my first questions, the authors replied that the transcriptional upregulation of YAP and the resulting increase in total YAP protein is the primary mechanism. I don't think their answer is a correct answer to my question.

I have more concerns about the authors' response to my second point. Basically, the authors replied that YAP overexpression provides a rich environment for efficient CVB replication and amplification, based on the data obtained using poly(I:C) infection and Pleconaril treatment. Based on their experimental data, they concluded that the pro-apoptotic and pro-inflammatory effects of YAP were dependent on viral replication. I could not understand their logic. As shown in Figure 7C,

overexpression of YAP alone did not induce an inflammatory response. If Pleconaril was given after viral infection to inhibit viral replication, wouldn't the situation be the same as if only YAP was overexpressed? Also, poly(I:C) infection is in some ways similar to CVB infection, but I cannot accept the conclusion that the lack of potentiation of the inflammatory response by poly(I:C) is due to the lack of replication ability of poly(I:C). It is possible that some other factors in CVB, not present in poly(I:C), may potentiate the inflammatory response.

If the pro-proliferative property of YAP is the main mechanism for YAP-mediated viral replication and amplification, they should have performed an experiment to test whether overexpression of a pro-proliferative oncogene such as c-myc can enhance the CVB-mediated inflammatory response to prove their hypothesis.

Version 2:

Reviewer comments:

Reviewer #3

(Remarks to the Author)

The authors have added supporting mechanisms and a detailed explanation of the significance of viral replication to the Discussion section, as well as additional references to the revised manuscript, which has fully resolved my concerns. Thank you to the authors for their efforts; I now fully support the acceptance of this manuscript.

REVIEWER COMMENTS

Reviewer #1 (Remarks to the Author):

1) Geravandi et al have discovered that the Hippo terminal effector Yes-associated protein (YAP) is expressed with increased intensity in pancreata of organ donors positive for pancreatic islet autoantibodies (AAb) or diagnosed with type 1 diabetes (T1D). In donors positive for coxsackievirus, YAP expression was seen in cells positive for coxsackievirus genome and in cells adjacent to virus positive cells. Through the results of various in vitro and ex vivo experiments in cells of pancreatic exocrine and endocrine origin, the authors concluded that presence of active YAP facilitates coxsackievirus replication. Increased virus replication was associated with higher expression of type I IFN and genes associated with innate immune responses, suggesting that YAP has a direct or indirect immunoregulatory function during coxsackievirus infection. In addition, it was found that the artificial introduction of active YAP or coxsackievirus infection increased the expression of STK4/MST1. MST1 is a negative regulator of YAP, as the authors also demonstrated. Based on their results, the authors propose that YAP is of importance for coxsackievirus replication but simultaneously induces MST1 as a negative feedback loop to avoid excessive activation of YAP and associated heightened virus replication.

This is a carefully performed study using a large panel of methods and material to study the possible role of active YAP in coxsackievirus infection. The authors present interesting data suggesting that YAP activation is associated with a concomitant induction of a negative feedback loop through the activation of MST1. An interesting scenario would have been if coxsackievirus infection resulted in increased YAP expression or activation of constitutively expressed YAP, as YAP is known to positively control the expression of genes involved in the cell cycle and coxsackievirus replication is much more efficient in cells which are actively dividing compared to those who are not (e.g. PMID: 11932410). However, judging by the data presented in Figures 3M, 5A and 5F (controls + CBV4), this does not appear to be the case. Instead, the virus induces the expression of MST1 which is shown by the authors to negatively regulate virus replication.

Response:

Thank you for this insightful and detailed feedback. We appreciate the opportunity to address the important point raised regarding the relationship between CVB infection, YAP expression, and MST1 regulation. At the time of the initial submission, we had begun investigating this aspect in greater depth. As the reviewer suggests, we now confirm that coxsackievirus infection indeed results in increased YAP mRNA (major transcript *Yap1*) expression in a time-dependent manner. This observation is critical in understanding the interplay between viral replication and host cellular pathways. The reviewer correctly notes that the data in the original Figure 3M (now Figure 6L) suggested no induction of YAP by CVB4. However, due to the rapid and strong appearance of exogenously transduced YAP in the gel, we were unable to capture endogenous YAP expression. To address this, we have now conducted a comprehensive, time-dependent analysis of CVB-induced *Yap1* expression and its downstream target genes in human islets and exocrine cells. Our new data (presented in Figure 5) clearly demonstrate a transient increase in *Yap1* mRNA levels at 6 and 12 hours post-infection (p.i.) and a corresponding increase in YAP protein levels at 12 and 24 h p.i. To further substantiate this finding, we reanalyzed publicly available RNA-seq data from human stem cell-derived β (SC- β) cells exposed to CVB4. This analysis corroborates our results, showing an induction of YAP target genes in CVB-infected SC- β cells (see Figure S6). Importantly, our data from original submission also reveal that CVB infection upregulates MST1, aligning with the model that MST1 serves as a compensatory negative regulator in response to increased YAP activity. We propose that CVB-induced YAP upregulation is a primary mechanism by which the virus enhances proliferative, transcriptional, and translational gene networks necessary for viral replication and amplification. However, the concomitant

upregulation of MST1 acts as a feedback mechanism to restrain excessive YAP activation, thereby limiting viral replication. This hypothesis is supported by our genetic MST1 inhibition experiments, where MST1 knockdown significantly enhanced CVB replication (now Figure 10). These findings suggest that MST1, induced as a result of CVB-driven YAP upregulation, plays a potent inhibitory role in controlling viral replication. This dynamic highlights the virus's ability to exploit host cell pathways for replication while simultaneously triggering cellular mechanisms to modulate this process.

We also appreciate the reference to the work by Ralph Feuer et al. (PMID: 11932410). Motivated by the reviewer's insight, we reanalyzed the publicly available dataset of single-cell RNA sequencing (scRNA-seq) from human pancreatic cells infected with CVB4, which was recently published¹. Our goal was to assess the cellular proliferative status in relation to YAP expression under CVB infection. We focused on CVB4-infected cells from two major YAP-expressing pancreatic cell types that are not quiescent and exhibit relative proliferative activity: ductal and stellate cells. Based on the cutoff for *Yap1* expression, we categorized these cells into two subpopulations: "High YAP" and "Low YAP". Gene Set Enrichment Analysis (GSEA) of Hippo signaling confirmed the enrichment of YAP target genes in the High YAP versus Low YAP groups (Figure S10). Further GSEA analysis revealed that pathways and processes directly linked to proliferation - such as positive regulation of cell population proliferation, DNA replication, mitotic cell cycle, and mRNA transcription - were highly enriched in the CVB4-infected High YAP subpopulation compared to the Low YAP subpopulation in both ductal and stellate cells (Figure S10). This finding suggests a clear correlation between YAP expression and proliferation in CVB4-infected cells. These data align with the reviewer's point that CVB replication is more efficient in actively dividing cells, which are enriched for high YAP expression. However, it is also important to note, as supported by our initial study², that even quiescent pancreatic cells such as islet cells can harbor viral RNA for extended periods. This persistence may contribute to the long-term retention of Coxsackievirus in the pancreas without active viral production. Thus, the active replication of the virus in proliferating exocrine cells, along with subsequent β -cell infection and the retention of viral RNA, may support the notion that CVB infection establishes a self-perpetuating cycle of replication, persistence, inflammation, and immune activation within the pancreas. This cycle could contribute to the pathophysiology of T1D.

2) The study is of significant interest because it demonstrates novel cellular pathways that regulate coxsackievirus replication. Drugs that have antiviral activity against coxsackieviruses are lacking, and the findings presented here provide information about a new cellular pathway that could be targeted by a drug with a goal to attenuate virus replication. On the other hand, the study leaves the reader with the question of the biological role of increased pancreas YAP protein expression in the etiology of the disease type 1 diabetes. The title claims that "...YAP boosts enterovirus replication in type 1 diabetes". Given that the authors have presented data clearly showing that coxsackievirus replication is facilitated by active YAP and that YAP is efficiently regulated by MST1, the authors would at least have to demonstrate that a) YAP is in active form in the pancreas studies (or signs thereof) and b) the expression of MST1 is not upregulated in the pancreases with increased YAP expression, to make this claim.

Response:

Thank you for raising this important point. We appreciate the opportunity to clarify and provide further evidence regarding the functional activity of YAP in the pancreas in T1D and its relevance to enterovirus replication. We are confident that YAP is upregulated at the transcriptional level in the pancreas of individuals with T1D, as demonstrated by *Yap1* mRNA upregulation (RNAscope) and increased YAP protein expression (IHC) (Figure 1). This is further supported by new data from CVB-infected *in vitro* cultures of human islets and exocrine cells (Figure 6), as well as reanalysis of publicly available datasets from CVB-infected cells (Figure S10E). These findings are detailed in our response to Comment 1. To address the

question of whether YAP is functionally active, we performed RNAscope analysis for *CTGF*, a well-established YAP target gene. Expression analysis across the pancreas revealed significant upregulation of *CTGF* in AAb+ and T1D pancreases, mirroring the pattern of YAP expression (see new Figure 3). Additionally, in our *in vitro* model, we observed upregulation of both *Yap1* and *CTGF* in islets following YAP overexpression (proof of concept) and in response to CVB infection (new Figure 5C). Supporting this, we re-analyzed single-cell RNA-seq data from the Human Pancreas Analysis Program (HPAP). GSEA revealed significant enrichment of the Hippo pathway score and YAP target genes (comprising 22 well-established genes) in β -cells and α -cells. This enrichment was also observed in major YAP-expressing pancreatic cells, including ductal and stellate cells, from individuals with T1D compared to non-diabetic controls. Collectively, these data provide strong evidence of elevated YAP activity in the pancreas of individuals with T1D.

Regarding MST1, we acknowledge the reviewer's point on the balance between YAP and MST1. In our previous study³, analysis of MST1 and phospho-MST1 (P-MST1) in paraffin-embedded pancreases of organ donors with T2D did not yield conclusive results due to limitations by the fixation protocol of organ samples. Such analyses could only be reliably performed in freshly isolated human islets with much shorter fixation protocol, which are not available from T1D donors. However, our *in vitro* data show that MST1 expression does not necessarily need to be downregulated for YAP activation to occur. MST1 can, in fact, be upregulated under conditions of high YAP activity, whether triggered by CVB infection or artificially induced by YAP overexpression (Figures 9 and S11). This aligns with the model of a YAP-driven feedback loop, wherein MST1 acts as a compensatory regulator to restrain excessive YAP activation. As outlined above, we propose that the transient upregulation of YAP induced by enterovirus infection is sufficient to trigger MST1 feedback. Notably, inhibition of MST1 further enhanced CVB replication, as evidenced by increased intracellular viral RNA and capsid VP1 protein levels (Figure 10). This suggests that MST1 upregulation acts as a critical checkpoint to limit viral replication driven by YAP.

Determining whether this feedback loop remains functional or becomes impaired in T1D requires further investigation, potentially through single-cell spatial analysis to simultaneously evaluate YAP and MST1 protein expression. We have now included this consideration in the discussion section (1st chapter page 17). Additionally, we investigated the efficacy of the antiviral drug pleconaril, a viral capsid inhibitor that has shown promise in a proof-of-concept clinical study in T1D⁴. *In vitro*, pleconaril effectively blocked CVB4 replication⁵, confirmed by us in human islets (Figure 7E) and prevented YAP-induced potentiation of inflammation during enteroviral infection (Figure 7F). This reinforces the idea that viral amplification is a key driver of YAP-induced inflammation, underscoring the therapeutic potential of targeting viral replication to mitigate YAP-driven pathogenic processes.

3) In addition to what was mentioned above, the following improvements would strengthen the work:

3.1) Please, explain in the text how the human pancreas cases were selected. Was it known in advance whether they were positive for viruses and selected based on this, or randomly picked (i.e. how well do the selected cases represent individuals with AAbs or type 1 diabetes)? The number of cases studied varies depending on what analysis was carried out, e.g. in Figure 1 A and B there were 13 controls, 15 AAb+ donors and 15 T1D donors compared to 10 controls, 10 AAb+ donors and 15 T1D donors in Figure 1 C and D. For RNA scope the numbers were different again. Why is it like this?

Response:

The nPOD tissue collection allows analyses from well-preserved organ donors throughout a large collaborative initiative. Analyses goes through an application process, as material is restricted. For this study, donor IDs were selected randomly and without bias for all analyses, with the exception of matching for key variables such as age, BMI, and gender across the three groups: control, AAb+, and T1D. This approach was taken to ensure the most representative and unbiased comparison possible between groups, minimizing potential confounding factors.

Regarding the variability in the number of cases analyzed across different experiments, this discrepancy primarily reflects the limited availability of specific tissue sections from certain donors. While our goal was to analyze more than 10 donors for key findings (such as in Figures 1A-D), the restricted availability of sections for some donors meant that confirmatory experiments were performed on 3-5 donors across multiple independent rounds of staining (such as Figures 1E-H). Importantly, for each experimental round, donors were selected to ensure equal representation across all groups, and analyses were blinded and conducted immediately following each round of staining. This methodological approach ensures the robustness and reproducibility of our findings while acknowledging the inherent limitations in donor tissue availability.

3.2) It was interesting to see that there was no correlation between YAP expression and age, BMI of Hb1AC. Was there any difference in expression between single AAb+ and double/multiple AAb+ donors?

Response:

Thank you so much for this insightful comment! Indeed, we found a correlation that aligns with our previous study². The cohort analyzed consisted of 4 double/multiple AAb+ donors and 11 single AAb+ donors. In our previous paper, which detected CVB RNA in the pancreas², we hypothesized an increased number of CD45+ immune cells infiltrating islets, as well as a higher number of enteroviral RNA+/CD45+ immune cells in double AAb+ donors compared to single AAb+ donors. However, the limited number of donors had prevented the observation from reaching significance.

Now, in this study, we found a significant increase in CD45+ immune cells within islets in double/multiple AAb+ donors compared to single AAb+ donors, when normalized to both β -cell area or to islet number in the pancreas (Figure S3D,E). Also, analysis of β -cell area in the pancreas revealed a significant reduction in double/multiple AAb+ donors compared to single AAb+ donors in this cohort, suggesting a correlation of islet size and double/multiple AAb+ status (Figure S3C). This raises the possibility that either an early reduction in islet size predisposes double/multiple AAb+ donors to T1D development, or that the presence of multiple AAbs leads to a decrease in β -cell area. Most interestingly, the percentage of YAP+ cells in islets, but not in the exocrine pancreas, was also significantly increased in double/multiple AAb+ donors compared to single AAb+ donors (Figure S3A,B).

3. To appreciate the variation in YAP expression between donors, it would be of interest to see that the data presented in Figure 1B is presented as in Figure 1D, i.e. the mean YAP+/pancreas area (%) per donor and not per "independent position".

Response:

We have done this analysis, and the differences remain consistent. The results are now additionally presented in Figure S2A. We prefer to also retain original Figure 1B, as it shows the extensive datasets derived from examining the entire pancreas, using over 15 independent shots/positions per donor. It also highlights the heterogeneous YAP expression observed even within a single donor.

4. The human islet protein blot shown in Figure 3M suggests that YAP is expressed by isolated human pancreatic islets (e.g. Ad-LacZ control, lane 1). This seem to be at odds with what is shown in the tissue sections...

Response:

Islet preparations typically contain at least 10-20% exocrine contamination, and the YAP expression in the exocrine tissue is a potential source of the YAP band observed in Western Blotting, we included this in the figure legend (now Figure 6).

...Is CBV replication reduced if cells are treated with VP when infected in vitro?

Response:

Indeed, CVB replication was reduced when cells were treated with VP during in vitro infection. We had already shown such data in both primary exocrine cells and the ductal cell line PANC1 in the presence of YAP/TEAD inhibitor VP (Figures 6 & S7). To further strengthen these findings, we now provide a 3rd model including four human islet isolations of mixed purity (50% exocrine/50% islets). This analysis shows that CVB replication, indicated by double staining for VP1 and insulin, was reduced in the presence of VP in both CVB3- and CVB4-infected human islets (see new Figure 6I). These results demonstrate that even endogenous YAP expression in CVB-infected exocrine cells can promote viral replication within islets.

5. Page 8, line 261 reads:also IFN-induced expression of CXCL10... should probably read:also virus-induced expression of CXCL10...

Response:

Both options are correct, as CXCL10 is a direct target of IFN and is induced by viruses. However, we have changed this as suggested by the reviewer.

Reviewer #2 (Remarks to the Author):

The manuscript by Geravandi et al. investigates the role of the Yes-associated protein (YAP) in regulating enteroviral amplification in the pancreas. Cadaveric human donor pancreatic tissue sections, rat insulinoma cells (INS-1E), and primary human islets were used to elucidate the role of YAP in pancreatic beta cells. The authors identified that YAP expression was elevated in both the exocrine and endocrine compartments of autoantibody positive (AAb+) donors as well as in donors with type 1 diabetes (T1D). In addition, through overexpression and YAP inhibitors in both INS-1E and human islets, the authors show that YAP controlled the replication of Coxsackievirus B (CVB, a known inducer of islet inflammation). The authors also identify that the mammalian STE20-like protein kinase 1 (MST1) negatively regulates CVB induced viral replication. Overall, the study is interesting and provides new insight that potentially contributes to the understanding of T1D pathophysiology. Weaknesses include some concerns over rigor of tissue immunostaining and the lack of a disease model in vivo, which would otherwise make the relevance to T1D more convincing. Specific concerns are enumerated below:

Response:

We would like to thank the reviewer for the thoughtful and constructive feedback on our manuscript. We are encouraged by the reviewer's recognition of the novelty and relevance of our study in contributing to the understanding of T1D pathophysiology. We appreciate the opportunity to address the concerns regarding tissue immunostaining rigor and the absence of an in vivo disease model.

In response to the reviewer's comments, we have undertaken additional work to strengthen the rigor and breadth of our study. For tissue immunostaining, we have now quantified hundreds of independent sections and images derived from the entire pancreas of control, AAb+, and T1D donors. These tissues were obtained under the strict regulations and application procedures of the nPOD pancreas collection, as detailed in the methods section. The application of stringent protocols ensured the reproducibility and accuracy of our immunostaining data.

To further substantiate our findings, we have expanded the immunostaining analyses by including:

- Immunostaining and quantification of the YAP target gene CTGF in the human pancreas (new Figure 3), confirming functional activation of YAP.

- Evaluation of viral replication upon YAP inhibition to block endogenous downstream YAP signaling in human islets (Figure 6I, Supplementary Figure 7B), highlighting the functional consequence of YAP perturbation in the context of CVB infection.
- Inclusion of low-magnification images of pancreata (Figure S1) to provide a broader and more representative view of the tissue architecture and staining patterns.
- Sub-cluster analyses of pancreata from multiple and single AAb+ donors (Figure S3), addressing inter-donor variability and ensuring robustness of our findings across independent samples.
- Quantification of YAP protein expression specifically in CVB-positive cells (Figure 4G), confirming a direct association between viral infection and YAP protein at the single-cell level.
- Investigation of the impact of viral replication on YAP-induced apoptosis and inflammation by targeting viral replication using replication-deficient double-strand RNA (Poly(I:C)) and antiviral inhibitor (Figure 7). These experiments provide mechanistic insight into how viral replication influences YAP-driven cellular responses.
- Reanalysis of publicly available single-cell RNA sequencing datasets from pancreata of T1D patients and CVB-infected pancreatic cells, reinforcing our conclusions through complementary transcriptomic data (Figures 2 and S10).

In addition to the *in vitro* and *ex vivo* analyses, we have now incorporated *in vivo* data from two independent mouse models to enhance the translational relevance of our findings:

1. Metabolic analyses in inducible β -cell-specific homozygous YAP overexpressing mice (Figures 8 and S9), demonstrating the functional impact of YAP overexpression on β -cell physiology and function and glucose homeostasis.
2. YAP expression analyses in the non-obese diabetic (NOD) mouse model, a well-established model of T1D. These mice were infected with CVB, and the effects on YAP expression were evaluated (Figure S5). This work was conducted in collaboration with Dr. Marc Horwitz at the University of British Columbia, further strengthening the *in vivo* relevance of our study.

We believe that the inclusion of these additional datasets, spanning tissue, cellular, molecular, and *in vivo* models, significantly enhances the robustness and comprehensiveness of our manuscript. We are confident that these extensive efforts address the reviewer's concerns and provide compelling evidence for the role of YAP in T1D pathophysiology, particularly in the context of viral infection.

Major

1. Figure 1A: The uniformity of YAP staining within the pancreas cannot be gauged using the representative images provided. A low-magnification image or a full scan of the pancreata stained with YAP with high-magnification insets (as shown currently) would be beneficial in gauging the uniformity of staining.

Response:

We now provide low-magnification images for YAP staining, which confirm the uniformity of YAP staining within the pancreas (please see Figure S1).

2. Supp. Figure 1B and Figure 1F: Based on the representative images provided, it does not appear that YAP is colocalized with chromogranin, and YAP staining seems to be outside the chromogranin staining. Yet, quantification in Figure 1F shows that there is an increase in the percentage of YAP+/Chromogranin+ cells. The evidence provided is not particularly convincing.

Response:

We thank the reviewer for raising this point. Physiologically, YAP is minimally or not expressed in fully differentiated, mature endocrine cells, as YAP is recognized as one of the disallowed genes in β -cells⁶⁻⁸. This explains why many YAP-positive cells do not colocalize with chromogranin (CHG). YAP expression is predominantly observed in the exocrine pancreas, which accounts for the presence of YAP-positive cells outside the islets. As shown in Figures 1C and 1D, we observed a striking increase in the number of YAP-positive cells within the islets of T1D patients. Our primary objective was to quantify the subset of YAP-positive cells of endocrine origin, which co-express CHG. The data presented in Figure 1F highlights this subset, demonstrating a significant increase in YAP+/CHG+ cells in T1D patients. A direct comparison of the total YAP+ cells within islets (Figure 1D) and the YAP+/CHG+ cells (Figure 1F) reveals that up to 19% of YAP+ cells in islets originate from endocrine cells in T1D. This finding underscores the relevance of YAP upregulation not only in non-islet cells but also in islet cells, contributing to the pathophysiology of T1D. Importantly, this observation aligns with the reanalysis of single-cell RNA sequencing datasets from HPAP, which showed that despite low baseline YAP expression in normal islet cells, YAP target genes are significantly upregulated in both α and β cells, in addition to their primary expression in ductal and pancreatic stellate cells (GSEA data, Figure 2). This clarification has been incorporated into the results section (1st paragraph) to ensure the rationale and significance of our approach are clearly conveyed.

3. Figure 2C and D: There is discrepancy with the results- total percentage of YAP+/CVB+ cells in Figure 2C (purple) is slightly increased in T1D. However, Figure 2D shows a twofold increase in T1D.

Response:

There seems to be a misunderstanding. Figure 2C (now Figure 4C) is not a comparison between AAb+ and T1D; rather, it presents the percentage fraction of each category of CVB+ cells for each group independently, without direct comparison between AAb+ and T1D donors. The presence of YAP expression in CVB+ cells in both AAb+ and T1D donors suggests that YAP may indeed be triggered in CVB-infected cells. We now show this through analyses of human islets/exocrine co-cultures infected with CVB (see new Figure 5). In contrast, Figure 2D (now Figure 4D) directly compares YAP+/CVB+ double-positive cells in AAb+ and T1D donors for both the single and cluster fractions. We have better clarified this now in the results part, 2nd paragraph.

4. Figure 2E: Please show also Virus+/Endocrine cells

Response:

Enteroviral RNA+ endocrine cells are rare, and in many cases, none were detected by single-molecule *in situ* hybridization (smFISH) using enterovirus-specific Stellaris probes (please see our previous study²). This was confirmed in the current study by the RNAScope method for CVB3/4-RNA detection in the FFPE pancreas samples. As a result, such a correlation would not be possible.

5. Figure 2F: Please provide quantification of YAP+/Enterovirus mRNA+ cells

Response:

We have now included quantification of YAP+protein-enteroviral RNA+ co-positive cells. In control donors, only a single enteroviral RNA+ cell was identified in this cohort, which was YAP-negative. Comparative analysis of YAP protein/enteroviral RNA shows increased numbers in both categories of YAP+/CVB+, and n-YAP+/CVB+ cells in T1D, compared to AAb+ donors (Figure 4G).

6. Figure 3O: There are more TUNEL+ cells in CVB3 or CVB4 treatment groups. However, it looks like TUNEL+ cells are outside the INS+ region.

Response:

We now provide improved representative images of TUNEL staining, clearly showing TUNEL⁺ in β -cells (insulin⁺); now moved to Figure 6P. In general, we observe reduced insulin level in response to CVB3/4, consistent with previous analyses showing reduced insulin expression and loss of β -cell function⁹⁻¹³. This is now included in the results, 5th paragraph.

7. Suppl. Figure 3F-G: Why is there a reduction in apoptosis in control samples with YAP overexpression while Figure 3M-N shows an increase?

Response:

We would like to thank the reviewer for this interesting observation. Indeed, there appears to be a small reduction in apoptosis in Figure S3F-G (now Figure 7A,B) in INS1E cells, while a slight increase in apoptosis is observed in human islets under control conditions (now Figure 6N,O). It is important to highlight that the experimental models used -INS1E cells and human islets- are different. In the pure β -cell INS1E model, YAP overexpression seems to exert an intrinsic anti-apoptotic effect, consistent with its well-documented oncogenic role in cancer cells. This effect is recapitulated in the proliferative, immortalized INS1E cell line under control condition. In contrast, human islet preparations, which include a mix of quiescent islet and low to moderate proliferating exocrine cells, display a slight increase in apoptosis. This difference could potentially stem from YAP's differential impact on exocrine cells present in the human islet preparation. However, under CVB infection, YAP consistently enhances β -cell apoptosis in both models, highlighting its pro-apoptotic role in the context of viral infection. As the mechanistic exploration of these differential responses falls outside the scope of the current study, and pooled data from several donors are not significant, we prefer not to speculate further at this stage.

8. Figure 5F-G: Why is there a reduction in cleaved caspase 3 while the viral load (VP1 protein level) is increased? The discussion points can be improved.

Response:

We thank the reviewer for this important observation. The data, now presented in Figure 10, illustrate the role of MST1 as a pro-apoptotic kinase that regulates cell death. When MST1 activity is inhibited - either through genetic knockdown or by overexpressing its dominant-negative form- CVB-induced cell death is reduced. However, we also show that MST1 deficiency potentiates viral replication in infected YAP-overexpressing cells. This highlights MST1's dual role, regulating viral replication through YAP as a classical upstream negative regulator in the Hippo pathway. Importantly, YAP, functioning as a transcriptional co-regulator, induces the transcription of STK4 (the gene encoding MST1), establishing a negative feedback loop that autonomously regulates MST1 expression. This feedback mechanism, identified in our study, acts to restrain excessive YAP activity, preventing hyperactivation that could lead to abnormal proliferation in normal cells or enhanced viral amplification during infection. Our data support this mechanism, as genetic MST1 inhibition leads to increased viral replication due to the loss of MST1's inhibitory action on YAP, allowing for unrestricted viral amplification. Simultaneously, the loss of MST1, which normally promotes apoptosis as a pro-death kinase, results in reduced apoptosis in infected cells. We have elaborated on this point in the results section (now the 8th paragraph) to provide greater clarity and address the reviewer's valuable feedback.

9. Inclusion of a disease model in vivo would have greatly strengthened the study and its relevance to T1D. Experiments using virus induced diabetes mouse models would add more impact to the study. For example, YAP inhibitors or beta cell specific-AAV8-

based YAP overexpression system in virus induced diabetes mouse models would provide direct evidence of whether modulating YAP expression in the beta cells would alleviate virus induced diabetes or not.

Response:

While we had initially focused our investigations on human T1D, we have now also included two mouse models which show the involvement of chronically elevated YAP in β -cell failure and induction of YAP expression in pancreases of CVB-infected mice. Unfortunately; very strict animal regulatory guidelines in Bremen did not allow the generation of new mouse models such as the combination of viral infection or AAV8-mediated gene delivery. However, we decided to test a similar approach using two available models;

(1) Chronic β -cell-specific homozygous overexpression of YAP has deleterious metabolic consequences. Chronic YAP re-expression in β -cells induced glucose intolerance by impairing insulin secretion and β -cell dedifferentiation. Extensive analyses of the pancreas and isolated islets from these mice revealed a decline in glucose stimulated insulin secretion, a significant reduction in essential β -cell genes related to maturation, function and identity, and the induction of beta cell dedifferentiation. These findings directly connect the pathological upregulation of YAP observed in the pancreas and in islets of organ donors with T1D to β -cell failure and metabolic deregulation. This is now included in new Figures 8 and S9.

(2) We have also analyzed YAP expression in a widely used mouse model of T1D, namely the non-obese diabetic NOD mouse. CVB accelerates diabetes in NOD mice, with mice developing diabetes as early as one-week post-infection^{14,15}. This makes CVB-infected NOD mice an ideal model to study whether pancreatic YAP expression is induced in the CVB-infected mice. As expected, we did not detect YAP expression in islets of WT C57Bl6/J mice (Figure S5). In contrast, NOD mice displayed YAP expression in islets throughout the pancreas. Intriguingly, both active (unphosphorylated) and total YAP were found within islets and increased in the exocrine pancreas of CVB3 and CVB4 infected mice. While this mouse model can provide crucial mechanistic insights into how and when YAP is activated, further investigation is required, which is beyond the scope of this study. These results are now included in Figure S5.

10. Statistics: multiple different statistical tests and post-tests were performed for the data presented. The rationale for which each test was performed is not clear. A statement in the statistics sections of the methods should justify when and why each specific test was preferred.

Response:

Thank you for your comment. Given that most experiments included a sample size of fewer than 10 per group, formal normality testing (e.g., D'Agostino-Pearson test) was not reliably accurate due to limited statistical power. In such cases, we followed standard practice in the field and applied parametric tests under the assumption that the data approximates normality, which is often reasonable for relatively small biological sample sizes ($n \leq 10$), as commonly accepted in biological sciences. To ensure consistency and avoid unnecessary complexity, group comparisons (e.g., control vs. infected, untreated vs. treated, etc.) were analyzed using a paired or unpaired two-tailed Student's t-test, where appropriate. For comparisons involving more than two groups (e.g., control, AAb+, and T1D or glucose tolerance tests (GTT), where we analyzed two factors - time and glucose levels), a one-way or two-way analysis of variance (ANOVA) or a mixed-effects model, with Holm-Sidak multiple comparisons correction, was used to account for multiple testing. The Statistics section has been revised accordingly. We appreciate the reviewer's input, which has helped us refine our statistical approach for improved clarity and rigor.

Minor concerns:

1. The authors discuss the negative regulation of YAP by MST1 in both introduction and discussion, but no reference to prior literature is provided.

Response:

Thanks lot; we have added an amazing review on the Hippo pathway by Kun-Liang Guan's lab¹⁶.

2. Figure 2A: No labeling is provided for the panels. Are the authors showing the following in the three panels? YAP+/CVB+, nYAP+/CVB+, or YAP-/CVB+

Response:

We had intended to only show the staining. Great idea to also label, done now in the 1st panel. These 3 different types are also shown in a larger magnification in Figure S4.

3. Figure 3A, D: Labeling is incorrect- First column says "C" whereas CVB4 treatments were performed.

Response:

Thank you! This is now corrected.

4. Figure 4L-N: Individual data points needs to be shown.

Response:

Individual data points have been added to Figure 4L-N (now Figure 8L-N).

5. VP1 levels seem to be elevated (Figure 5G) when compared to YAP-GFP but not labeled as significant (although there is at least 2-fold increase). Similar data with siRNA (Figure 5A-B) with just a slight increase is labeled as significant. Please clarify that all comparisons that are significant are labeled. Notably, it is indicated that the significance is assessed by two-tailed t test, rather than one-way ANOVA for multiple comparisons. Please clarify if this is correct. It would appear that ANOVA would be the more appropriate statistical test.

Response:

Thank you for your insightful comment. Upon re-evaluating our data, we realized that the statistical analysis for Figure 5G (now Figure 10G) was initially performed using a two-tailed t-test, whereas one-way ANOVA was used for Figure 5B (now Figure 10B). As these comparisons involve multiple groups, we agree with the reviewer's suggestion and have now applied one-way ANOVA for Figure 10G to ensure consistency and statistical appropriateness. With this correction, our updated analysis confirms that MST1-K59 overexpression significantly increases VP1 levels compared to GFP-transfected cells under YAP-overexpressing conditions. This finding aligns with the data from MST1 silencing, as presented in Figure 10B.

Reviewer #3 (Remarks to the Author):

Geravandi et al. found that the level of YAP was increased in the exocrine and endocrine pancreas of AAb+ and T1D organ donors. They show that CVB-infected pancreatic cells were co-localized with YAP or located near the YAP-positive cells in AAb+ and T1D pancreases, suggesting a pathological association between YAP and enteroviruses. They also showed that hyperactivation of YAP in cell-culture models of β -cells enhances the replication of CVB, β -cell apoptosis, and the expression of genes involved in innate immunity and antiviral defense. Finally, they provide a negative

feedback mechanism by showing that the level of MST is increased by YAP overexpression in a YAP-TEAD-dependent manner. Overall, the authors found an unexpected role for YAP as a host factor for enteroviral amplification in the pancreas and provide therapeutic points for treating T1D. The authors provided exciting findings, and the data quality is sound. However, the main problem of this manuscript is that the authors only provided their findings and co-relations without providing mechanisms for the role of YAP in fostering viral DNA replication and how CVB infection enhances YAP expression in T1D pancreases. Due to the prominent deficits, although their findings are interesting, I hesitate to support this manuscript to be accepted in Nature Communications. The following are specific points that should be improved.

Response:

Thank you for these important and valuable points. Addressing them has now significantly strengthened our findings. We now show:

- CVB indeed induces YAP expression in the exocrine pancreas as well as in islets, establishing a vicious cycle where viral infection induces YAP, and YAP, in turn, provides a supportive environment for viral amplification (Figure 5).
- We undertook mechanistic investigations into the impact of viral replication on YAP-induced apoptosis and inflammation by using replication-deficient double-strand RNA (Poly I:C). These experiments provide insight into how viral replication influences YAP-driven cellular responses, demonstrating that YAP exhibits pro-inflammatory effects primarily under the lytic influence of the virus. This is further validated by the inhibition of viral replication using pleconaril, antiviral inhibitor, which blocks YAP's pro-inflammatory effects (Figure 7).
- Beyond correlation, we now show that chronic β -cell-specific YAP overexpression directly impairs insulin secretion, induces β -cell dedifferentiation, and causes glucose intolerance. Analysis of pancreas and islets from these mice showed reduced glucose-stimulated insulin secretion and loss of key β -cell identity genes. These findings link pathological YAP upregulation in T1D to β -cell failure and metabolic dysfunction, now shown in Figures 8 and S9.
- Through reanalysis of scRNA-seq data from CVB4-infected pancreatic cells, we found that high YAP-expressing cells showed increased expression of proliferation-related genes and pathways. This suggests that CVB-infected cells with higher YAP activity are enriched in host pathway proliferation and cell cycle progression genes (Figure S10).

1. The authors showed that the mRNA and protein levels of YAP were highly upregulated in the pancreas of T1D and AAb+ organ donors. It would be interesting to know how the level of YAP is upregulated. They should have examined whether the upstream regulators of Hippo signaling components are differentially regulated in the pancreas of T1D and AAb+ organ donors by examining the changes of phosphorylation MST, LATS, and YAP and nuclear localization of YAP.

1a. Also, it would be crucial to examine how CVB infection enhances the level of YAP.

Response:

We appreciate this insightful comment and would like to highlight several key points that address the reviewer's comments. As noted in our response to Reviewer 1, we are confident that transcriptional upregulation of YAP is the primary mechanism underlying elevated YAP protein levels and activity in the pancreas of individuals with T1D.

This is demonstrated by:

1. *Yap1* mRNA upregulation (RNAscope) and the corresponding increase in YAP total protein expression (IHC) as presented in our original submission (Figure 1).
2. New data from CVB-infected in vitro cultures of human islets and exocrine cells clearly demonstrate a transient increase in *Yap1* mRNA levels at 6 and 12 hours post-infection (p.i.) and a corresponding increase in YAP protein levels at 24 hours p.i. (Figure 5).
3. To further substantiate this, we reanalyzed publicly available RNA-seq data from human pancreatic cells infected with CVB4. This analysis corroborates our findings, showing an induction of *Yap1* in CVB-infected pancreatic cells (Figure S10).
4. Our protein analyses of total YAP, active YAP (unphosphorylated form), and pYAP (S127, representing the cytoplasmic sequestered form) show that all forms are elevated following CVB infection. However, normalization of active YAP or pYAP to total YAP protein reveals no significant changes, suggesting that the increase in YAP protein levels is driven by transcriptional upregulation rather than post-translational modifications. The proportional increase in p-YAP reflects the overall rise in YAP protein expression (Figure 5).

To address the functional activity of YAP, we conducted the following complementary experiments:

1. RNAscope analysis for CTGF, a well-established YAP target gene, revealed significant upregulation across the pancreas in AAb+ and T1D donors, mirroring the pattern of YAP expression (new Figure 3). In our in vitro model, both YAP and CTGF were upregulated in islets following YAP overexpression (proof of concept) and in response to CVB infection (new Figure 5D).
2. Supporting this, re-analysis of single-cell RNA-seq data from the Human Pancreas Analysis Program (HPAP) revealed significant enrichment of the Hippo pathway score and YAP target genes in β -cells and α -cells, as well as in ductal and stellate cells from individuals with T1D compared to non-diabetic controls (GSEA analysis) (new Figure 2).
3. To further substantiate this, we reanalyzed publicly available RNA-seq data from human stem cell-derived β (SC- β) cells exposed to CVB4. This analysis corroborates our findings, showing an induction of YAP target genes in CVB-infected stem cell derived β cells (Figure S6).

Collectively, these data provide strong evidence of YAP transcriptional upregulation and subsequent elevated YAP activity in the pancreas of individuals with T1D as well as in CVB infection model of human pancreatic cells. While investigating post-translational modifications such as YAP phosphorylation could provide additional insights, we believe it may not be necessary for this study. Our previous attempts to immunostain pancreatic tissues for phosphorylated forms of YAP, MST, and LATS faced significant challenges, yielding unreliable results due to the known limitations of phospho-antibodies in fixed tissues. In our prior study³, we analyzed p-MST1 in paraformaldehyde-fixed, paraffin-embedded pancreases from T2D organ donors but obtained inconclusive results; only in freshly isolated islets with much shorter fixation protocol, analyses were possible.

We appreciate the reviewer's understanding of these technical challenges and believe the data provided sufficiently address the regulatory levels of YAP expression and activity.

2. The authors showed that hyperactivation of YAP in cell-culture models of β -cells enhances the replication of CVB and β -cell apoptosis. They should have examined how the activation of YAP enhances the replication of CVB. It is well-known that ectopic expression of YAP enhances the expression of genes involved in anti-apoptosis. The authors need to provide mechanisms for β -cell apoptosis by hyperactivation of YAP.

Response:

We thank the reviewer for this important comment. Actively replicating cells create a favorable environment for virus replication, a concept also highlighted by Reviewer 1. Our initial aim was to demonstrate that YAP promotes cell proliferation and inhibits apoptosis, including in β -cells, as shown by our previous study in human islets¹⁷ and now in mice through inducible β -cell-specific YAP overexpression. YAP significantly induced β -cell proliferation, as documented by two independent markers: Ki67 and BrdU incorporation. However, the consequence of this elevated β -cell replication is a loss of β -cell identity, dedifferentiation, and impaired insulin secretion (new Figures 8 and S9). We also previously demonstrated that these pro-proliferative effects are mediated by increased expression of FOXM1, a master regulator of cellular proliferation in β -cells, which controls G1/S transition, S-phase progression, and G2/M transition¹⁷. To directly associate proliferation to YAP expression under CVB infection, we reanalyzed the publicly available dataset of scRNA-seq from human pancreatic cells infected with CVB4, which was recently published¹. Our goal was to assess the cellular proliferative status in relation to YAP expression under CVB infection. We focused on CVB4-infected cells from two major YAP-expressing pancreatic cell types that are not quiescent and exhibit relative proliferative activity: ductal and stellate cells. Based on the cutoff for YAP expression, we categorized these cells into two subpopulations: “High YAP” and “Low YAP”. GSEA of Hippo signaling confirmed the enrichment of YAP target genes in the High YAP versus Low YAP groups. Further GSEA analysis revealed that pathways and processes directly linked to proliferation - such as positive regulation of cell population proliferation, DNA replication, mitotic cell cycle, and mRNA transcription - were highly enriched in the CVB4-infected High YAP subpopulation compared to the Low YAP subpopulation in both ductal and stellate cells. This finding suggests a clear correlation between YAP expression and proliferation in CVB4-infected cells. These data align with the point that CVB replication is more efficient in actively dividing cells, which are enriched for high YAP expression.

Also, YAP naturally enhances anti-apoptotic mechanisms as shown by us through upregulating components of the Trx system in the β -cells¹⁷. In line with this and under normal conditions without viral infection, YAP does not induce apoptosis, as confirmed in our current study. Indeed, in direct comparisons, YAP only became pro-apoptotic in the presence of the virus. Interestingly, when using the replication-deficient dsRNA viral mimic poly(I:C) under the same conditions, YAP inhibited poly(I:C)-induced apoptosis (Figure 7A,B). This highlights that YAP's protective effect on host cells creates a supportive environment to viral replication, ultimately leading to increased cell death. This mechanism is further illustrated by our use of the antiviral agent Pleconaril, which prevents enteroviral replication by binding to the viral capsid protein VP1, thereby blocking viral RNA exposure. Pleconaril effectively prevented CVB4 replication in human islets (Figure 7F) and fully inhibited YAP-induced inflammation under enteroviral infection (Figure 7E). Thus, the mechanism we present shows that YAP is inherently pro-proliferative, which serves as the primary driver of YAP-mediated viral replication and amplification. However, YAP-induced MST1 activation through a negative feedback loop increases cell death in infected cells. Genetic inhibition of MST1 reduced YAP-induced apoptosis in infected cells (please also refer to our response to Reviewer 1).

3. The authors show that infection of human islets with CVB3 and CVB4 induced a potent type I interferon response, and YAP overexpression further enhanced it. The authors need to provide mechanisms for how YAP synergistically enhances immune responses.

Response:

As discussed in the manuscript, our data show that YAP upregulates the interferon response during CVB infection, which may seem paradoxical given YAP's inhibitory role in the antiviral response. However, we now present complementary data supporting the notion that YAP-induced viral replication is the primary mechanism driving the upregulation of the innate immune response.

1. Using replication-deficient viral dsRNA Poly I:C (which mimics viral infection without inducing replication), we demonstrate that YAP does not potentiate the CVB-induced innate immune response and, in some donors, even decreases it (Figures 7C,D and S8). This finding aligns with previously established data showing that YAP cell-autonomously suppresses the innate immune response by blocking TBK1 signaling, an upstream regulator of IRF3-IFN signaling^{18,19}.
2. We now observe that YAP's pro-inflammatory effect is fully dependent on CVB replication, as evidenced by its inhibition through Pleconaril (Figure 7E,F). Pleconaril effectively inhibits CVB replication in human islets model of CVB infection. Notably, the use of Pleconaril has been linked to the preservation of residual insulin production in new-onset T1D patients in a proof-of-concept clinical study⁴. In vitro, Pleconaril efficiently blocks viral replication, reinforcing our conclusion that YAP enhances immune responses through its role in promoting viral replication.

These new findings clarify the mechanism by which YAP synergistically enhances immune responses during CVB infection, providing further insight into the interplay between viral replication, YAP activity, and inflammation. Both pro-and anti-inflammatory effects have been observed by YAP under different scenario²⁰. For example, YAP enhances the expression of pro-inflammatory cytokines through the YAP-TEAD-binding motif in the promoter region of inflammatory cytokines²¹, shown before in a non-alcoholic steatohepatitis (NASH) model.

This is included now in the discussion.

4. The authors showed an increase in total MST1 protein level in the YAP-overexpressing or CVB4 infected β -cells and suggested that it serves as a negative feedback loop. It would be interesting to know how CVB4 infection enhances, and both CVB4 infection and YAP OE synergistically increases the MST level.

Response:

MST1 is upregulated under conditions of high YAP activity, both triggered by CVB infection (Figure 5) or endogenously induced by YAP overexpression (Figure 9). This aligns with the model of a YAP-driven feedback loop, wherein MST1 acts as a compensatory regulator to restrain excessive YAP activation. As outlined above, we propose that the transient upregulation of YAP induced by enterovirus infection is sufficient to trigger MST1 feedback. Notably, inhibition of MST1 further enhances CVB replication, as evidenced by increased intracellular viral RNA and capsid protein levels (Figure 10). This suggests that MST1 upregulation acts as a critical checkpoint to limit viral replication driven by YAP. We performed additional experiments to address the question whether both CVB4 infection and YAP overexpression synergistically increase the MST1 level. While they independently had a consistent effect; CVB4 as well as YAP-OE induced MST1, there was no synergistical increase in MST1 (see new Figure S11F).

5. It is curious to know that ectopic expression of YAP increased the level of MST in a pancreatic cell-specific manner. Usually, ectopic expression of YAP does not increase the level of MST in other cell lines.

Response:

We thank the reviewer for this interesting comment. While previous work has identified known feedback mechanisms within the pathway, such as YAP-LATS²² or YAP-miR-YAP²³, our identification of the YAP-MST1 negative feedback loop is novel and represents a unique contribution to the field. To address the reviewer's point and to investigate whether this feedback loop is specific to β -cells or a more universal mechanism, we performed additional experiments in HeLa (cervical) and HEK293 (kidney) cells. Our new findings suggest a distinct specificity in the upregulation of YAP-induced MST1 within β -cells. This phenomenon was not

observed in other cell types, where STK4 mRNA expression and total MST1 protein levels remained unchanged upon YAP overexpression (new Figure S12).

References

1. Yang, L., *et al.* Human vascularized macrophage-islet organoids to model immune-mediated pancreatic β cell pyroptosis upon viral infection. *Cell Stem Cell* **31**, 1612-1629. e1618 (2024).
2. Geravandi, S., Richardson, S., Pugliese, A. & Maedler, K. Localization of enteroviral RNA within the pancreas in donors with T1D and T1D-associated autoantibodies. *Cell Rep Med* **2**, 100371 (2021).
3. Ardestani, A., *et al.* MST1 is a key regulator of beta cell apoptosis and dysfunction in diabetes. *Nat Med* **20**, 385-397 (2014).
4. Krogvold, L., *et al.* Pleconaril and ribavirin in new-onset type 1 diabetes: a phase 2 randomized trial. *Nat Med* **29**, 2902-2908 (2023).
5. Berg, A.K., Olsson, A., Korsgren, O. & Frisk, G. Antiviral treatment of Coxsackie B virus infection in human pancreatic islets. *Antiviral Res* **74**, 65-71 (2007).
6. George, N.M., Day, C.E., Boerner, B.P., Johnson, R.L. & Sarvetnick, N.E. Hippo signaling regulates pancreas development through inactivation of Yap. *Mol Cell Biol* **32**, 5116-5128 (2012).
7. Gao, T., *et al.* Hippo signaling regulates differentiation and maintenance in the exocrine pancreas. *Gastroenterology* **144**, 1543-1553, 1553 e1541 (2013).
8. George, N.M., Boerner, B.P., Mir, S.U., Guinn, Z. & Sarvetnick, N.E. Exploiting Expression of Hippo Effector, Yap, for Expansion of Functional Islet Mass. *Mol Endocrinol* **29**, 1594-1607 (2015).
9. Oshima, M., *et al.* Virus-like infection induces human beta cell dedifferentiation. *JCI Insight* **3**(2018).
10. Hodik, M., *et al.* Enterovirus infection of human islets of Langerhans affects beta-cell function resulting in disintegrated islets, decreased glucose stimulated insulin secretion and loss of Golgi structure. *BMJ Open Diabetes Res Care* **4**, e000179 (2016).
11. Gallagher, G.R., *et al.* Viral infection of engrafted human islets leads to diabetes. *Diabetes* **64**, 1358-1369 (2015).
12. Nyalwidhe, J.O., *et al.* Coxsackievirus-Induced Proteomic Alterations in Primary Human Islets Provide Insights for the Etiology of Diabetes. *J Endocr Soc* **1**, 1272-1286 (2017).
13. Yoon, J.W., Onodera, T., Jenson, A.B. & Notkins, A.L. Virus-induced diabetes mellitus. XI. Replication of coxsackie B3 virus in human pancreatic beta cell cultures. *Diabetes* **27**, 778-781 (1978).
14. Blum, S.I., *et al.* MDA5-dependent responses contribute to autoimmune diabetes progression and hindrance. *JCI Insight* **8**(2023).
15. Horwitz, M.S., *et al.* Diabetes induced by Coxsackie virus: initiation by bystander damage and not molecular mimicry. *Nat Med* **4**, 781-785 (1998).
16. Yu, F.X. & Guan, K.L. The Hippo pathway: regulators and regulations. *Genes Dev* **27**, 355-371 (2013).
17. Yuan, T., *et al.* Proproliferative and antiapoptotic action of exogenously introduced YAP in pancreatic beta cells. *JCI Insight* **1**, e86326 (2016).
18. Zhang, Q., *et al.* Hippo signalling governs cytosolic nucleic acid sensing through YAP/TAZ-mediated TBK1 blockade. *Nat Cell Biol* **19**, 362-374 (2017).

19. Munoz-Wolf, N. & Lavelle, E.C. Hippo interferes with antiviral defences. *Nat Cell Biol* **19**, 267-269 (2017).
20. Chen, L., Jin, X., Ma, J., Xiang, B. & Li, X. YAP at the progression of inflammation. *Front Cell Dev Biol* **11**, 1204033 (2023).
21. Song, K., *et al.* Yes-Associated Protein in Kupffer Cells Enhances the Production of Proinflammatory Cytokines and Promotes the Development of Nonalcoholic Steatohepatitis. *Hepatology* **72**, 72-87 (2020).
22. Moroishi, T., *et al.* A YAP/TAZ-induced feedback mechanism regulates Hippo pathway homeostasis. *Genes Dev* **29**, 1271-1284 (2015).
23. Zhang, Z.W., *et al.* miR-375 inhibits proliferation of mouse pancreatic progenitor cells by targeting YAP1. *Cell Physiol Biochem* **32**, 1808-1817 (2013).

Reponses to Reviewer 3:

Geravandi et al. have extensively revised the manuscript in response to the reviewers' comments. They have performed many experiments to answer my questions, many of which were common questions raised by other reviewers, and I really appreciate the authors' efforts. However, although the authors provide a lot of new data, in many cases they have not provided the specific answer to my question. In the first review, I wrote that the quality of their data was good. I did not ask for more data to confirm that CVB infection increases YAP mRNA levels. I accept their findings that CVB infection increases YAP levels and that increased YAP enhances CVB-mediated islet inflammation and b-cell apoptosis.

Response:

We thank the reviewer for their thoughtful comments on our revised manuscript. We greatly appreciate the recognition of our extensive efforts to address the concerns raised, as well as the positive feedback regarding the quality of our data and the additional experiments performed. We are pleased that the reviewer acknowledges our findings demonstrating that CVB infection increases YAP levels and that YAP contributes to CVB-mediated islet inflammation and β -cell apoptosis. Below, we address the remaining concerns in detail.

However, my point was that the authors need to provide a mechanism for the large gap between CVB infection and the increase in YAP levels. I also asked the authors to provide mechanisms for how YAP enhances CVB-mediated islet inflammation and b-cell apoptosis.

To my first questions, the authors replied that the transcriptional upregulation of YAP and the resulting increase in total YAP protein is the primary mechanism. I don't think their answer is a correct answer to my question.

Response:

As a first step toward addressing this question, it was essential to determine the regulatory level at which CVB affects YAP expression. YAP, like many transcriptional regulators, is subject to multilayered control - including transcriptional, post-transcriptional, and post-translational mechanisms. To define the dominant regulatory mechanism in our system, we performed a comprehensive set of experiments. Our data clearly demonstrate that CVB infection increases YAP expression at the transcriptional level, as shown by elevated *Yap1* mRNA and downstream targets detected by qPCR, confirmed by RNAscope double-labelling for *Yap1* and *CTGF* in human islets and endocrine/exocrine co-cultures, increased total YAP protein levels by Western blotting (Figure 5), and further supported by RNA-seq data from CVB4-infected stem cell-derived β -cells (Figure S6).

These findings provide direct evidence of transcriptional upregulation of YAP in human pancreatic tissues following CVB infection. These data are consistent with our human T1D tissue observations, where we detect widespread *YAP1* mRNA upregulation and co-localization of CVB RNA in *YAP1*-expressing cells. This seems a vicious cycle of CVB infection promoting YAP expression and in turn YAP promoting inflammation and rapid β -cell destruction seen in T1D. As shown in Figures 10 and S11, we observed that MST1 induction during later stages of infection may represent a compensatory cellular response, limiting YAP activity via canonical Hippo signaling

and acting as a checkpoint in the cycle of CVB-induced β -cell apoptosis and islet inflammation.

To provide a more detailed mechanistic framework as requested, we have now expanded the discussion in the revised manuscript to include several plausible upstream pathways through which CVB infection may induce *Yap1* transcription:

1. **Activation of the unfolded protein response (UPR):** CVB-induced ER stress and UPR activation via PERK is likely to increase *Yap1* transcription, as PERK knockdown has been shown to reduce ER stress-induced *Yap1* mRNA increases ¹, and enteroviruses are known to activate the PERK pathway ².
2. **Modulation of transcription factors such as GABP:** The GABP β transcription factor complex, known to regulate *Yap1* promoter activity ³ and implicated in viral infections such as HIV-1 ⁴, may also be activated during CVB infection to promote *Yap1* transcription.
3. **Disruption of microRNA-mediated repression:** YAP mRNA turnover is tightly regulated by microRNAs. CVB infection has been shown to alter host microRNA expression in human islets, including the downregulation of miR-149-5p ⁵, whose inhibition has been reported to relieve repression of *Yap1*, resulting in increased *Yap1* mRNA expression ⁶.
4. **Innate immune and inflammatory signaling:** CVB activates inflammatory transcription factors, and interferon-stimulated pathways ⁷, all of which may enhance *Yap1* transcription either directly or by modifying chromatin accessibility at its promoter.

We also acknowledge that post-transcriptional mechanisms could contribute to increased YAP protein levels in CVB-infected cells. For example, disruption of the Hippo pathway via loss of cell polarity and tight junction integrity by viral infections ⁸ could inhibit MST1/2 and LATS1/2 kinase activity. This would reduce YAP phosphorylation, thereby promoting its stabilization and nuclear accumulation.

Collectively, these mechanisms provide a biologically plausible framework for how CVB infection may regulate YAP expression at multiple levels. While the full dissection of these pathways is beyond the scope of this manuscript, we have added this expanded mechanistic context to the Discussion to better reflect current knowledge and to highlight ongoing research in our lab focused on these upstream drivers.

We hope this expanded explanation, along with the addition of supporting mechanisms in the revised manuscript, now fully addresses the reviewer's concern.

I have more concerns about the authors' response to my second point. Basically, the authors replied that YAP overexpression provides a rich environment for efficient CVB replication and amplification, based on the data obtained using poly(I:C) infection and Pleconaril treatment. Based on their experimental data, they concluded that the pro-apoptotic and pro-inflammatory effects of YAP were dependent on viral replication. I could not understand their logic. As shown in Figure 7C, overexpression of YAP alone did not induce an inflammatory response. If Pleconaril was given after viral infection to inhibit viral replication, wouldn't the situation be the same as if only YAP was overexpressed? Also, poly(I:C) infection is in some ways similar to CVB infection, but I cannot accept the conclusion that the lack of potentiation of the inflammatory response by poly(I:C) is due to the lack of replication ability of poly(I:C). It is possible that some other factors in CVB, not present in poly(I:C), may potentiate the inflammatory response.

We appreciate the reviewer's thoughtful engagement with our data. Below, we clarify the logic behind our conclusion that the pro-apoptotic and pro-inflammatory effects of YAP are dependent on active CVB replication, and address specific points raised.

1. "Overexpression of YAP alone did not induce an inflammatory response. If Pleconaril was given after viral infection to inhibit viral replication, wouldn't the situation be the same as if only YAP was overexpressed?"

Response:

We agree that YAP overexpression alone does not induce an inflammatory response (Figure 7C). In the absence of CVB, YAP exhibits its canonical anti-apoptotic and pro-proliferative functions. However, the cellular environment changes in the context of active CVB infection and replication. Pleconaril is a capsid inhibitor that prevents uncoating and subsequent release of viral RNA, thereby rendering the virus replication-deficient. While it allows initial viral entry, it blocks productive replication. As a result, the virus does not progress beyond early stages, and no new virions are produced. In this context, the cellular outcome resembles that of YAP overexpression in the absence of productive viral replication: the expected pro-inflammatory and pro-apoptotic phenotypes are not observed. This is indeed different from active CVB infection, where viral replication creates a dynamic interaction with host factors, including YAP. Thus, Pleconaril-treated infections and poly(I:C) stimulation provide crucial control conditions where the absence of viral replication demonstrates the requirement for viral amplification in eliciting YAP-dependent inflammation and apoptosis.

2. "Poly(I:C) infection is in some ways similar to CVB infection, but I cannot accept the conclusion that the lack of potentiation of the inflammatory response by poly(I:C) is due to the lack of replication ability of poly(I:C)."

Response:

Yes. Poly(I:C), while a widely accepted mimic of viral RNA, does not fully recapitulate all aspects of live viral infection. However, our aim was not to equate poly(I:C) with CVB, but rather to use it as a proxy for viral RNA in the absence of replication, similar to Pleconaril-treated CVB. Poly(I:C) activates pattern recognition receptors such as MDA5 and TLR3, which are also engaged during CVB infection. Nevertheless, poly(I:C) is a synthetic analog and cannot replicate or generate viral proteins or particles. Thus, any cellular response to poly(I:C) represents the early innate immune activation phase, uncoupled from the downstream consequences of active replication. Our data clearly show that in both poly(I:C)-treated and Pleconaril-treated conditions, YAP does not potentiate apoptosis or inflammation. Only in the context of replicating virus does YAP promote β -cell apoptosis and hyperinflammatory responses. This suggests that the replication step - and not merely the presence of viral RNA - is essential to shift YAP's function from protective to pathogenic.

3. "It is possible that some other factors in CVB, not present in poly(I:C), may potentiate the inflammatory response."

Response:

CVBs possess unique viral proteins and replication intermediates that poly(I:C) does not recapitulate. We have not excluded the possibility that CVB-specific factors

contribute to the observed inflammation. However, our study focuses specifically on the requirement for active viral replication as a prerequisite for YAP's pro-inflammatory effects as discussed above. This is supported by published data showing that UV-inactivated CVB, which cannot replicate, fails to induce β -cell death and shows limited induction of proinflammatory cytokines (see discussion; refs 21 and 56). Taken together, these findings, along with our data, strongly argue that the inflammatory response observed in the context of YAP overexpression is tightly coupled to viral replication. We have now added clarifications to the revised Discussion section to acknowledge that while viral replication is a key driver, other CVB-specific factors absent in poly(I:C) may also contribute to the full inflammatory phenotype.

If the pro-proliferative property of YAP is the main mechanism for YAP-mediated viral replication and amplification, they should have performed an experiment to test whether overexpression of a pro-proliferative oncogene such as c-myc can enhance the CVB-mediated inflammatory response to prove their hypothesis.

Response:

We thank the reviewer for this suggestion. The hypothesis that pro-proliferative signaling enhances viral replication is indeed well-supported in the literature ⁹. For example, overexpression of c-Myc has been shown to promote replication of viruses such as SV40 ¹⁰, Kaposi's sarcoma-associated herpesvirus (KSHV) ¹¹, and HIV ¹², partly by creating a cellular environment more permissive to viral replication. Once survival and proliferative pathways are activated, host cells provide a more favorable niche for viral propagation, which can in turn amplify inflammatory responses. This is also supported by our own reanalysis of publicly available single-cell RNA-seq data from CVB-infected pancreatic cells, where we observed a strong correlation between YAP expression and proliferative signatures (Figure S10). Notably, Myc was among the genes enriched in the GSEA analysis of proliferation pathways in high YAP CVB4-infected pancreatic cells. These data reinforce the concept that CVB replication is more efficient in actively dividing cells with high YAP activity.

Given that this mechanism is already broadly established, we believe that repeating the experiment using Myc in our model would not yield additional mechanistic insight beyond what is already known. Demonstrating a similar effect with another proliferative gene would support a generalizable principle, but would not specifically advance our central conclusions regarding YAP's unique role in promoting CVB-induced β -cell inflammation and apoptosis.

We have now included references to these studies in the revised manuscript to place our findings within this broader biological context and to strengthen our interpretation of the YAP-driven phenotype.

1. Wu, H., *et al.* Integration of Hippo signalling and the unfolded protein response to restrain liver overgrowth and tumorigenesis. *Nat Commun* **6**, 6239 (2015).
2. Hirano, J., *et al.* Enterovirus 3A protein disrupts endoplasmic reticulum homeostasis through interaction with GBF1. *J Virol* **98**, e0081324 (2024).
3. Wu, H., *et al.* The Ets transcription factor GABP is a component of the hippo pathway essential for growth and antioxidant defense. *Cell Rep* **3**, 1663-1677 (2013).

4. Granberg, F., Svensson, C., Pettersson, U. & Zhao, H. Adenovirus-induced alterations in host cell gene expression prior to the onset of viral gene expression. *Virology* **353**, 1-5 (2006).
5. Kim, K.W., *et al.* Coxsackievirus B5 Infection Induces Dysregulation of microRNAs Predicted to Target Known Type 1 Diabetes Risk Genes in Human Pancreatic Islets. *Diabetes* **65**, 996-1003 (2016).
6. Li, J., *et al.* YY1-induced DLEU1/miR-149-5p Promotes Malignant Biological Behavior of Cholangiocarcinoma through Upregulating YAP1/TEAD2/SOX2. *Int J Biol Sci* **18**, 4301-4315 (2022).
7. Lin, X., *et al.* Viral infection induces inflammatory signals that coordinate YAP regulation of dysplastic cells in lung alveoli. *J Clin Invest* **134**(2024).
8. Thomas, M. & Banks, L. Upsetting the Balance: When Viruses Manipulate Cell Polarity Control. *J Mol Biol* **430**, 3481-3503 (2018).
9. Thai, M., *et al.* MYC-induced reprogramming of glutamine catabolism supports optimal virus replication. *Nat Commun* **6**, 8873 (2015).
10. Classon, M., Henriksson, M., Sumegi, J., Klein, G. & Hammarskjold, M.L. Elevated c-myc expression facilitates the replication of SV40 DNA in human lymphoma cells. *Nature* **330**, 272-274 (1987).
11. Li, X., Chen, S., Feng, J., Deng, H. & Sun, R. Myc is required for the maintenance of Kaposi's sarcoma-associated herpesvirus latency. *J Virol* **84**, 8945-8948 (2010).
12. Sun, Y. & Clark, E.A. Expression of the c-myc proto-oncogene is essential for HIV-1 infection in activated T cells. *J Exp Med* **189**, 1391-1398 (1999).